# SNAP: Low-Latency Test-Time Adaptation with Sparse Updates

**Hyeongheon Cha[1]  Dong Min Kim[1]  Hye Won Chung[1]  Taesik Gong[2]\*  Sung-Ju Lee[1]\***

[1]School of Electrical Engineering, KAIST, Daejeon, Republic of Korea
[2]Department of Computer Science and Engineering, UNIST, Ulsan, Republic of Korea

{hyeongheon,dongmin.kim,hwchung,profsj}@kaist.ac.kr
taesik.gong@unist.ac.kr

## Abstract

Test-Time Adaptation (TTA) adjusts models using unlabeled test data to handle dynamic distribution shifts. However, existing methods rely on frequent adaptation and high computational cost, making them unsuitable for resource-constrained edge environments. To address this, we propose SNAP, a sparse TTA framework that reduces adaptation frequency and data usage while preserving accuracy. SNAP maintains competitive accuracy even when adapting based on only 1% of the incoming data stream, demonstrating its robustness under infrequent updates. Our method introduces two key components: (i) Class and Domain Representative Memory (CnDRM), which identifies and stores a small set of samples that are representative of both class and domain characteristics to support efficient adaptation with limited data; and (ii) Inference-only Batch-aware Memory Normalization (IoBMN), which dynamically adjusts normalization statistics at inference time by leveraging these representative samples, enabling efficient alignment to shifting target domains. Integrated with five state-of-the-art TTA algorithms, SNAP reduces latency by up to 93.12%, while keeping the accuracy drop below 3.3%, even across adaptation rates ranging from 1% to 50%. This demonstrates its strong potential for practical use on edge devices serving latency-sensitive applications. The source code is available at https://github.com/chahh9808/SNAP.

## 1  Introduction

Deep learning models often suffer performance degradation under domain shifts caused by environmental changes or noise [37]. Test-Time Adaptation (TTA) offers a promising solution for domain shifts by utilizing only unlabeled test data without requiring source data. While TTA algorithms have advanced in complexity to improve accuracy in data streams [48, 30, 50, 53, 31, 43], they are typically designed for resource-rich servers, overlooking the computational limitations crucial for real-world deployment. Operations such as backpropagation, data augmentation, and model ensembling [50, 53, 55] result in substantial latency and memory consumption, making state-of-the-art (SOTA) TTA methods inefficient for practical use.

For edge devices with limited computational power, such as mobile devices or IoT sensors, the adaptation latency from TTA methods becomes a critical bottleneck, particularly in delay-sensitive applications such as autonomous driving and real-time health monitoring. Moreover, the model must keep up with the data stream in those applications, but high computational overhead could cause it to miss critical samples, resulting in inference lags and reduced accuracy. This issue is exacerbated with fast data streams, such as high-frame-rate videos or high-performance sensors. For example, even a

---

\*Corresponding authors.

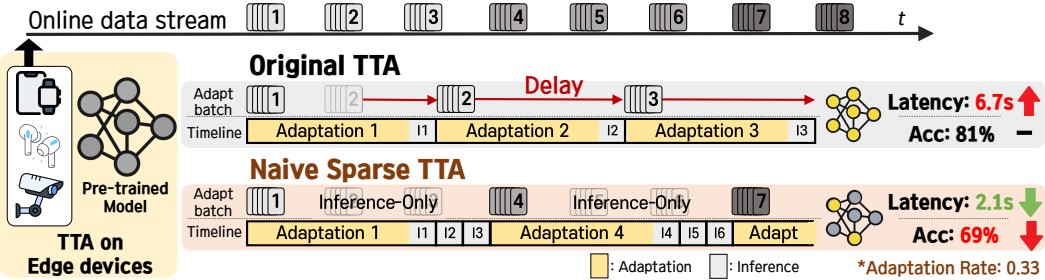

Figure 1: Comparison of average latency per batch and accuracy between the Original and Naïve Sparse TTA approaches on edge devices processing an online data stream. With an adaptation rate of 0.33, adaptation occurs once every three batches, reducing latency proportional to the rate but leading to a significant accuracy drop compared with fully adapting Original TTA.

slight delay in processing sensor data can lead to dangerous situations in autonomous driving. A high adaptation latency accumulating with each batch not only undermines real-time performance but also limits the potential of TTA algorithms in latency-sensitive applications (Section 3). In response to this challenge, forward-only TTA methods [13, 29] propose adjusting lightweight components such as prototypes or prompts during inference. While these mechanisms improve efficiency, their reliance on fixed model parameters fails to adapt to dynamic distribution shifts [7, 53, 29]. Consequently, a robust performance of backpropagation-based approaches across diverse conditions (Appendix B.10) underscores the necessity of efficient model updates.

In online TTA scenarios where rapid response is required under strict resource constraints, *Sparse TTA (STTA)* offers a practical compromise by reducing the frequency of adaptation rather than eliminating it entirely. By adapting intermittently instead of at every batch, STTA significantly reduces computational overhead and latency. However, naïvely reducing update frequency degrades performance since only a limited data portion is utilized (Figure 1). Thus, the success of STTA hinges on strategically selecting representative stream samples to enable effective adaptation under sparse update schedules (detailed analysis in Section 5).

Existing sampling-based TTA methods are especially designed for handling dynamic data stream such as non-i.i.d. [6, 31, 53] or noisy data [7]. However, they are not optimized for data efficiency and continue to utilize a large proportion of samples. For example, EATA [30] reduces sample usage by filtering out unreliable samples but experiences performance degradation when reductions become too aggressive. Meanwhile, research in data-efficient deep learning has shown that selecting easy, class-representative samples is effective at low sampling rates (e.g., below 0.4) [51, 2]. However, these approaches depend on ground-truth labels, which are not available in TTA.

We propose **SNAP**: **S**parse **N**etwork **A**daptation for **P**ractical Test-Time Adaptation, a low-latency unsupervised domain adaptation framework for resource-constrained devices. SNAP balances adaptation accuracy with computational efficiency through two key components: Class and Domain Representative Memory (CnDRM) and Inference-only Batch-aware Memory Normalization (IoBMN). CnDRM stores a pool of *class-representative* samples (high pseudo-label confidence, balanced across predictions) and *domain-representative* samples (closest to the target domain centroid in feature space). This approach enables the model to adapt effectively to domain shifts with minimal data (Section 4.1). Meanwhile, IoBMN dynamically refines the normalization layers during inference by utilizing CnDRM's class-domain representative statistics to correct skewed feature distributions at each inference step. This keeps the model aligned with the evolving data distribution, enabling effective batch-wise adaptation without backpropagation (Section 4.2).

SNAP is a lightweight module that reduces latency while seamlessly integrating with existing TTA methods, preserving their adaptation behavior. To assess its effectiveness, we integrated SNAP with five SOTA TTA algorithms: Tent [48], EATA [30], SAR [31], CoTTA [50], and RoTTA [53]. We tested it on three widely-used TTA benchmarks (CIFAR10-C, CIFAR100-C, and ImageNet-C [10]) across various adaptation rates (Section 5). We also validate SNAP on ImageNet-R [9] and ImageNet-Sketch [49] to assess generalization (Appendix B.11).

In addition, we measured SNAP's latency and memory usage on three popular edge devices—Raspberry Pi 4 [38], Raspberry Pi Zero 2 W [39], and NVIDIA Jetson Nano [32]—to assess its real-world applicability. SNAP significantly reduces latency while minimizing performance degradation from existing TTA methods. On a Raspberry Pi 4 testbed, it reduced CoTTA's latency by

up to 93.12% at an adaptation rate of 0.1 in CIFAR10-C, with no loss in performance. Moreover, SNAP maintained performance comparable to original TTA methods across adaptation rates from 0.01 to 0.5, achieving 77.12%–81.74% for Tent, close to the full adaptation accuracy of 80.43%. SNAP also operates efficiently under memory constraints, with low memory overhead and seamless integration with a memory-efficient TTA module [12], as detailed in Appendices B.7 and B.8.

## 2 Related work

**Test-time adaptation.** Test-time adaptation aims to improve model performance on out-of-distribution data by using only the unlabeled test data stream to adapt the model. Test-time normalization [28, 40] adjusts the batch normalization (BN) statistics using test data to improve performance. Other works mainly involve updating the parameters of the model during test time. Tent [48] adapts the affine parameters of the BN layers to minimize the entropy of its predictions. EATA [30] builds upon Tent, sampling reliable and non-redundant samples and utilizing an anti-forgetting regularizer for efficiency. Other works introduce more complex schemes, primarily to improve robustness against more practical test-time scenarios. CoTTA [50] addresses a continually changing test-time environment by using weight-averaged and augmentation-averaged predictions with stochastic restoring. SAR [31] filters samples with large and noisy gradients to stabilize the model during wilder test-time scenarios. RoTTA [53] targets a practical test-time setting of changing distributions and correlative sampling by introducing a memory bank and a teacher-student model. We further analyze how prior TTA methods perform under sparse-update regimes and how our approach differs in Appendix B.3.

**Test-time adaptation on edge devices.** TTA on edge devices primarily inherit the challenges of on-device learning:, including limited memory and reduced computational efficiency [21]. Several memory-efficient TTA works have been proposed in this regard. MECTA [12] aims to reduce the memory consumption of gradient-based TTA, proposing an adaptive normalization layer to reduce the intermediate caches for backpropagation. EcoTTA [43] proposes memory-efficient continual TTA by adapting lightweight meta networks instead of the originals to reduce the size of intermediate activations. Despite works to promote memory-efficiency, the latency of TTA, especially on resource-constrained edge devices, has been generally overlooked. While many adaptation-based TTA [48, 30, 31, 53] update only the affine parameters for general time and memory concerns, they still involve computationally-heavy operations every batch, which can lead to high latency on edge devices. A recent work [1] introduces a more practical TTA evaluation protocol that penalizes slow TTA methods by providing them with fewer samples for adaptation.

**Data-efficient deep learning.** Data-efficient deep learning methods enable deep learning models to achieve competitive performance with less data. Among these methods, data selection, or data sampling, involves utilizing a small subset of the training data in an attempt to match that of full-dataset training. A branch of data-selection is score-based selection, which scores each sample based on some predefined metric, such as a sample's influence [16], difficulty [46, 34], prediction confidence [35], or consistency [14], and selects samples with scores in a certain range. Another set of data-selection methods involves optimization-based selection, which formulates an optimization problem to find an optimal subset that can best approximate full-dataset training [26, 52, 36]. While these approaches work well in their preconceived settings, they generally suffer performance drop as their settings change, such as a change in sampling rates. More recent studies such as Moderate Coreset [51] propose a more robust selection approach by using the distance of a sample to the class center as a score criterion, for an effective representation of the dataset.

## 3 Preliminaries

Our work addresses the challenge of test-time adaptation latency on edge devices, where efficient, low-latency inference must be achieved despite limited resources.

**Test-time adaptation and its latency challenge on edge devices.** In unsupervised domain adaptation, the source domain data $\mathcal{D}_\mathcal{S} = \mathcal{X}^\mathcal{S}, \mathcal{Y}$ is drawn from the distribution $P_\mathcal{S}(\mathbf{x}, y)$, while the target domain data $\mathcal{D}_\mathcal{T} = \mathcal{X}^\mathcal{T}, \mathcal{Y}$ follows $P_\mathcal{T}(\mathbf{x}, y)$, typically without known labels $y_j$. Given a pre-trained model $f(\cdot; \Theta)$ on the source domain $\mathcal{D}_\mathcal{S}$, TTA adjusts the model to the target distribution $P_\mathcal{T}$ using only target instances $\mathbf{x}_j$, updating the parameters $\Theta$ to reduce domain discrepancy [48].

On resource-constrained edge devices, frequent adaptation poses a significant bottleneck. Our experiments on a Raspberry Pi 4 [38] revealed that existing TTA methods incur a minimum latency of 3.83 seconds per batch (Figure 2), severely limiting real-time inference for fast data streams (e.g., autonomous driving [45, 22]). Additional latency tracking for other devices is reported in Appendix B.6. Even lightweight TTA algorithms suffer from considerable back-propagation overhead, creating bottlenecks on resource-constrained devices without GPU-level computation. More computationally intensive methods like CoTTA, which depend on data augmentation and ensembles, require over 70 seconds per adaptation step, rendering them impractical for edge devices (Figure 2).

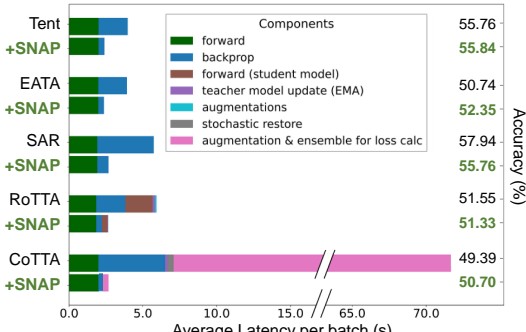

Figure 2: Component-wise latency and overall accuracy comparison between full SOTA TTA and SNAP (sparse update with frequency 0.1) on CIFAR100-C, measured on Raspberry Pi 4. SNAP matches accuracy with significantly lower cost.

A recent work [1] recognized latency as a crucial problem and proposed a TTA evaluation protocol that penalizes methods slower than the data stream rate. Instead of penalizing a model for being slow, we propose *Sparse TTA*, where the model adapts at sparse intervals to sustain real-time throughput. As real deployments involve devices with different computational capabilities and data streams of varying speeds, we believe a framework that effectively maintains various TTA methods' performance across different latency requirements is crucial.

**Sparse test-time adaptation and adaptation rates.**  Sparse Test-Time Adaptation (STTA) lowers the frequency of model updates, a key factor in reducing adaptation latency on resource-constrained devices. Unlike conventional TTA methods that process full batches and incur significant overhead, STTA updates the model using only a subset of batches (Figure 1). The core parameter of STTA, the *Adaptation Rate (AR)*, determines the proportion of batches or samples used for adaptation compared to the Original TTA. By tuning the AR, STTA balances the performance and computational latency. Furthermore, STTA's periodic adaptation can be optimized by strategically distributing sparse model updates across selected intervals during inference. This approach helps distribute the adaptation overhead, smooths latency fluctuations across inference batches, and preserves overall performance.

## 4  Methodology

SNAP framework resolves the high latency and inefficiency issue of existing TTA methods. By introducing a Sparse TTA (STTA) strategy combined with a novel sampling method, SNAP minimizes adaptation delays while maintaining accuracy. The overall system, illustrated in Figure 3, consists of two primary components: (i) Class and Domain Representative Memory (CnDRM) for efficient sampling and (ii) Inference-only Batch-aware Memory Normalization (IoBMN) to correct feature distribution shifts during inference. Together, these components enable effective STTA with minimal computational overhead.

### 4.1  Class and Domain Representative Memory

CnDRM is a core component of SNAP that addresses the challenges of efficient data sampling for STTA. As the adaptation rate directly impacts the number of samples used for adaptation, this necessitates a careful sampling strategy to optimize performance with minimal data. Given this limited sampling rate, CnDRM selects both class and domain-representative samples to maintain model performance while minimizing adaptation overhead.

**Motivation.**  Effective data sampling is essential for data-efficient deep learning, particularly when only a few samples are available. While score-based methods that prioritize difficult samples perform well at high sampling rates, selecting easy, class-representative samples is more effective at lower rates [2]. Moderate Coreset [51] also demonstrates that selecting samples near the class center improves performance in noisy-label settings, a principle that aligns with the STTA scenario where ground truth is unavailable.

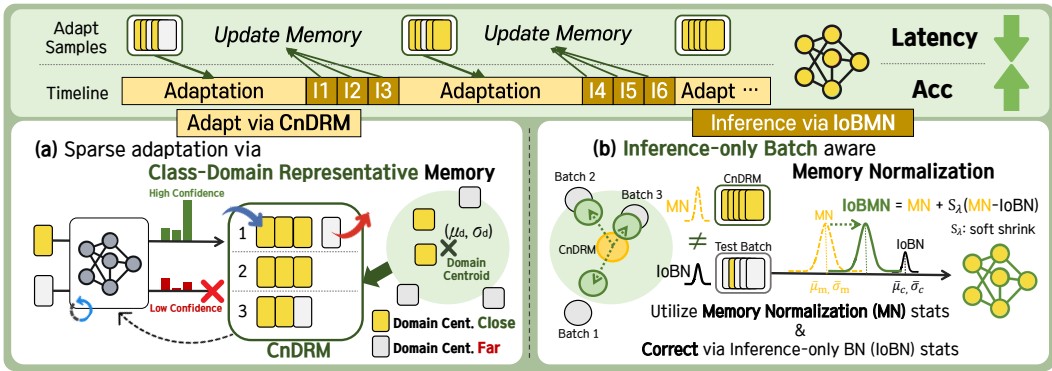

Figure 3: Design overview of SNAP. The framework consists of two primary components: (a) Class and Domain Representative Memory (CnDRM), which efficiently selects representative samples to minimize adaptation overhead, and (b) Inference-only Batch-aware Memory Normalization (IoBMN), which corrects feature distribution shifts during inference. Together, these components implement the Sparse TTA (STTA) strategy, reducing latency while maintaining model accuracy.

In addition, on real-world deployments, latency constraints often limit adaptation frequency, requiring models to function at low adaptation rates (e.g., 0.1). At such low rates, class-representative sampling alone is insufficient (Table 2), as it fails to capture distributional shifts between source and target domains. To overcome this limitation, we propose selecting both **class- and domain-representative samples** to enhance adaptation efficiency in low-data environments. Detailed theoretical analysis on the proposed efficient sampling strategy is in Appendix B.1.

**Criteria 1: class representation.**    To ensure stable adaptation without ground truth labels, CnDRM selects high-confidence samples, avoiding low-confidence samples that often lie near decision boundaries and carry incorrect pseudo-labels. This ensures stable learning signals and reduces error propagation from incorrect pseudo-labels, supporting more effective and stable adaptation (Details in Appendix B.5). The confidence score $C(\mathbf{x})$ for each sample $\mathbf{x}$ is calculated as: $C(\mathbf{x}) = \max_{y \in \mathcal{Y}} p(y|\mathbf{x}; \Theta)$ where $p(y|\mathbf{x}; \Theta)$ is the softmax probability for class $y$. Only samples with confidence above a threshold $\tau_{conf}$ are retained. For a balanced representation across diverse classes, CnDRM selects these high-confidence samples in a prediction-balanced manner. This helps maintain the model's overall classification capability by preventing bias towards certain classes when only a low sample rate is available for adaptation. By leveraging both high confidence and prediction balance, CnDRM effectively selects class-representative samples that are diverse and reliable, even without access to ground-truth labels.

**Criteria 2: domain representation.** In addition to class-representative sampling, Cn-DRM selects domain-representative samples to facilitate adaptation to new domain conditions. Building on the efficient class-representative sampling criteria, we argue that *selecting samples close to the domain centroid* would enhance performance in STTA. Our preliminary experiment results validate improved performance when selecting samples near the centroid (Figure 4). For ImageNet-C Gaussian noise, TTA with the closest 20% of samples achieved 26.65% accuracy, whereas the farthest 20% showed a lower accuracy of 18.52%.

As early layers in deep learning models tend to retain domain-specific features [54, 19, 42], we utilize the hidden features of early layers to identify domain-representative samples (Appendix B.4). Specifically, CnDRM uses the feature statistics (mean and variance) of the first normalization layer to assess domain representation, since domain discrepancies can be effectively mitigated through normaliza-

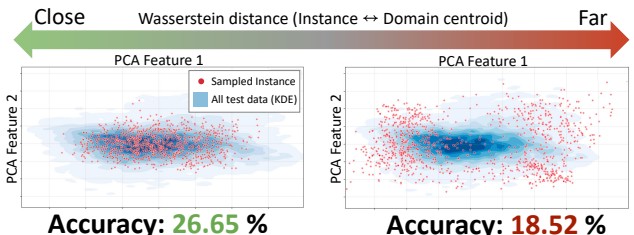

Figure 4: Sampling visualization and accuracy comparison between the closest 20% and farthest 20% samples from the domain centroid on ImageNet-C Gaussian noise.

tion adjustments using these statistics [28, 40]. Domain discrepancies in hidden features are substantially reduced after passing through a single normalization layer, significantly minimizing domain shift [20]. While deeper layers provide detailed information, using the first layer balances capturing domain-specific information and maintaining computational efficiency.

The domain centroid $\mathbf{c}_d$ is computed using a momentum-based update of batch statistics from the normalization layer: $\mu_{domain} \leftarrow (1 - \beta)\mu_{domain} + \beta\mu_t$ and $\sigma^2_{domain} \leftarrow (1 - \beta)\sigma^2_{domain} + \beta\sigma^2_t$, where $\mu_t$ and $\sigma^2_t$ are the mean and variance of the current batch $t$, and $\beta$ is the momentum parameter. In our preliminary study, we found that using only the mean and standard deviation values before the first normalization was sufficient to calculate the domain centroid. The sampled instances effectively represented the domain and were correctly positioned in the embedding space for each criterion (Figure 4).

We formally define a *domain-representative sample* as one whose early-layer feature statistics are closest to the domain centroid, as measured by the Wasserstein distance [47]. The Wasserstein distance quantifies the similarity between two distributions by considering their mean and variance, evaluating how well a sample represents the domain. It is useful for capturing domain characteristics, thus widely used in domain generalization [42]. Following common practice in domain adaptation [20, 27], we approximate channel-wise feature distributions as independent univariate Gaussians (i.e., with diagonal covariance) to efficiently estimate mean/variance-level domain shifts, which yields the following closed-form expression:

$$W(\mathbf{x_t}, \mathbf{c}_{domain}) = \sqrt{(\mu_{\mathbf{x_t}} - \mu_{domain})^2 + (\sigma_{\mathbf{x_t}} - \sigma_{domain})^2}. \tag{1}$$

For each sample $\mathbf{x_t}$, the feature statistics $(\mu_{\mathbf{x_t}}, \sigma_{\mathbf{x_t}})$ are taken from the input to the normalization layer. Further clarification and assumption details are provided in Appendix B.2.

---

**Algorithm 1** Class and Domain Representative Memory (CnDRM) Management

---

**Require:** test data stream $x_t$, memory $M$ with capacity $N$, confidence threshold $\tau_{conf}$, adaptation rate $1/k$
1: **for** batch $b \in \{1, \dots, B\}$ **do**
2: $\quad \hat{Y}_b \leftarrow f(b; \Theta)$
3: $\quad$ **for** each sample $x_t$ in batch $b$ **do**
4: $\quad\quad \hat{y}_t \leftarrow \hat{Y}_b[t]$
5: $\quad\quad$ confidence $\leftarrow C(x_t; \Theta)$
6: $\quad\quad \mathbf{c}_t(\mu_{\mathbf{x_t}}, \sigma_{\mathbf{x_t}}) \leftarrow$ mean & variance of early feature
7: $\quad\quad w_{x_t} \leftarrow W(x_t, \mathbf{c}_{domain})$
8: $\quad\quad$ **if** confidence $> \tau_{conf}$ **then**
9: $\quad\quad\quad$ Add $\mathbf{s}_t(x_t, \hat{y}_t, c_t, w_{x_t})$ to $M$ $\qquad\qquad\qquad$ ▷ Add class-representative samples
10: $\quad\quad\quad$ **if** $|M| > N$ **then**
11: $\quad\quad\quad\quad L^* \leftarrow$ class with most samples in $M$
12: $\quad\quad\quad\quad$ **if** $\hat{y}_t \notin L^*$ **then** $\qquad\qquad\qquad$ ▷ Remove domain-centroid farthest sample
13: $\quad\quad\quad\quad\quad \mathbf{s}_{farthest} \leftarrow \arg\max_{\mathbf{s}_i \in M \wedge \hat{y}_i \in L^*} w_{x_i}$
14: $\quad\quad\quad\quad$ **else**
15: $\quad\quad\quad\quad\quad \mathbf{s}_{farthest} \leftarrow \arg\max_{\mathbf{s}_i \in M \wedge \hat{y}_i = \hat{y}_t} w_{x_i}$
16: $\quad\quad\quad\quad$ Remove $\mathbf{s}_{farthest}$ from $M$
17: $\quad\quad \mathbf{c}_{domain} \leftarrow (1 - \beta)\mathbf{c}_{domain} + \beta\mathbf{c}_t$ $\qquad\qquad\qquad$ ▷ Update domain-centroid
18: $\quad\quad$ Recalculate $w_{\mathbf{s}_i}$ for all $\mathbf{s}_i$ in $M$
19: $\quad\quad$ **if** $b \mod k == 0$ **then** $\qquad\qquad\qquad$ ▷ Adaptation occurs every $k$ batches
20: $\quad\quad\quad$ Update model $\Theta$ using samples in $M$

---

**Memory management algorithm.** CnDRM maintains a compact yet adaptive memory that jointly preserves class balance and domain representativeness while keeping computational overhead minimal. To achieve this, the memory size is fixed to match the batch size for efficiency. Within this limited capacity, samples are managed so that each class remains well-represented while the overall memory distribution stays close to the domain centroid. Specifically, when the memory reaches its capacity, the farthest samples from the domain centroid (those with the largest Wasserstein distance) are replaced by new, high-confidence samples that better align with both class balance and domain characteristics. This joint management ensures that the memory continually retains the most class- and domain-representative samples under dynamic distribution shifts.

Algorithm 1 implements these procedures: lines 8~16 handle both *class balancing* and the *remove domain-centroid farthest sample* operation, where the least representative sample (i.e., the one with the largest Wasserstein distance within an overrepresented class) is discarded. Lines 17~20 perform the *update domain-centroid* operation using a momentum-based moving average (with $\beta = 0.9$) that enables the centroid to smoothly adapt to the evolving feature distribution. This linkage clarifies how CnDRM maintains a unified class-domain representative memory throughout continuous adaptation on edge devices.

## 4.2 Inference-only Batch-aware Memory Normalization

**Motivation.** Sparse Test-Time Adaptation (STTA) requires models to adapt to domain shifts with limited update opportunities. Consequently, stored adaptation batch statistics may become misaligned with subsequent inference data when updates are skipped. Traditional normalization methods, relying solely on test batch statistics, also struggle with such shifts. To address this, we propose Inference-only Batch-aware Memory Normalization (IoBMN), which stabilizes adaptation by leveraging representative memory statistics while selectively adjusting for distributional mismatches. This approach ensures both robustness and adaptability in STTA, significantly improving model stability, as demonstrated in our ablation study (Section 5).

**Approach.** Given a feature map $f \in \mathbb{R}^{B \times C \times L}$, where $B$ is the batch size, $C$ is the number of channels, and $L$ is the number of spatial locations, the batch-wise statistics $\bar{\mu}_c$ and $\bar{\sigma}_c^2$ for the $c$-th channel are calculated as follows:

$$\bar{\mu}_c = \frac{1}{B \times L} \sum_{b=1}^{B} \sum_{l=1}^{L} f_{b,c,l}, \quad \bar{\sigma}_c^2 = \frac{1}{B \times L} \sum_{b=1}^{B} \sum_{l=1}^{L} (f_{b,c,l} - \mu_{b,c}), \qquad (2)$$

where $\bar{\mu}_m$ and $\bar{\sigma}_m^2$ are calculated from the most recent adapted CnDRM samples in the same way with Equation 2, using the memory capacity $M$ with $m$ representing the memory. We assume that $\mu_m$ and $\sigma_m^2$ follow the *sampling distribution* of the feature map size $L$ and memory capacity $M$. The corresponding variances for the memory mean $\mu_m$ and variance $\sigma_m^2$ are calculated as:

$$s_{\mu_m}^2 := \frac{\bar{\sigma}_m^2}{L \times M}, \quad s_{\sigma_m^2}^2 := \frac{2\bar{\sigma}_m^4}{L \times M - 1}. \qquad (3)$$

For the normalization process to adapt efficiently to the current inference batch statistics, IoBMN corrects $(\bar{\mu}_m, \bar{\sigma}_m^2)$ only when $\bar{\mu}_c$ (and $\bar{\sigma}_c^2$) significantly differ from $\bar{\mu}_m$ (and $\bar{\sigma}_m^2$) through soft shrinkage function:

$$\mu_m^{\text{IoBMN}} = \bar{\mu}_m + S_\lambda(\bar{\mu}_c - \bar{\mu}_m; \alpha s_{\mu_m}), \quad (\sigma_m^{\text{IoBMN}})^2 = \bar{\sigma}_m^2 + S_\lambda(\bar{\sigma}_c^2 - \bar{\sigma}_m^2; \alpha s_{\sigma_m^2}), \qquad (4)$$

where $\alpha \geq 0$ in IoBMN controls the reliance on the normalization layer statistics. A larger $\alpha$ gives more weight to the last adapted memory normalization statistics, whereas a smaller $\alpha$ emphasizes the current inference batch normalization statistics. The soft shrinkage function $S_\lambda(x; \lambda)$ is defined as:

$$S_\lambda(x; \lambda) = \text{sign}(x) \cdot \max(|x| - \lambda, 0), \qquad (5)$$

where $\lambda$ is the threshold and $x$ is the input. The function allows for proportional adjustments based on the magnitude of the values, where smaller values are adjusted less and larger values more, preserving the critical information inherent in the adapted memory normalization statistics.

Finally, the output of the IoBMN for each feature $f_{b,c,l}$ is computed as:

$$\text{IoBMN}(f_{b,c,l}; \bar{\mu}_m, \bar{\sigma}_m^2, \mu_m^{\text{IoBMN}}, (\sigma_m^{\text{IoBMN}})^2) := \gamma \cdot \frac{f_{b,c,l} - \mu_m^{\text{IoBMN}}}{\sqrt{(\sigma_m^{\text{IoBMN}})^2 + \epsilon}} + \beta, \qquad (6)$$

where $\gamma$ and $\beta$ are learnable affine parameters of normalization layer, and $\epsilon$ is a small constant added for numerical stability. In our experiments, we chose $\alpha$ as 4 to handle various out-of-distribution scenarios effectively. The parameter $s$ is a hyperparameter that determines the degree of adjustment desired and can be tuned based on specific requirements.

IoBMN utilizes CnDRM's class-domain representative statistics and adjusts them based on the current inferencing batch statistics. This dual-statistic approach allows IoBMN to correct the outdated and skewed distribution of the memory, ensuring alignment with the data distribution at each inference point. By leveraging the statistics of the data used during model update points, IoBMN adapts effectively without significant computational overhead. Additionally, this method mitigates the performance degradation caused by the prolonged intervals between adaptations so that the model remains well-aligned with the evolving data distribution.

Table 1: Classification accuracy (%) and latency per batch (s) measured on a Raspberry Pi 4, comparing with and without SNAP (AR=0.1) on CIFAR100-C (ResNet18) and ImageNet-C (ResNet50). **Bold** numbers indicate the highest accuracy on the sparse setting. Extended results for CIFAR10-C and other ARs (0.01, 0.03, 0.05, 0.3, and 0.5) are in Appendix C.

| Methods | Gau. | Shot | Imp. | Def. | Gla. | Mot. | Zoom | Snow | Fro. | Fog | Brit. | Cont. | Elas. | Pix. | JPEG | Avg. | Δ(Acc.) | Lat. |
|---|---|---|---|---|---|---|---|---|---|---|---|---|---|---|---|---|---|---|
| | | | | | | | CIFAR100-C | | | | | | | | | | | |
| Tent | 46.71 | 48.06 | 40.98 | 65.19 | 44.10 | 62.78 | 63.95 | 55.43 | 55.46 | 59.32 | 67.43 | 63.83 | 53.89 | 59.40 | 49.91 | 55.76 | - | 4.54 |
| + STTA | 43.55 | 44.25 | 37.95 | 62.56 | 41.80 | 59.45 | 62.13 | 53.04 | 51.60 | 56.76 | 64.60 | 61.19 | 51.01 | 56.42 | 46.28 | 52.84 | (-2.92) | 3.34 |
| + SNAP | 46.51 | 47.68 | 39.92 | 65.39 | 44.14 | 63.29 | 64.53 | 55.20 | 55.55 | 59.71 | 68.05 | 64.90 | 53.91 | 59.28 | 49.58 | 55.84 | (+0.08) | 3.67 |
| CoTTA | 42.14 | 42.92 | 37.92 | 55.40 | 41.01 | 55.18 | 55.39 | 49.46 | 50.61 | 50.86 | 61.35 | 47.44 | 48.69 | 54.38 | 48.11 | 49.39 | - | 74.77 |
| + STTA | 28.53 | 29.53 | 26.45 | 42.19 | 30.34 | 44.69 | 41.88 | 34.44 | 33.93 | 39.03 | 45.49 | 31.17 | 37.25 | 36.17 | 36.84 | 35.86 | (-13.53) | 4.94 |
| + SNAP | 41.72 | 42.62 | 37.46 | 58.43 | 41.24 | 57.33 | 57.96 | 50.34 | 51.17 | 52.29 | 63.59 | 51.32 | 49.68 | 54.78 | 47.89 | 50.52 | (+1.13) | 4.95 |
| EATA | 38.42 | 39.96 | 32.64 | 62.35 | 38.73 | 59.93 | 61.07 | 50.50 | 50.79 | 55.30 | 64.38 | 60.63 | 49.66 | 53.63 | 43.02 | 50.74 | - | 4.25 |
| + STTA | 38.41 | 39.03 | 32.29 | 61.07 | 38.45 | 58.21 | 60.62 | 49.59 | 49.19 | 54.23 | 62.88 | 57.39 | 49.00 | 53.01 | 42.05 | 49.70 | (-1.04) | 3.13 |
| + SNAP | 40.62 | 41.53 | 34.31 | 64.08 | 40.29 | 61.32 | 63.04 | 52.00 | 51.77 | 56.85 | 65.98 | 61.96 | 51.05 | 55.67 | 44.80 | 52.35 | (+1.61) | 3.51 |
| SAR | 50.75 | 52.00 | 43.87 | 65.44 | 46.30 | 63.60 | 64.68 | 58.41 | 58.26 | 61.34 | 68.03 | 67.68 | 54.53 | 61.52 | 52.72 | 57.94 | - | 6.68 |
| + STTA | 43.92 | 45.28 | 38.64 | 63.36 | 42.58 | 60.36 | 62.78 | 53.39 | 52.23 | 57.54 | 65.41 | 60.88 | 52.07 | 56.80 | 47.16 | 53.49 | (-4.45) | 2.95 |
| + SNAP | 46.29 | 47.60 | 39.95 | 65.26 | 44.00 | 63.09 | 64.97 | 55.08 | 55.17 | 59.73 | 68.13 | 64.72 | 53.84 | 58.98 | 49.54 | 55.76 | (-2.18) | 3.09 |
| RoTTA | 38.54 | 39.85 | 33.73 | 63.45 | 40.74 | 60.54 | 62.03 | 51.61 | 51.75 | 56.20 | 65.14 | 61.55 | 51.22 | 54.42 | 42.50 | 51.55 | - | 6.71 |
| + STTA | 36.28 | 37.12 | 31.38 | 61.20 | 38.36 | 58.26 | 60.30 | 49.20 | 48.21 | 53.54 | 62.80 | 56.78 | 49.61 | 52.28 | 41.26 | 49.11 | (-2.44) | 2.96 |
| + SNAP | 37.83 | 38.42 | 32.38 | 63.73 | 39.72 | 61.32 | 62.58 | 51.38 | 51.18 | 55.61 | 65.70 | 61.39 | 51.36 | 54.51 | 42.85 | 51.33 | (-0.22) | 2.99 |
| | | | | | | | ImageNet-C | | | | | | | | | | | |
| Tent | 27.03 | 28.98 | 28.64 | 24.66 | 23.63 | 38.70 | 45.77 | 44.82 | 38.06 | 54.59 | 64.61 | 16.84 | 51.64 | 55.54 | 49.38 | 39.53 | - | 38.33 |
| + STTA | 22.00 | 23.51 | 23.07 | 19.38 | 18.86 | 32.15 | 42.29 | 39.70 | 34.33 | 51.62 | 63.70 | 15.79 | 47.74 | 52.35 | 45.54 | 35.47 | (-4.06) | 18.01 |
| + SNAP | 26.21 | 27.85 | 27.50 | 23.62 | 22.73 | 36.01 | 44.11 | 42.19 | 38.15 | 52.95 | 64.57 | 30.23 | 48.56 | 53.71 | 47.09 | 39.03 | (-0.50) | 18.76 |
| CoTTA | 13.12 | 13.98 | 13.94 | 12.44 | 12.18 | 23.74 | 35.22 | 31.78 | 30.26 | 44.40 | 62.40 | 15.13 | 40.42 | 45.26 | 36.53 | 28.72 | - | 300.23 |
| + STTA | 10.97 | 11.92 | 11.98 | 11.45 | 11.38 | 22.39 | 34.96 | 30.88 | 29.89 | 44.09 | 61.96 | 13.08 | 40.20 | 45.27 | 36.71 | 27.81 | (-0.91) | 161.98 |
| + SNAP | 15.13 | 16.03 | 15.91 | 13.86 | 14.02 | 24.90 | 36.51 | 32.56 | 31.81 | 46.02 | 63.60 | 15.69 | 41.94 | 46.78 | 38.03 | 30.19 | (+1.47) | 163.24 |
| EATA | 29.62 | 31.79 | 31.17 | 26.89 | 26.30 | 40.65 | 47.44 | 46.29 | 40.78 | 55.57 | 64.97 | 38.02 | 52.66 | 56.03 | 50.26 | 42.56 | - | 31.98 |
| + STTA | 22.43 | 23.78 | 23.26 | 19.38 | 19.42 | 32.18 | 43.22 | 40.65 | 36.64 | 52.38 | 63.87 | 24.59 | 48.13 | 52.89 | 46.33 | 36.61 | (-5.95) | 16.00 |
| + SNAP | 26.10 | 27.29 | 27.13 | 22.38 | 22.15 | 33.45 | 43.92 | 40.96 | 36.68 | 52.71 | 63.77 | 27.93 | 48.47 | 53.23 | 47.46 | 38.24 | (-4.32) | 17.45 |
| SAR | 29.23 | 31.14 | 29.88 | 29.29 | 27.39 | 39.76 | 44.13 | 45.98 | 29.39 | 55.13 | 63.71 | 17.34 | 52.31 | 56.09 | 49.35 | 39.34 | - | 78.15 |
| + STTA | 26.12 | 27.56 | 26.93 | 22.51 | 23.35 | 36.03 | 44.48 | 43.19 | 37.26 | 53.82 | 64.15 | 19.87 | 50.78 | 54.78 | 48.43 | 38.62 | (-0.72) | 21.39 |
| + SNAP | 30.28 | 31.97 | 31.30 | 26.67 | 26.31 | 39.66 | 46.08 | 45.43 | 40.26 | 54.76 | 64.62 | 36.12 | 51.26 | 55.42 | 49.63 | 41.99 | (+2.65) | 23.99 |
| RoTTA | 20.60 | 22.83 | 19.81 | 10.46 | 10.10 | 21.31 | 31.83 | 39.66 | 32.09 | 46.08 | 62.22 | 20.27 | 42.54 | 47.47 | 40.67 | 31.20 | - | 87.00 |
| + STTA | 14.77 | 15.59 | 15.33 | 13.17 | 13.19 | 23.85 | 35.38 | 32.73 | 30.77 | 45.22 | 63.08 | 15.62 | 41.05 | 46.15 | 37.19 | 29.54 | (-1.66) | 45.98 |
| + SNAP | 15.35 | 16.20 | 16.01 | 13.67 | 13.66 | 24.27 | 35.62 | 33.04 | 31.02 | 45.38 | 62.95 | 15.96 | 41.06 | 46.17 | 37.44 | 29.85 | (-1.36) | 47.47 |

## 5 Experiments

This section outlines our experimental setup and presents the results obtained under various STTA settings. Refer to Appendix A for further details.

**Scenario.** We varied the **Adaptation Rates (AR)** to examine how different update frequencies affect both model accuracy and latency under latency-constrained scenarios. In our setup, AR controls how frequently the model is adapted and also corresponds to the memory sampling rate, as the memory size equals the batch size. The main evaluation was run with diverse AR values: 0.01, 0.03, 0.05, 0.1, 0.3, and 0.5. We report the mean accuracy and standard deviation over three random seeds. Latency was measured on three representative edge devices: Raspberry Pi 4 [38], Zero 2W [39], and Jetson Nano [32].

**Dataset and model.** We used three standard TTA benchmarks: **CIFAR10-C**, **CIFAR100-C** and **ImageNet-C** [10] for main evaluation. These datasets include 15 different types of corruption with five levels of severity, and we used the highest one. We employed **ResNet18** [8] as the backbone network, utilizing models pre-trained on CIFAR10 and CIFAR100 [18]. We also use **ResNet50** [8] and **Vit-Base** [4] pre-trained on ImageNet [3] from the TorchVision [25] library.

**Baselines.** SNAP is designed to integrate with existing TTA algorithms. Therefore, testing existing *TTA algorithms under different ARs* serves as our baseline (implementation details, including hyperparameters, are in Appendix A.1). We selected five SOTA TTA algorithms: (i) **Tent** [48] updates only BN affine parameters, (ii) **CoTTA** [50] updates the entire model parameters using a teacher-student framework, (iii) **EATA** [30], (iv) **SAR** [31], and (v) **RoTTA**[53].

**Overall performance across various adaptation rates.** Table 1 and Appendix C provide a performance comparison of baseline state-of-the-art (SOTA) TTA methods and SNAP integration across adaptation rates from 0.01 to 0.5 on CIFAR10/100-C and ImageNet-C. The results reveal that while STTA reduces adaptation latency by up to 93.38%, conventional SOTA algorithms suffer significant accuracy degradation in STTA settings. In contrast, SNAP effectively mitigates this performance drop. By utilizing minimal updates with only a fraction of the samples, SNAP consistently outperforms baseline methods and achieves accuracy comparable to fully adapted models. These findings highlight SNAP's ability to balance efficiency and performance, preserving or even improving classification accuracy in sparse adaptation scenarios.

Figure 5 further illustrates that SNAP maintains STTA performance even at adaptation rates as low as 0.01 while significantly reducing latency. In contrast, naïve STTA suffers substantial performance degradation as the adaptation rate decreases. Notably, computationally complex and latency-intensive method CoTTA benefits more from SNAP. This is because CoTTA updates all model parameters, making it highly dependent on an effective sampling strategy, underscoring the effectiveness of Cn-DRM. At higher adaptation rates (0.5 or 0.3), SNAP can even surpass fully adapted methods by selectively utilizing the most informative

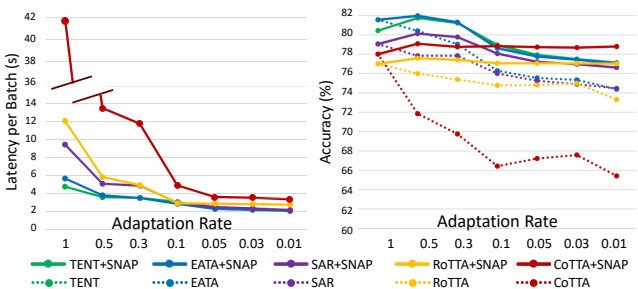

Figure 5: Latency on Raspberry Pi 4 and CIFAR10-C accuracy across adaptation rates. Due to SNAP's negligible overhead, solid and dotted lines overlap in the latency plot. Marker size indicates standard deviation.

samples, similar to existing sampling-based TTA methods [30, 31, 6, 7]. With sufficient samples and update frequency, SNAP's class-domain representative sampling filters harmful data points, further improving performance. Overall, these results confirm that SNAP significantly reduces per-batch latency while preserving accuracy, demonstrating its effectiveness in resource-constrained environments. Extended results on various adaptation rates and datasets (CIFAR10/100-C and ImageNet-C) are reported in Appendix C.

**Contribution of SNAP's individual components.** We conducted an ablative evaluation to understand the effects of the individual components of SNAP (Table 2; more results on various adaptation rates and datasets are in Appendix C.4). CRM denotes prediction-balanced sampling with a confidence threshold, and CnDRM denotes both Class and Domain Representative sampling (the first component of SNAP). For inference, the default uses test batch normalization statistics, EMA uses the exponential moving average of the test batch, and IoBMN uses memory samples' statistics corrected to match that of the test batch (the second component of SNAP).

Contrary to the belief that low-entropy samples benefit TTA [30, 31], LowEntropy performed worse than Rand for STTA, due to limited updates causing poor or slow convergence. CRM, originally for data-efficient supervised learning [2, 51], outperformed Rand but remained inferior to CnDRM due to reliance on uncertain pseudo-labels instead of ground truth. The highest accuracy was achieved with IoBMN, which mainly leverages memory

Table 2: Classification accuracy (%) comparison of ablative settings on the STTA (AR=0.1). Performance averaged over all 15 CIFAR10-C corruptions.

| Methods | Tent | CoTTA | EATA | SAR | RoTTA |
|---|---|---|---|---|---|
| Naïve | 76.81 ±0.18 | 66.42 ±0.12 | 76.29 ±0.11 | 76.01 ±0.07 | 74.78 ±0.15 |
| Random | 77.08 ±0.14 | 65.61 ±0.08 | 76.59 ±0.10 | 76.33 ±0.13 | 75.01 ±0.16 |
| LowEntropy | 75.66 ±0.09 | 63.19 ±0.14 | 74.89 ±0.12 | 74.41 ±0.18 | 72.60 ±0.10 |
| CRM | 77.77 ±0.05 | 65.71 ±0.19 | 77.18 ±0.08 | 74.36 ±0.11 | 75.27 ±0.17 |
| CnDRM | 77.46 ±0.07 | 77.69 ±0.10 | 77.17 ±0.06 | 76.85 ±0.09 | 75.64 ±0.08 |
| CnDRM+EMA | 78.02 ±0.12 | 72.19 ±0.15 | 77.05 ±0.11 | 76.84 ±0.13 | 76.18 ±0.05 |
| CnDRM+IoBMN | **78.95** ±0.09 | **78.83** ±0.06 | **78.61** ±0.13 | **78.06** ±0.07 | **77.07** ±0.10 |

statistics and adapts minimally to each test batch. These indicate that combining CnDRM and IoBMN in SNAP enhances performance in low-latency STTA.

**Performance validation across diverse edge-devices.** SNAP significantly reduces adaptation latency across a range of edge devices. At an adaptation rate of 0.05, latency was reduced by up to 91.3% on Raspberry Pi 4 [38], 86.2% on Jetson Nano [32], and 93.7% on Raspberry Pi Zero 2 W [39]. This consistent trend across varying hardware confirms SNAP's effectiveness in latency-sensitive edge deployments. Complete results across all adaptation rates and devices are provided in Appendix B.6.

**Memory overhead and compatibility with memory-efficient TTA.** SNAP's memory overhead primarily comes from the memory buffer in CnDRM and statistics stored for IoBMN. Empirical results confirm that this overhead is minimal, accounting for only 0.02% to 1.74% of the original memory usage across all algorithms. Additionally, SNAP improves average memory efficiency by reducing backpropagation frequency. Further theoretical and experimental analyses of memory usage are provided in Appendix B.7. SNAP is also compatible with memory-efficient TTA modules like MECTA [12]. Integrating MECTA with Tent + SNAP reduces peak memory usage by 32.08%, showcasing its effectiveness in meeting both latency and memory constraints (Appendix B.8).

**SNAP on vision transformer.** We validate SNAP on ViT-Base [4] by adapting CnDRM and IoBMN to instance-level layer normalization (LN), replacing batch statistics. This confirms that our core strategies, class-domain sample selection and normalization shift mitigation, generalize to LN. SNAP achieves up to 2.9× latency reduction while preserving or improving full adaptation accuracy across all five baselines (Figure 6). Details are in Appendix B.9.

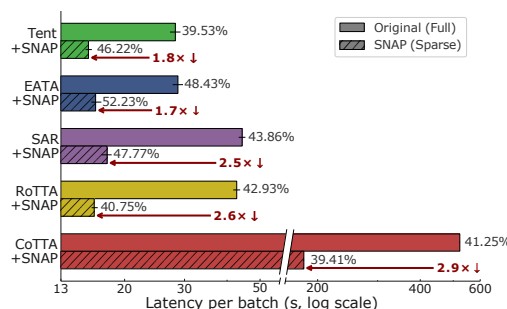

Figure 6: Latency comparison of full (original) and sparse (SNAP) TTA on ViT-Base [4], ImageNet-C. Accuracy values are annotated to the right of each bar.

**Robustness under continuous and persistent distribution shifts.** SNAP adapts efficiently to evolving domains by smoothly moving its domain centroid with minimal overhead. In a continuous stream of corruptions from ImageNet-C, it outperforms naïve STTA by over 2.5% on average. We further assess long-term robustness under temporally correlated and recurring shifts [11]. Combined with SNAP, accuracy remains consistently above 50% over 10 adaptation rounds, whereas naive STTA alone degrades sharply, dropping to 16.97%. These results highlight SNAP's ability to maintain stable performance across both continuous and persistent distribution changes. Details are in Appendices B.13 and B.14.

**Robustness in single-sample (BS=1) adaptation scenario.** In highly constrained environments where the batch size is limited to one, SNAP maintains strong performance. It achieves 51.80% accuracy with an adaptation rate of 0.1, closely matching the full SAR [31] baseline at 52.21% and performing over 5× better than naïve STTA. CnDRM continues to effectively select informative samples, while IoBMN leverages memory-based statistics to adaptively normalize each input under this extreme regime. Further details are provided in Appendix B.15.

**Impact of memory size and learning rate.** SNAP demonstrates robustness to both memory sizes and learning rates. It adapts effectively with a memory size equal to the batch size, as larger sizes offer only 1∼2% marginal gains before saturation. Likewise, it outperforms all baselines across learning rates, showing up to 5∼10% absolute gains under sparse adaptation. These results underscore SNAP's efficiency and stability under constrained adaptation. Full analyses are in Appendices B.16 and B.17.

## 6 Discussion and conclusion

**Limitations and societal impacts.** SNAP uses a fixed adaptation rate, but dynamically adjusting it based on distribution shifts or system load could improve responsiveness. The confidence threshold in CnDRM is also fixed as a simple safeguard, which may limit adaptability. Dynamically tuning this threshold based on data characteristics could further enhance sampling efficiency. In addition, our implementation reduces average latency by adapting sparsely across batches, rather than explicitly optimizing backpropagation delay, due to PyTorch [33] constraints that require backpropagation to run as a single block. Future work could explore distributing the backpropagation step allocation across batches to further enhance applicability. Furthermore, deploying deep learning on edge devices at scale can raise societal concerns, such as carbon emissions [41]. By lowering computational overhead, SNAP helps mitigate these environmental impacts. It also reduces the need to transmit user data to the server, supporting stronger privacy in real-world applications.

**Conclusion.** We highlight the often-overlooked issue of TTA latency, a critical factor for resource-constrained edge devices. To address this, we propose SNAP, a lightweight STTA framework that significantly reduces latency while preserving accuracy. SNAP leverages class-domain representative memory for adaptation and optimizes inference by adapting normalization layers using memory to account for domain shifts. Extensive experiments and ablation studies validate its effectiveness.

## Acknowledgments and Disclosure of Funding

This work was partly supported by the Institute of Information & communications Technology Planning & Evaluation (IITP) grant funded by the Korea government (MSIT) (No.2024-00444862, Non-invasive near-infrared based AI technology for the diagnosis and treatment of brain diseases), the National Research Foundation of Korea (NRF) grant funded by the Korea government (MSIT) (RS-2024-00337007), and the Institute of Information & Communications Technology Planning & Evaluation (IITP) grant funded by the Korea government (MSIT) (RS-2025-25442824, AI Star Fellowship Program (Ulsan National Institute of Science and Technology)). ※ MSIT: Ministry of Science and ICT

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

# A Experiment details

All experiments presented in this paper were conducted using three random seeds (0, 1, 2), and we report the average accuracies along with their corresponding standard deviations. To ensure efficiency in experimentation, accuracy measurements were obtained using NVIDIA GeForce RTX 3090 GPUs, as the performance differences attributable to the random seed are negligible. Latency measurements were mainly conducted on a Raspberry Pi 4 [38], equipped with a Quad-core Cortex-A72 (ARM v8) 64-bit SoC @ 1.8GHz CPU and 4GB RAM. In addition, two more edge-devices: NVIDIA Jetson Nano [32] and Raspberry Zero 2 W [39] are also utilized for latency measurements.

## A.1 Baseline implementation details

In this study, we utilized the official implementations of the baseline methods. To ensure consistency, we adopted the reported best hyperparameters documented in the respective papers or source code repositories as much as possible. Also, we present information about the implementation specifics of the baseline methods and provide a comprehensive overview of our experimental setup, including detailed descriptions of the employed hyperparameters.

We adopt hyperparameters from the original papers or the official code of the baselines for consistency. To assess the generality of SNAP, the test batch sizes were set to 16 for all baseline methods to ensure a fair comparison. To minimize overhead and maintain consistency with inference batches, we set the size of CnDRM equal to the batch size. TTA is conducted in an online manner, with adaptation or inference performed per batch. When there was a conflict between the implementation of SNAP and certain components of the existing baseline methods, we prioritized SNAP's features for fair evaluation at the STTA setting.

**Tent [48]** We update the BN affine parameters using the SGD optimizer [24] with a learning rate of $l = 1e - 3$ for CIFAR10/100C and $l = 1e - 4$ for ImageNet-C. For separate experimentation on the ViT, we used a learning rate of $l = 2e - 4$.

**CoTTA [50]** We update all model parameters using the Adam optimizer [15] with a learning rate of $l = 1e - 4$. Furthermore, we set CoTTA's teacher model EMA factor to $\alpha = 0.99$, the restoration factor to $p = 0.1$, and the anchor probability to $p_{\text{th}} = 0.9$.

**EATA [30]** We use the SGD optimizer with a learning rate of $l = 1e - 4$. We set the entropy threshold as $E_0 = 0.4 \times \ln |N|$, where $N$ is the total number of classes.

**SAR [31]** We use SAM [5] with the base optimizer as SGD with a learning rate of $l = 1e - 3$. For fair evaluation, we replaced the sample filtering scheme with SNAP's CnDRM.

**RoTTA [53]** We use the SGD optimizer with a learning rate of $l = 1e - 3$. For fair evaluation, we replaced RoTTA's RBN and CSTU with SNAP's CnDRM and IoBMN. For the teacher-student structure, we set the teacher model's exponential moving average update rate as $v = 1e - 3$.

Finally, we list the hyperparameters specific to the components of SNAP. The confidence threshold for CnDRM $\tau_{conf}$ is set to 0.4 for CIFAR10-C, 0.45 for CIFAR100-C, and 0.5 for ImageNet-C. The entropy threshold for our ablation study $\tau_{entr}$ is set to $\log(10) \times 0.40$ for CIFAR10-C and $\log(100) \times 0.40$ for CIFAR100-C, as referenced in a previous work using entropy-based filtering [30]. Additionally, the parameters for the soft shrinkage function in IoBMN are fixed with $\alpha = 4$ for Tent, CoTTA, SAR, RoTTA, and $\alpha = 2$ for EATA.

# B Additional discussions

## B.1 Theoretical analysis on class and domain representative sampling criteria

The sampling strategy in SNAP's CnDRM module is grounded in theoretical insights from data-efficient learning and generalization under constrained adaptation settings. Under latency-constrained scenarios, models can only adapt intermittently. This raises the following question:

*Which subset of streaming samples yields the greatest adaptation gain when only a fraction $\rho = n/N$ of them can be used for weight updates?*

**Selection ratio and phase transition.** Let $\rho = n/N \in (0, 1]$ denote the adaptation ratio, where $n$ is the number of selected samples for model update out of $N$ total seen test samples. Theoretical and empirical studies [2, 44, 17] show that the optimal selection strategy varies significantly depending on the value of $\rho$:

- When $\rho > 0.6$ (high adaptation rate), selecting low-confidence samples near decision boundaries provides maximal information gain.

- When $\rho \leq 0.5$ (sparse adaptation), selecting high-confidence, representative samples near class or domain centroids leads to better generalization.

This dichotomy is supported analytically in [2], which shows that in underparameterized regimes, boundary samples may inject noisy gradients and destabilize learning.

**Illustrative example.** Consider a binary classification task where inputs $x_i \sim \mathcal{N}(0, \frac{1}{\sqrt{d}} I_d)$, and labels are assigned by $y_i = \text{sign}(x_{i1})$. The Bayes-optimal classifier aligns with $e_1$, the first basis vector. Suppose we select a subset $(X_S, y_S) \in \mathbb{R}^{d \times n} \times \{-1, 1\}^n$ and compute the ridge regression solution:

$$w_S = \arg \min_{w \in \mathbb{R}^d} \|y_S - X_S^\top w\|^2. \tag{7}$$

As shown in prior work [2], in the *sample-deficient regime* ($n \ll d$), the solution $w_S$ aligns best with the true decision direction when trained on high-confidence samples. In contrast, in the *sample-sufficient regime* ($n \gg d$), boundary samples become more beneficial for refining decision boundaries.

**Sampling criterion under sparse adaptation.** Based on this, the optimal update subset $\mathcal{S}^* \subset \mathcal{D}_{test}$ under $\rho \ll 1$ can be defined as:

$$\mathcal{S}^* = \arg \max_{\mathcal{S}} \sum_{x_i \in \mathcal{S}} \|f(x_i) - \mu_{c_i}\|_2^2 \quad \text{s.t.} \quad c_i = \arg \max_j f_j(x_i), \tag{8}$$

where $\mu_{c_i}$ is the estimated feature-space class centroid. This motivates our use of class- and domain-representative memory (CnDRM), which prioritizes confident, centroid-aligned samples for parameter updates under low-frequency adaptation.

Under sparse adaptation ($\rho \ll 1$), selecting the most informative subset of test samples becomes critical. Our method prioritizes samples that are both semantically reliable (class-representative) and statistically aligned (domain-representative).

For the class-representative criterion, we follow insights from [2], which show that in the sample-deficient regime, high-confidence samples, those far from the decision boundary, yield better generalization than boundary samples. Rather than explicitly computing class centroids, we approximate class-representative samples by selecting those with the highest prediction confidence:

$$\mathcal{S}^*_{\text{class}} = \{x_i \in \mathcal{D}_{test} \mid \text{Conf}(x_i) \geq \tau\}, \tag{9}$$

where $\text{Conf}(x_i) = \max_j f_j(x_i)$ is the softmax confidence score and $\tau$ is a confidence threshold.

To additionally enforce domain-level representativeness, we compute the Wasserstein-2 distance between each candidate sample and the estimated domain distribution. Specifically, each domain $d$ is modeled as a Gaussian $\mathcal{N}(\nu_d, \Sigma_d)$, where:

$$\nu_d = \frac{1}{|\mathcal{D}_d|} \sum_{x_i \in \mathcal{D}_d} f(x_i), \qquad \Sigma_d = \frac{1}{|\mathcal{D}_d|} \sum_{x_i \in \mathcal{D}_d} (f(x_i) - \nu_d)(f(x_i) - \nu_d)^\top. \tag{10}$$

Each sample $x_i$ is treated as an empirical distribution $\mathcal{N}(\mu_i, \Sigma_i)$ using a local batch of neighboring features. The closed-form squared 2-Wasserstein distance between two Gaussians is given by:

$$W_2^2 \left(\mathcal{N}(\mu_i, \Sigma_i), \mathcal{N}(\nu_d, \Sigma_d)\right) = \|\mu_i - \nu_d\|_2^2 + \text{Tr} \left(\Sigma_i + \Sigma_d - 2(\Sigma_d^{1/2} \Sigma_i \Sigma_d^{1/2})^{1/2}\right). \tag{11}$$

Combining both criteria, our final selection strategy becomes:

$$\mathcal{S}^* = \arg \min_{\mathcal{S} \subset \mathcal{S}^*_{\text{class}}} \sum_{x_i \in \mathcal{S}} W_2^2(\mathcal{N}(\mu_i, \Sigma_i), \mathcal{N}(\nu_d, \Sigma_d)), \tag{12}$$

i.e., we select a subset of high-confidence samples that are also distributionally aligned with the estimated domain centroid. This ensures that parameter updates occur on samples that are both semantically stable and statistically representative under test-time domain shift.

**Practical implementation.** In our system, we operate with $\rho < 0.5$ (e.g., update every two test batches). While this reduces update opportunities, the degradation in performance is mitigated via CnDRM's informed sample selection. In addition, SNAP integrates IoBMN (Inference-only Batch-aware Memory Normalization), which updates batch norm statistics from all incoming samples, even if they are not used for weight updates. This reduces covariate shift and maintains normalization stability across domains.

## B.2 Assumptions and derivation of the Wasserstein formulation

The closed-form expression in Equation 1 assumes that feature distributions are Gaussian with diagonal covariance. This section clarifies the underlying assumptions and derivation.

**Distribution definition.** The distribution in Equation 1 refers to the empirical distribution of scalar *feature activations per channel*, rather than the input data distribution. We consider each channel's activation statistics (mean and variance) in a deep layer's feature map.

**Wasserstein approximation and assumptions.** To compute the Wasserstein distance efficiently, the feature distributions are approximated as *univariate Gaussian distributions* for each channel. These distributions are assumed to be independent, corresponding to a *diagonal covariance* assumption. The detailed assumptions are as follows:

- Gaussian assumption: Each channel's activations are assumed to follow a Gaussian distribution $\mathcal{N}(\mu, \sigma^2)$, which simplifies the formulation since Gaussian distributions are fully determined by their mean $\mu$ and variance $\sigma^2$. This approximation is commonly adopted in deep learning, where normalized feature activations tend to exhibit near-Gaussian behavior in high-dimensional feature spaces.

- Diagonal covariance: The covariance matrix for each Gaussian is assumed to be diagonal, implying independence across channels. This assumption is widely used in domain adaptation and transport-based methods, as it reduces the computational complexity of full covariance estimation while focusing on per-channel variance shifts.

- 2-Wasserstein distance for Gaussian distributions: Under these assumptions (independent Gaussian distributions), the squared 2-Wasserstein distance between two univariate Gaussian distributions $\mathcal{N}(\mu_1, \sigma_1^2)$ and $\mathcal{N}(\mu_2, \sigma_2^2)$ is given by:

$$W_2^2 = (\mu_1 - \mu_2)^2 + (\sigma_1 - \sigma_2)^2.$$

This closed-form expression enables efficient computation of the Wasserstein distance without estimating full covariance matrices, which is computationally expensive.

Such approximations are widely adopted in the domain adaptation literature [20, 27], as they balance computational efficiency and empirical performance while providing a meaningful measure of domain-level similarity.

## B.3 Comparison with prior TTAs under sparse update constraints

While prior TTA studies have partially explored using memory banks and the correction of Batch Normalization (BN) statistics at inference time, our key contribution lies in systematically redesigning these components for sparse-update regimes in resource-constrained environments, which impose fundamentally different computational and statistical constraints.

**CnDRM vs. Prior memory-based sampling (RoTTA [53], NOTE [6], SAR [31], and EATA [30])**
Previous methods assume frequent adaptation, updating every batch and using more than 50% of test samples. They typically select temporally balanced or low-entropy samples for updates. In contrast, our goal is *Sparse TTA* (e.g., 10% adaptation rate), where such filtering leaves too few useful samples for effective adaptation. To address this, CnDRM avoids low-entropy filtering and instead selects *representative samples* based on domain centroids and class confidence. This approach enables efficient adaptation with minimal latency. Theoretical analysis (Appendix B.1) and ablation results (Table 2, Appendix C.4) show that prior entropy-based filtering performs worse than even random selection under sparse-update settings.

**IoBMN vs. Instance-wise BN correction (NOTE [6])**  NOTE corrects BN statistics per instance, which introduces latency proportional to the number of test samples. In contrast, IoBMN leverages domain-class statistics from CnDRM's memory to correct batch BN statistics efficiently while remaining compatible with batch inference. The memory statistics are adaptively shifted toward batch statistics to mitigate skew from sparse updates. This design enhances computational efficiency and normalizes using adaptation-involved samples, yielding consistent performance gains (Table 2, Appendix C.4).

## B.4    Domain influence in early layer representations

In deep learning models, early layers capture low-level features such as textures, edges, and frequency components [54]. These features are inherently domain-specific, making these layers more sensitive to shifts in input data distribution—a critical challenge for tasks requiring domain adaptation and generalization [19, 42]. This sensitivity arises because early layers encapsulate domain-specific patterns that may not generalize to new distributions. Under the covariate shift assumption [37], while input distributions differ between source and target domains, the conditional distribution of labels remains the same. This discrepancy between input distributions makes early layers particularly vulnerable to domain shifts.

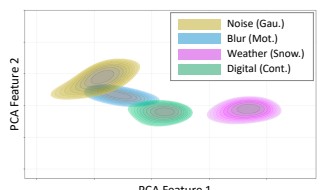

Figure 7: PCA embedding of early layer features for one domain from each of the four main CIFAR10-C corruption categories, showing clear separation between domains.

Visualizing early layer feature embeddings using 2D PCA on CIFAR-10C domains reveals distinct domain-specific patterns, highlighting the significant influence of domain information in these representations (Figure 7). Our preliminary experiments further confirm that sparse TTA, using the Wasserstein distance between moving batch normalization statistics and instance-specific statistics derived from early layer hidden features, can significantly improve performance. Selecting instances closer to the target domain distribution center using this distance metric yields better adaptation results, as demonstrated by performance comparisons between the top 20% and bottom 20% of samples (Figure 4). These findings emphasize the crucial role of domain-sensitive early layers in achieving effective adaptation.

## B.5    Analysis on confidence threshold on pseudo-label accuracy

We analyzed the impact of using a confidence threshold for pseudo-label selection by comparing random sampling with high-confidence sampling across three benchmarks: CIFAR10-C, CIFAR100-C, and ImageNet-C. Table 3 shows that high-confidence sampling consistently outperformed random sampling, achieving significantly higher pseudo-label accuracy in all datasets. This result demonstrates the effectiveness of selecting high-confidence samples to improve the quality of pseudo-labels, thereby enhancing model adaptation under domain shift conditions.

Table 3: Pseudo-label accuracy comparison between random and high-confidence sampling on three benchmakrs: CIFAR10-C, CIFAR100-C, and ImageNet-C. **Bold** numbers are the highest accuracy.

|  | CIFAR10-C | CIFAR100-C | ImageNet-C |
|---|---|---|---|
| Random | $69.91_{\pm 0.31}$ | $45.30_{\pm 0.20}$ | $23.90_{\pm 0.19}$ |
| **HighConf** | $\mathbf{74.80}_{\pm 0.15}$ | $\mathbf{59.38}_{\pm 0.26}$ | $\mathbf{59.40}_{\pm 0.04}$ |

## B.6 Latency tracking of SNAP on diverse edge-devices

To evaluate the latency efficiency of SNAP on resource-constrained edge devices, we measured the adaptation latency across three devices: NVIDIA Jetson Nano [32], Raspberry Pi 4 [38], and Raspberry Pi Zero 2 W [39]. These experiments compared the latency of SNAP with the Original TTA framework, specifically focusing on five state-of-the-art TTA algorithms: Tent [48], EATA [30], SAR [31], RoTTA [53], and CoTTA [50]. The experiments were conducted at an adaptation rate of 0.1, demonstrating the effectiveness of SNAP in reducing adaptation latency while maintaining competitive accuracy. Figure 8 illustrates the latency performance for each device. It is evident

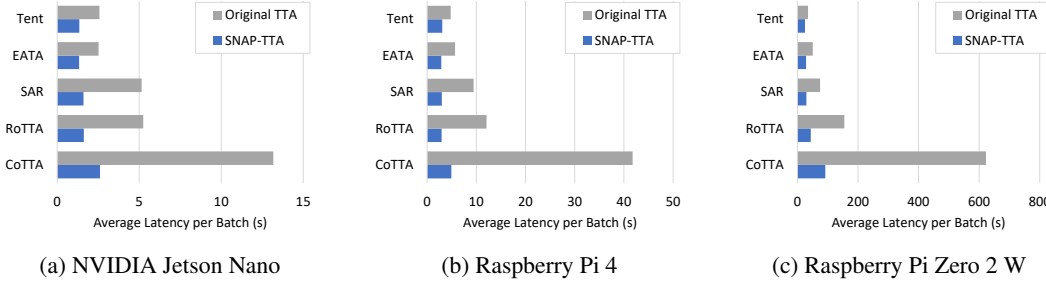

|  (a) NVIDIA Jetson Nano | (b) Raspberry Pi 4 | (c) Raspberry Pi Zero 2 W |

Figure 8: Latency comparison between SNAP-TTA and Original TTA across five state-of-the-art TTA algorithms (Tent, EATA, SAR, RoTTA, CoTTA) on three edge devices: (a) NVIDIA Jetson Nano, (b) Raspberry Pi 4, and (c) Raspberry Pi Zero 2 W. SNAP-TTA demonstrates significant latency reductions while maintaining competitive adaptation performance. The experiments were conducted at an adaptation rate of 0.1.

that SNAP achieves a significant reduction in adaptation latency compared to the Original TTA framework. Notably, the latency reduction was proportional to the adaptation rate, validating the efficiency of SNAP in sparse adaptation scenarios. For instance, the latency for CoTTA was reduced by up to 87.5% on the Raspberry Pi 4, emphasizing the practical benefits of SNAP in latency-sensitive environments. Additionally, similar trends were observed across other devices, including the resource-limited Raspberry Pi Zero 2 W. Since SNAP is hardware-agnostic, accuracy was not measured separately for each device, and no accuracy differences are expected. The results confirm that SNAP not only ensures substantial latency reductions but also adapts effectively to real-world conditions on diverse edge devices, proving its suitability for deployment in latency-sensitive applications.

Table 4: Latency Measurements (AR=1, 0.3, 0.1, 0.05) on Jetson Nano [32].

| Methods | AR=1 | AR=0.3 | AR=0.1 | AR=0.05 |
|---|---|---|---|---|
| Tent | 2.57 | 1.97 (-23.51%) | 1.35 (-47.62%) | 1.19 (-53.75%) |
| EATA | 2.52 | 1.90 (-24.70%) | 1.33 (-47.22%) | 1.19 (-52.79%) |
| SAR | 5.15 | 2.87 (-44.29%) | 1.60 (-68.94%) | 1.32 (-74.28%) |
| RoTTA | 5.24 | 2.91 (-44.46%) | 1.62 (-69.13%) | 1.32 (-74.81%) |
| CoTTA | 13.18 | 6.13 (-53.46%) | 2.61 (-80.22%) | 1.82 (-86.19%) |

Table 5: Latency Measurements (AR=1, 0.3, 0.1, 0.05) on Raspberry Pi 4 [38].

| Methods | AR=1 | AR=0.3 | AR=0.1 | AR=0.05 |
|---|---|---|---|---|
| Tent | 4.78 | 3.54 (-26.09%) | 3.09 (-35.45%) | 2.35 (-50.87%) |
| EATA | 5.68 | 3.52 (-38.00%) | 2.87 (-49.45%) | 2.31 (-59.28%) |
| SAR | 9.45 | 4.88 (-48.34%) | 2.98 (-68.41%) | 2.54 (-73.16%) |
| RoTTA | 12.07 | 4.95 (-58.97%) | 2.94 (-75.62%) | 2.91 (-75.91%) |
| CoTTA | 41.77 | 11.80 (-71.76%) | 4.93 (-88.19%) | 3.64 (-91.29%) |

Table 6: Latency Measurements (AR=1, 0.3, 0.1, 0.05) on Raspberry Pi Zero 2 W [39].

| Methods | AR=1 | AR=0.3 | AR=0.1 | AR=0.05 |
|---|---|---|---|---|
| Tent | 34.96 | 24.67 (-29.42%) | 25.06 (-28.32%) | 17.07 (-51.16%) |
| EATA | 50.72 | 27.01 (-46.75%) | 28.43 (-43.93%) | 17.00 (-66.48%) |
| SAR | 74.64 | 47.79 (-35.96%) | 29.56 (-60.40%) | 18.64 (-75.02%) |
| RoTTA | 154.88 | 86.54 (-44.13%) | 44.08 (-71.54%) | 22.44 (-85.51%) |
| CoTTA | 622.28 | 228.03 (-63.36%) | 92.01 (-85.21%) | 39.22 (-93.70%) |

## B.7 Memory overhead of SNAP

The SNAP framework achieves substantial latency reduction and accuracy improvements with minimal memory overhead, even under resource-constrained scenarios like edge devices. In this section, we present both a theoretical analysis of the memory requirements and empirical results obtained from evaluations on a Raspberry Pi 4[38] (CPU-only edge device).

The memory overhead of SNAP arises from two main components: (1) the memory buffer in Class and Domain Representative Memory (CnDRM) for storing representative samples, including both feature statistics (mean and variance) and the raw image samples, and (2) the statistics required for Inference-only Batch-aware Memory Normalization (IoBMN). For a batch size $B$, the total theoretical memory overhead can be expressed as: Memory Overhead $= B \times$ (Image Size $+ 2 \times$ Feature Dimension $\times$ Bytes per Value)$+$Feature Dimension$\times$Bytes per Value$\times$ 2. The last term accounts for the storage of IoBMN statistics (mean and variance for each feature channel). The image size is calculated based on the dataset resolution and data type.

For ResNet18 on CIFAR10-C, CIFAR10 images have a resolution of $32 \times 32 \times 3$ with each value stored as 1 byte. For a feature dimension of 512 and batch size $B = 16$, the total overhead is: Image Overhead $= 16 \times (32 \times 32 \times 3 \times 1) = 49,152$ bytes (48 KB), Feature Overhead (CnDRM) $= 16 \times (512 \times 2 \times 4) = 65,536$ bytes (64 KB), Feature Overhead (IoBMN) $= 512 \times 2 \times 4 = 4,096$ bytes (4 KB). Thus, the total memory overhead is: Total Overhead $= 48$ KB $+ 64$ KB $+ 4$ KB $= 116$ KB.

For ResNet50 on ImageNet-C, ImageNet images have a resolution of $224 \times 224 \times 3$, stored as 1 byte per value. For a feature dimension of 2048 and batch size $B = 16$, the total overhead is: Image Overhead $= 16 \times (224 \times 224 \times 3 \times 1) = 12,044,928$ bytes (11.5 MB), Feature Overhead (CnDRM) $= 16 \times (2048 \times 2 \times 4) = 262,144$ bytes (256 KB), Feature Overhead (IoBMN) $= 2048 \times 2 \times 4 = 16,384$ bytes (16 KB). Thus, the total memory overhead is: Total Overhead $= 11.5$ MB $+ 256$ KB $+ 16$ KB $\approx 11.77$ MB.

Table 7 shows the empirical memory usage of SNAP compared to Original TTA methods (Tent, EATA, CoTTA, SAR, and RoTTA). The results were averaged across three seeds of experiments and represent the memory footprint observed in a CPU-only edge device, Raspberry Pi 4. While minor variations in measurements are expected due to the nature of CPU memory footprint tracking, the results robustly indicate that the actual memory overhead of SNAP on edge devices is extremely low across all algorithms, ranging from 0.02% to 1.74%. Furthermore, while peak memory usage is either slightly increased or remains comparable to Original TTA methods, the average memory usage of SNAP is consistently lower. This is because SNAP performs backpropagation infrequently, which is the most memory-intensive operation in TTA.

Table 7: Comparison of memory usage (Average Memory, Peak Memory, and Memory Overhead) between Original TTA and SNAP (adaptation rate 0.3) across various methods (Tent, EATA, CoTTA, SAR, and RoTTA) tested on Raspberry Pi 4. **Bold** numbers are the lowest memory usage.

| Methods | Average Mem (MB) | | Peak Mem (MB) | | Mem Overhead (MB) |
|---|---|---|---|---|---|
| | Original TTA | SNAP | Original TTA | SNAP | SNAP - Original |
| Tent | 764.24 | **751.35** | **822.93** | 828.46 | 5.52 (0.67%) |
| CoTTA | 1133.52 | **1099.64** | **1211.21** | 1227.99 | 16.78 (1.13%) |
| EATA | 816.69 | **749.95** | **847.73** | 862.51 | 14.78 (1.74%) |
| SAR | 786.65 | **753.69** | **863.77** | 865.18 | 1.41 (0.02%) |
| RoTTA | 933.23 | **871.64** | **972.23** | 983.94 | 11.71 (1.20%) |

These findings demonstrate that **SNAP's memory overhead is negligible compared to its benefits in latency reduction and accuracy improvements**. By leveraging a small memory buffer for representative samples and minimizing backpropagation operations, SNAP not only achieves a lightweight memory profile but also becomes more efficient in terms of average memory usage compared to Original TTA. This lightweight design, combined with its advantages in latency and accuracy, underscores the practicality of SNAP for deployment in latency-sensitive applications on edge devices.

## B.8 Integration of SNAP with memory-efficient TTA algorithm

This section evaluates the integration of SNAP with MECTA [12], a memory-efficient TTA algorithm, to demonstrate its applicability for resource-constrained edge devices. The experimental setup follows the evaluation settings presented in the MECTA paper to ensure a fair and consistent comparison. Specifically, we analyze the performance of Tent and EATA, enhanced with MECTA and further integrated with SNAP, using the ResNet50 model with a batch size of 64 on the ImageNet-C dataset.

Table 8 presents the classification accuracy and peak memory usage for Tent+MECTA and EATA+MECTA configurations with and without SNAP. Integrating SNAP with Tent+MECTA improves accuracy from 35.21% to 39.52%, while reducing peak memory usage by approximately 30% compared to the Tent baseline. Similarly, SNAP boosts the accuracy of EATA+MECTA from 35.55% to 42.86% while maintaining an efficient memory footprint.

Table 8: Comparison of classification (%) and memory peak (MB) in STTA with an adaptation rate of 0.1. MECTA significantly reduces memory consumption, and SNAP is applied alongside it to boost the performance of sparse adaptation. The accuracy is the average over 15 corruptions in ImageNet-C. **Bold** numbers indicate either the lowest memory usage or the highest accuracy.

| Methods | Accuracy (%) | Max Memory (MB) |
|---|---|---|
| Tent | $35.21_{\pm 0.09}$ | 6805.26 |
| +MECTA | $37.62_{\pm 0.16}$ | **4620.25 (-32.10%)** |
| + SNAP | $\mathbf{39.52}_{\pm 0.13}$ | 4622.12 (-32.08%) |
| EATA | $35.55_{\pm 0.19}$ | 6541.02 |
| +MECTA | $41.41_{\pm 0.37}$ | **4512.38 (-31.01%)** |
| + SNAP | $\mathbf{42.86}_{\pm 0.20}$ | 4535.44 (-30.66%) |

Further details are provided in Table 9, which evaluates the combination of SNAP with MECTA across various corruption types and adaptation rates (AR = 0.3, 0.1, and 0.05). These results show that SNAP consistently outperforms baseline configurations across all adaptation rates and corruption types. This demonstrates the robustness of SNAP when integrated with MECTA and its suitability for real-world applications.

By adhering to the evaluation settings of the MECTA paper, this study ensures high reliability and comparability of results. The findings confirm that SNAP is highly compatible with MECTA, significantly improving both accuracy and memory efficiency. This synergy highlights the potential of combining SNAP and MECTA for deployment in resource-constrained environments such as edge devices.

Table 9: Evaluation of SNAP with MECTA on ImageNet-C through Adaptation Rates(AR) (0.3, 0.1, and 0.05). **Bold** numbers are the highest accuracy.

| AR | Methods | Gau. | Shot | Imp. | Def. | Gla. | Mot. | Zoom | Snow | Fro. | Fog | Brit. | Cont. | Elas. | Pix. | JPEG | Avg. |
|---|---|---|---|---|---|---|---|---|---|---|---|---|---|---|---|---|---|
| 0.3 | Tent + MECTA | 28.20 ±0.30 | 30.13 ±0.41 | 29.58 ±0.08 | 23.07 ±0.22 | 23.35 ±0.47 | 34.49 ±0.13 | 45.95 ±0.13 | 40.97 ±0.15 | 35.68 ±0.41 | 55.66 ±0.04 | 66.56 ±0.06 | 14.72 ±0.47 | 53.09 ±0.18 | 57.16 ±0.05 | 50.74 ±0.15 | 39.29 ±0.22 |
| | + SNAP | **30.49** ±0.26 | **31.98** ±0.14 | **31.66** ±0.21 | **26.29** ±0.32 | **26.19** ±0.02 | **38.47** ±0.30 | **47.38** ±0.11 | **43.79** ±0.11 | **40.12** ±0.12 | **56.38** ±0.05 | **66.81** ±0.07 | **28.87** ±0.28 | **53.53** ±0.09 | **57.61** ±0.10 | **50.86** ±0.08 | **42.03** ±0.15 |
| | EATA + MECTA | 32.18 ±0.60 | 34.85 ±0.49 | 33.06 ±0.31 | 28.80 ±0.22 | 29.18 ±0.18 | 41.02 ±0.26 | 49.24 ±0.08 | 47.10 ±0.20 | 41.56 ±0.25 | 57.35 ±0.12 | 66.27 ±0.05 | 34.56 ±0.12 | 55.38 ±0.10 | 58.19 ±0.04 | **52.87** ±0.26 | 44.11 ±0.22 |
| | + SNAP | **33.67** ±0.19 | **35.76** ±0.24 | **34.86** ±0.10 | **30.35** ±0.11 | **30.29** ±0.04 | **42.78** ±0.06 | **49.55** ±0.10 | **47.46** ±0.10 | **42.32** ±0.05 | **57.50** ±0.15 | 66.18 ±0.06 | **39.08** ±0.81 | 55.38 ±0.16 | **58.35** ±0.12 | 52.72 ±0.02 | **45.08** ±0.15 |
| 0.1 | Tent + MECTA | 24.94 ±0.15 | 26.73 ±0.20 | 25.63 ±0.07 | 21.11 ±0.22 | 21.46 ±0.18 | 32.11 ±0.02 | 44.05 ±0.19 | 38.22 ±0.27 | 36.36 ±0.09 | **53.92** ±0.12 | 66.48 ±0.02 | 18.50 ±0.45 | 50.80 ±0.12 | **55.67** ±0.18 | 48.33 ±0.11 | 37.62 ±0.16 |
| | + SNAP | **27.49** ±0.08 | **28.90** ±0.14 | **28.26** ±0.16 | **23.49** ±0.17 | **23.76** ±0.12 | **34.92** ±0.06 | **45.18** ±0.13 | **40.21** ±0.09 | **38.40** ±0.18 | 53.78 ±0.14 | **66.54** ±0.03 | **27.72** ±0.20 | **51.00** ±0.20 | 55.48 ±0.13 | **47.61** ±0.17 | **39.52** ±0.13 |
| | EATA + MECTA | 29.42 ±0.67 | 31.72 ±0.30 | 29.44 ±0.32 | 24.41 ±0.74 | 25.48 ±0.45 | 37.04 ±0.18 | 47.10 ±0.15 | 43.60 ±0.19 | 39.43 ±0.38 | 55.95 ±0.13 | 66.42 ±0.14 | 28.85 ±1.18 | **53.70** ±0.15 | 57.34 ±0.15 | **51.20** ±0.36 | 41.41 ±0.37 |
| | + SNAP | **31.26** ±0.11 | **32.71** ±0.17 | **32.22** ±0.17 | **27.31** ±0.46 | **27.61** ±0.28 | **38.88** ±0.28 | **47.83** ±0.09 | **44.52** ±0.14 | **40.58** ±0.05 | **56.42** ±0.06 | 66.24 ±0.21 | **35.38** ±0.63 | 53.67 ±0.17 | **57.39** ±0.13 | 50.83 ±0.12 | **42.86** ±0.20 |
| 0.05 | Tent + MECTA | 21.22 ±0.13 | 23.19 ±0.22 | 21.90 ±0.13 | 18.69 ±0.18 | 19.39 ±0.20 | 29.89 ±0.13 | 42.02 ±0.10 | 36.53 ±0.22 | 35.23 ±0.05 | **51.75** ±0.15 | **66.23** ±0.04 | 19.64 ±0.27 | 48.43 ±0.03 | **53.54** ±0.13 | 45.43 ±0.11 | 35.54 ±0.14 |
| | + SNAP | **23.93** ±0.27 | **25.37** ±0.22 | **24.10** ±0.15 | **20.42** ±0.18 | **21.14** ±0.07 | **31.83** ±0.06 | **42.68** ±0.04 | **37.53** ±0.16 | **36.31** ±0.20 | 51.42 ±0.17 | 66.19 ±0.04 | **23.84** ±0.24 | **48.62** ±0.05 | 53.20 ±0.17 | 44.57 ±0.17 | **36.74** ±0.15 |
| | EATA + MECTA | 24.97 ±0.42 | 26.95 ±0.27 | 21.87 ±3.29 | 21.19 ±0.90 | 21.94 ±0.45 | 33.61 ±0.08 | 45.11 ±0.11 | 40.92 ±0.11 | 37.73 ±0.19 | 54.64 ±0.10 | 66.60 ±0.07 | 23.03 ±0.59 | 51.87 ±0.35 | **56.60** ±0.25 | 49.15 ±0.23 | 38.41 ±0.51 |
| | + SNAP | **28.39** ±0.57 | **30.10** ±0.38 | **29.45** ±0.22 | **24.32** ±0.20 | **25.12** ±0.07 | **35.54** ±0.20 | **46.04** ±0.27 | **41.87** ±0.07 | **39.16** ±0.15 | **55.12** ±0.01 | **66.61** ±0.09 | **30.34** ±0.34 | **52.06** ±0.24 | 56.42 ±0.11 | 49.11 ±0.07 | **40.64** ±0.20 |

Table 10: Classification accuracy (%) on ImageNet-C through SNAP (AR=0.1) using ViT-Base [4].

| Method | Gau. | Shot | Imp. | Def. | Gla. | Mot. | Zoom | Snow | Fro. | Fog | Brit. | Cont. | Elas. | Pix. | JPEG | Avg. |
|---|---|---|---|---|---|---|---|---|---|---|---|---|---|---|---|---|
| Tent | 40.56 ±0.11 | 41.30 ±0.28 | 41.69 ±0.22 | 35.76 ±0.15 | 31.81 ±0.07 | 42.01 ±0.26 | 38.02 ±0.13 | 44.33 ±0.18 | **53.53** ±0.06 | 20.69 ±0.25 | 72.41 ±0.14 | 30.42 ±0.21 | 45.87 ±0.17 | **51.95** ±0.12 | **56.11** ±0.29 | 43.10 ±0.19 |
| + SNAP | **40.98** ±0.08 | **41.72** ±0.24 | **42.18** ±0.19 | **37.16** ±0.06 | **32.30** ±0.27 | **42.89** ±0.15 | **38.44** ±0.13 | **46.19** ±0.23 | 52.50 ±0.29 | **53.11** ±0.18 | 72.25 ±0.12 | **39.25** ±0.26 | 46.77 ±0.14 | 51.53 ±0.11 | 55.99 ±0.22 | **46.22** ±0.09 |
| CoTTA | 20.05 ±0.16 | 17.12 ±0.20 | 20.43 ±0.21 | 20.06 ±0.14 | 16.62 ±0.25 | 12.87 ±0.13 | 14.50 ±0.12 | 9.68 ±0.23 | 28.31 ±0.27 | 16.01 ±0.23 | 35.79 ±0.27 | 1.96 ±0.19 | 15.60 ±0.20 | 12.09 ±0.29 | 15.99 ±0.17 | 17.14 ±0.18 |
| + SNAP | **34.85** ±0.13 | **33.35** ±0.22 | **36.21** ±0.28 | **31.54** ±0.18 | **25.77** ±0.06 | **35.57** ±0.24 | **32.96** ±0.11 | **42.23** ±0.25 | **55.10** ±0.07 | **51.63** ±0.27 | **71.72** ±0.16 | **5.86** ±0.29 | **42.18** ±0.08 | **39.96** ±0.19 | **52.27** ±0.21 | **39.41** ±0.13 |
| EATA | 20.12 ±0.14 | 21.52 ±0.17 | 21.40 ±0.21 | 20.90 ±0.09 | 23.42 ±0.20 | 15.71 ±0.10 | 18.00 ±0.27 | 16.12 ±0.11 | 28.35 ±0.24 | 22.24 ±0.16 | 35.97 ±0.22 | 11.33 ±0.15 | 19.78 ±0.29 | 20.22 ±0.18 | 19.99 ±0.23 | 21.00 ±0.12 |
| + SNAP | **40.74** ±0.15 | **43.22** ±0.19 | **43.11** ±0.13 | **40.63** ±0.27 | **44.59** ±0.11 | **51.58** ±0.21 | **50.63** ±0.18 | **54.77** ±0.12 | **58.32** ±0.25 | **61.50** ±0.20 | **73.91** ±0.14 | **33.85** ±0.29 | **60.19** ±0.10 | **63.35** ±0.17 | **63.01** ±0.23 | **52.23** ±0.13 |
| RoTTA | 21.44 ±0.14 | 18.64 ±0.21 | 22.08 ±0.13 | 19.97 ±0.18 | 16.87 ±0.22 | 13.70 ±0.09 | 14.42 ±0.27 | 15.73 ±0.19 | 28.57 ±0.19 | 17.71 ±0.24 | 35.05 ±0.21 | 8.52 ±0.13 | 15.80 ±0.20 | 13.03 ±0.11 | 13.27 ±0.25 | 18.32 ±0.10 |
| + SNAP | **35.68** ±0.16 | **34.60** ±0.17 | **36.86** ±0.21 | **31.20** ±0.13 | **25.81** ±0.23 | **36.24** ±0.09 | **33.47** ±0.26 | **42.72** ±0.12 | **55.50** ±0.27 | **51.74** ±0.18 | **71.84** ±0.15 | **17.84** ±0.29 | **42.86** ±0.14 | **42.24** ±0.10 | **52.67** ±0.24 | **40.75** ±0.11 |

Table 11: SNAP accuracy and latency per batch, using Vit-Base [4]. Performance averaged over ImageNet-C. Values in parentheses show the performance difference from full adaptation.

| Methods | Accuracy (%) | | Latency per batch (s) | |
|---|---|---|---|---|
| | Original TTA | SNAP (AR=0.1) | Original TTA | SNAP (AR=0.1) |
| Tent | 39.53 ±0.14 | **46.22 ±0.13 (+6.69)** | 28.19 ±0.08 | **15.66 ±0.06 (-44.43%)** |
| CoTTA | 41.25 ±0.12 | **39.41 ±0.15 (-1.83)** | 523.26 ±0.27 | **182.20 ±0.18 (-65.18%)** |
| EATA | 48.43 ±0.11 | **52.23 ±0.13 (+3.79)** | 28.69 ±0.07 | **16.44 ±0.06 (-42.71%)** |
| SAR | 43.86 ±0.13 | **47.77 ±0.12 (+3.91)** | 44.28 ±0.09 | **17.76 ±0.07 (-59.90%)** |
| RoTTA | 42.93 ±0.15 | **40.75 ±0.14 (-2.18)** | 42.70 ±0.10 | **16.28 ±0.08 (-61.88%)** |

## B.9 Modification for layer normalization of Vision Transformer

The main text describes the use of Batch Normalization (BN) statistics for calculating domain centroids and centroid-instance distances, with subsequent adjustment of memory statistics to match the target test batch using the Inference-only Batch-aware Memory Normalization (IoBMN) method. Specifically, these calculations leverage the mean and variance across batches as follows:

$$\bar{\mu}_c = \frac{1}{B \times L} \sum_{b=1}^{B} \sum_{l=1}^{L} f_{b,c,l}, \quad \bar{\sigma}_c^2 = \frac{1}{B \times L} \sum_{b=1}^{B} \sum_{l=1}^{L} (f_{b,c,l} - \mu_{b,c})^2, \tag{13}$$

where $B$ represents the batch size, $L$ the number of spatial locations, and $c$ the channel index.

However, modern models like Vision Transformer (ViT) utilize Layer Normalization (LN) instead of BN. Unlike BN, which calculates statistics across the entire batch, LN normalizes each instance independently by using the statistics calculated over individual feature dimensions. Specifically, for a feature vector $\mathbf{f}_b$ belonging to the $b$-th instance, LN computes:

$$\mu_b = \frac{1}{C} \sum_{c=1}^{C} f_{b,c}, \quad \sigma_b^2 = \frac{1}{C} \sum_{c=1}^{C} (f_{b,c} - \mu_b)^2, \tag{14}$$

where $C$ is the number of channels. This difference implies that LN operates without batch-level interactions, focusing solely on within-instance normalization, which makes the method inherently more suitable for handling variable batch sizes, particularly in latency-sensitive applications like those considered in our Test-Time Adaptation (TTA) setting.

Despite the differences between BN and LN, the fundamental mechanism of using feature statistics to capture domain information remains valid. The key domain characteristics in early layer features are preserved in both normalization types, enabling the construction of a domain centroid that reflects the distributional characteristics of the test data. For LN, this centroid can be computed by aggregating across instances instead of across batches:

$$\bar{\mu}_c^{\text{LN}} = \frac{1}{M} \sum_{b=1}^{M} \mu_b, \quad \bar{\sigma}_c^{2\text{LN}} = \frac{1}{M} \sum_{b=1}^{M} \sigma_b^2, \tag{15}$$

where $M$ is memory capacity. This modified approach allows the domain centroid to still represent the overall domain-specific characteristics effectively, despite the lack of direct batch-level statistics.

Furthermore, this methodology extends seamlessly to other normalization layers, such as Group Normalization (GN). In GN, the statistics are computed across smaller groups of channels within

each instance, but the procedure for aggregating these statistics to form a domain centroid remains the same—by averaging the group-level statistics across instances.

To maintain the core concept of selecting domain-representative samples with minimal modifications, we continue to use the memory of high-confidence domain-representative samples in the Inference-only Batch-aware Memory Normalization (IoBMN) strategy. The adjustment for LN requires: 1. Calculating LN-specific centroids as described in Equation 15. 2. Replacing BN statistics with LN statistics in the IoBMN module, thereby aligning the feature normalization during inference with the domain-representative information derived from memory.

The effectiveness of this modification was validated experimentally, as shown in Table 10, 11, where ViT models using LN showed improved performance even under sparse TTA conditions. This indicates that, with minimal adjustments, SNAP remains effective for ViT with LN. The core principle of utilizing domain-representative statistics for aligning test-time feature distributions continues to provide significant benefits, ensuring robust adaptation in shifting domains with limited latency and computational overhead.

### B.10   Comparison with forward-only TTA methods

Forward-only TTA methods, such as T3A [13] and FOA [29], aim to reduce computational burden by removing gradient-based updates. Instead, they update lightweight components: class prototypes in T3A and learned prompts in FOA. While these methods improve runtime efficiency, they exhibit structural limitations that hinder their robustness under dynamic distribution shifts.

Table 12: Performance comparison between T3A [13] and FOA [29] against Tent [48] + SNAP (adaptation rate 0.1) on ImageNet-C (i.i.d and non-i.i.d). Latency is measured on Raspberry Pi 4.

| Dataset | Method | Gau. | Shot | Imp. | Def. | Gla. | Mot. | Zoom | Snow | Fro. | Fog | Brit. | Cont. | Elas. | Pix. | JPEG | Avg. | Lat.(s) |
|---|---|---|---|---|---|---|---|---|---|---|---|---|---|---|---|---|---|---|
| | | | | | | | | | ResNet50 | | | | | | | | | |
| | T3A | 14.03 | 14.21 | 14.56 | 12.97 | 13.07 | 23.02 | 34.67 | 30.79 | 27.84 | 43.95 | 61.51 | 12.57 | 39.79 | 44.50 | 36.11 | 28.24 | 18.35 |
| | | ±0.09 | ±0.43 | ±0.07 | ±0.18 | ±0.06 | ±0.21 | ±1.17 | ±0.15 | ±0.29 | ±0.06 | ±0.22 | ±0.91 | ±0.19 | ±1.46 | ±0.51 | ±0.39 | ±0.59 |
| | Tent+SNAP | 26.21 | 27.85 | 27.50 | 23.62 | 22.73 | 36.01 | 44.11 | 42.19 | 38.15 | 52.95 | 64.57 | 30.23 | 48.56 | 53.71 | 47.09 | 39.03 | 18.76 |
| | | ±0.14 | ±0.19 | ±0.16 | ±0.10 | ±0.15 | ±0.23 | ±0.17 | ±0.12 | ±0.21 | ±0.18 | ±0.11 | ±0.27 | ±0.15 | ±0.13 | ±0.26 | ±0.19 | ±0.30 |
| ImageNet-C (i.i.d) | | | | | | | | | | ViT-Base | | | | | | | | |
| | FOA (K=28) | 41.34 | 40.96 | 42.68 | 36.27 | 29.94 | 40.34 | 40.89 | 47.20 | 59.07 | 63.71 | 72.95 | 46.98 | 44.75 | 45.40 | 55.87 | 47.22 | 694.8 |
| | | ±0.21 | ±0.39 | ±0.20 | ±0.06 | ±0.27 | ±0.34 | ±0.35 | ±2.03 | ±1.56 | ±0.14 | ±0.07 | ±0.12 | ±0.14 | ±0.38 | ±0.38 | ±0.51 | ±8.41 |
| | FOA (K=2) | 37.53 | 35.84 | 39.36 | 34.25 | 27.45 | 37.85 | 34.43 | 43.65 | 56.10 | 62.84 | 71.79 | 39.38 | 44.29 | 41.33 | 53.06 | 43.94 | 81.43 |
| | | ±0.30 | ±0.12 | ±0.03 | ±0.27 | ±0.16 | ±0.10 | ±0.12 | ±0.06 | ±0.11 | ±0.19 | ±0.04 | ±0.68 | ±0.18 | ±0.09 | ±0.05 | ±0.37 | ±4.12 |
| | Tent+SNAP | 41.98 | 42.72 | 43.18 | 38.16 | 33.30 | 43.89 | 39.44 | 47.19 | 53.50 | 54.11 | 73.25 | 40.25 | 47.77 | 52.53 | 56.99 | 47.23 | 83.51 |
| | | ±0.12 | ±0.22 | ±0.13 | ±0.11 | ±0.13 | ±0.21 | ±0.18 | ±0.09 | ±0.18 | ±0.20 | ±0.08 | ±0.25 | ±0.17 | ±0.12 | ±0.24 | ±0.16 | ±0.32 |
| | | | | | | | | | ResNet50 | | | | | | | | | |
| | T3A | 11.75 | 11.94 | 11.44 | 11.30 | 10.98 | 19.98 | 31.30 | 27.16 | 24.51 | 38.35 | 56.31 | 10.89 | 37.41 | 42.95 | 32.52 | 25.25 | 18.82 |
| | | ±0.10 | ±0.38 | ±0.06 | ±0.22 | ±0.05 | ±0.17 | ±1.34 | ±0.12 | ±0.27 | ±0.07 | ±0.18 | ±1.02 | ±0.23 | ±1.39 | ±0.49 | ±0.42 | ±0.57 |
| | Tent+SNAP | 24.00 | 24.69 | 24.78 | 21.37 | 21.15 | 32.20 | 42.83 | 39.26 | 35.41 | 51.17 | 62.89 | 21.25 | 46.95 | 52.31 | 45.60 | 36.39 | 18.98 |
| | | ±0.11 | ±0.25 | ±0.14 | ±0.13 | ±0.12 | ±0.19 | ±0.22 | ±0.10 | ±0.16 | ±0.24 | ±0.09 | ±0.28 | ±0.18 | ±0.15 | ±0.23 | ±0.14 | ±0.29 |
| ImageNet-C (non-i.i.d) | | | | | | | | | | ViT-Base | | | | | | | | |
| | FOA (K=28) | 38.36 | 36.23 | 39.47 | 33.07 | 25.35 | 36.44 | 35.80 | 42.86 | 55.45 | 61.67 | 70.60 | 39.87 | 41.81 | 43.05 | 52.42 | 43.50 | 710.2 |
| | | ±0.11 | ±0.41 | ±0.05 | ±0.20 | ±0.04 | ±0.19 | ±1.26 | ±0.14 | ±0.24 | ±0.08 | ±0.19 | ±0.96 | ±0.20 | ±1.43 | ±0.54 | ±0.40 | ±7.33 |
| | FOA (K=2) | 35.84 | 33.74 | 36.03 | 30.48 | 23.24 | 34.34 | 32.03 | 40.98 | 53.98 | 59.78 | 68.24 | 29.79 | 39.50 | 39.50 | 50.80 | 40.62 | 85.19 |
| | | ±0.17 | ±0.00 | ±0.81 | ±0.09 | ±0.04 | ±0.04 | ±0.06 | ±0.06 | ±0.07 | ±0.09 | ±0.04 | ±0.22 | ±0.03 | ±0.07 | ±0.10 | ±0.13 | ±5.84 |
| | Tent+SNAP | 38.02 | 38.31 | 40.22 | 35.37 | 30.20 | 40.17 | 37.20 | 44.57 | 54.78 | 52.01 | 73.19 | 24.68 | 45.64 | 47.08 | 55.66 | 43.81 | 87.54 |
| | | ±0.73 | ±0.50 | ±0.83 | ±0.11 | ±0.60 | ±0.48 | ±0.35 | ±0.54 | ±0.27 | ±0.77 | ±0.22 | ±0.32 | ±0.47 | ±0.81 | ±0.83 | ±0.52 | ±0.51 |

**Prototype-based.**   T3A maintains a fixed feature extractor and updates per-class prototypes using pseudo-labels from incoming samples. Although it avoids backpropagation, T3A accumulates a growing support set of query features per pseudo-label to refine class-wise prototypes. This not only increases memory usage, but also incurs non-negligible latency during inference, especially when the number of classes is large (e.g., 1000-way classification in ImageNet) or when operating on edge devices. Matching a test sample against all stored support features becomes increasingly costly in such settings. As shown in Table 12, T3A achieves latency comparable to Tent+SNAP but consistently performs worse in accuracy, especially under non-i.i.d. ImageNet-C streams. This illustrates the limitation of relying solely on forward-only label-space correction without feature-level adaptation.

**Prompt-based.**   FOA [29] introduces learnable prompts to adapt ViT encoders without backpropagation. However, it suffers from a trade-off between latency and accuracy, which is determined by the number of forward passes (K) required per test sample. While FOA theoretically avoids gradients, it performs K repeated forward steps to refine prompts, still incurring notable cost on edge devices. As shown in Table 12, FOA with its default configuration (K = 28) incurs significantly higher latency than Tent+SNAP, while achieving similar accuracy. Reducing K to 2 reduces latency to a comparable level, but results in substantial performance degradation, showing strong sensitivity to prompt update

depth. Moreover, FOA fails to generalize to CNNs, where the original paper [29] reports lower performance than Tent due to structural misalignment with the ViT-specific prompt.

**Conclusion.** In contrast, SNAP with Tent performs sparse backpropagation on confidently selected samples, striking a balance between low latency and high robustness. Despite using gradients, our latency profiling shows that Tent+SNAP remains within the latency range of forward-only methods, while outperforming them in accuracy across both i.i.d. and non-i.i.d. domains. This demonstrates the efficacy of targeted feature-level adaptation over purely forward-only correction.

## B.11 Robustness in challenging out-of-distribution domains

To validate SNAP under more visually challenging and abstract domain shifts, we additionally apply SNAP to two nontrivial out-of-distribution datasets: **ImageNet-R** and **ImageNet-Sketch**. These datasets feature semantic deformation (R), sketch-style abstraction (Sketch), and both differ significantly from typical texture- and structure-rich natural images seen in ImageNet.

We test SNAP with an adaptation rate (AR) of AR=0.1 on five representative TTA backbones (Tent, CoTTA, EATA, SAR, and RoTTA) and compare results with full adaptation (AR=1.0). As summarized in Table 13, SNAP consistently maintains competitive performance with considerably reduced latency, demonstrating its suitability for low-overhead deployment under difficult real-world shifts.

Table 13: Performance of SNAP (AR=0.1) on ImageNet-R and ImageNet-Sketch. Accuracy and latency are reported as the mean $\pm$ standard deviation over 3 seeds (0, 1, 2). Latency is measured by Raspberry Pi 4.

| Method | ImageNet-R | | ImageNet-Sketch | |
|---|---|---|---|---|
| | Accuracy (%) | Latency (s) | Accuracy (%) | Latency (s) |
| Tent (Full) | $40.53 \pm 0.22$ | $34.30 \pm 0.06$ | $28.43 \pm 0.18$ | $38.12 \pm 0.05$ |
| + SNAP (AR=0.1) | $38.26 \pm 0.20$ | $17.73 \pm 0.05$ | $28.42 \pm 0.16$ | $18.50 \pm 0.11$ |
| CoTTA (Full) | $37.53 \pm 0.25$ | $295.19 \pm 0.07$ | $24.07 \pm 0.19$ | $302.80 \pm 0.12$ |
| + SNAP (AR=0.1) | $36.73 \pm 0.21$ | $150.40 \pm 0.06$ | $23.01 \pm 0.19$ | $158.32 \pm 0.21$ |
| EATA (Full) | $42.88 \pm 0.18$ | $29.22 \pm 0.02$ | $30.52 \pm 0.20$ | $32.99 \pm 0.08$ |
| + SNAP (AR=0.1) | $39.45 \pm 0.17$ | $15.50 \pm 0.08$ | $27.43 \pm 0.18$ | $17.53 \pm 0.23$ |
| SAR (Full) | $40.37 \pm 0.22$ | $72.51 \pm 0.05$ | $27.03 \pm 0.19$ | $76.37 \pm 0.05$ |
| + SNAP (AR=0.1) | $37.32 \pm 0.20$ | $19.59 \pm 0.06$ | $27.95 \pm 0.17$ | $22.52 \pm 0.06$ |
| RoTTA (Full) | $39.08 \pm 0.21$ | $78.05 \pm 0.09$ | $26.05 \pm 0.16$ | $84.98 \pm 0.06$ |
| + SNAP (AR=0.1) | $36.79 \pm 0.19$ | $41.15 \pm 0.06$ | $24.08 \pm 0.15$ | $49.33 \pm 0.16$ |

## B.12 Efficient strategy for re-calculation of sample's distance

The domain centroid in our framework is updated using a momentum-based approach to effectively capture recent shifts in the target domain. This ensures that the centroid remains adaptive to evolving distributions without being overly influenced by temporary fluctuations. However, during sparse adaptation (SA), where model updates occur at extended intervals, the data distribution can shift substantially between updates. Consequently, distances calculated for older samples may become outdated, leading to inconsistencies when comparing them to more recently added samples that are evaluated based on the updated centroid.

To address this issue efficiently, our Class and Domain Representative Memory (CnDRM) recalculates the distance of samples only when the shift in the domain centroid exceeds a predefined significance threshold. Specifically, if the change in the domain centroid $\Delta \mathbf{c}_{\text{domain}}$ surpasses a threshold $\tau_\Delta$, the distances of all samples in memory are updated to reflect the new domain conditions. This threshold-based approach ensures that recalculations occur only when necessary, thereby minimizing computational costs while maintaining the representativeness of the memory.

In practice, we observed that the performance was not significantly affected as long as the threshold $\tau_\Delta$ was not set too high, indicating robustness to the choice of threshold. Based on these observations, we set $\tau_\Delta = 0.1$ and used this value consistently for all evaluations. By focusing recalculations on significant shifts, this strategy preserves consistency in sample selection, ensuring that both older and newer samples are compared fairly in the context of the current domain characteristics without excessive computational overhead.

## B.13 Strategy for continuous domain shift setting

In our proposed framework, the centroid used for selecting domain-representative samples naturally adapts to changes in the domain as new data is encountered. This mechanism inherently ensures that the centroid evolves to reflect the characteristics of the current domain, allowing for effective performance even under continual Test-Time Adaptation (TTA) scenarios, where the domain may gradually or abruptly shift during adaptation.

Instead of employing additional mechanisms like z-score evaluation to detect domain shifts, we rely on the natural adaptability of the centroid to adjust to the incoming data. This simplifies the design and avoids unnecessary overhead while maintaining robustness. As the domain characteristics evolve, the centroid continuously aligns with the new domain without requiring explicit detection of changes or manual intervention.

To validate the effectiveness of SNAP under continual domain shift scenarios, we conducted experiments across various benchmark datasets with incremental and abrupt domain shifts. Table 14 summarizes the results, demonstrating that SNAP maintains strong performance across evolving domains without requiring additional computational overhead for explicit domain shift detection.

These results indicate that SNAP effectively handles both incremental and abrupt domain shifts, consistently outperforming baseline methods. By leveraging the natural adaptability of the centroid, SNAP provides a robust solution for continual domain adaptation in real-world scenarios. Notably, SNAP mitigates catastrophic forgetting not only through its sparse adaptation strategy but also by leveraging domain centroid-based sampling, allowing performance to be sustained longer in continual shift scenarios. Unlike Tent, CoTTA is specifically designed for continual domain shift environments, which highlights its superior performance under such conditions.

Future work could explore augmenting this adaptive mechanism by incorporating techniques like z-score evaluation to enable even more responsive adjustments. For instance, a z-score-based approach could further refine the centroid's responsiveness to subtle, gradual domain shifts by monitoring discrepancies between incoming data statistics and the current centroid. Such enhancements could make the system even more effective at handling continual domain evolution, particularly in scenarios with complex or noisy data streams.

Table 14: Performance of SNAP under continual domain shift scenarios. The table reports the accuracy (%) for different datasets with incremental and abrupt shifts. **Bold** numbers are the highest accuracy.

| AR | Method | Gau. | Shot | Imp. | Def. | Gla. | Mot. | Zoom | Snow | Fro. | Fog | Brit. | Cont. | Elas. | Pix. | JPEG | Avg. |
|---|---|---|---|---|---|---|---|---|---|---|---|---|---|---|---|---|---|
| 0.1 | Tent | 24.68 ±0.45 | 19.65 ±1.27 | 5.12 ±1.22 | 0.63 ±0.05 | 0.43 ±0.02 | 0.40 ±0.04 | 0.44 ±0.06 | 0.41 ±0.03 | 0.30 ±0.03 | 0.33 ±0.04 | 0.42 ±0.05 | 0.24 ±0.04 | 0.32 ±0.02 | 0.31 ±0.05 | 0.31 ±0.04 | 3.60 ±0.23 |
| | + SNAP | **28.71** ±0.66 | **30.60** ±1.82 | **22.91** ±2.25 | **6.13** ±0.90 | **1.62** ±0.20 | **0.87** ±0.13 | **0.88** ±0.07 | **0.64** ±0.08 | **0.64** ±0.06 | **0.66** ±0.05 | **0.75** ±0.01 | **0.44** ±0.05 | **0.60** ±0.08 | **0.63** ±0.07 | **0.61** ±0.07 | **6.45** ±0.43 |
| | CoTTA | 10.99 ±0.40 | 12.21 ±0.04 | 11.54 ±0.30 | 11.28 ±0.13 | 11.13 ±0.15 | 22.08 ±0.07 | 34.80 ±0.18 | 30.69 ±0.10 | 29.45 ±0.04 | 43.87 ±0.19 | 61.92 ±0.09 | 12.76 ±0.16 | 40.03 ±0.13 | 44.99 ±0.14 | 36.43 ±0.16 | 27.61 ±0.15 |
| | + SNAP | **15.19** ±0.17 | **15.97** ±0.06 | **15.91** ±0.02 | **13.94** ±0.04 | **14.18** ±0.03 | **24.76** ±0.07 | **36.50** ±0.23 | **32.61** ±0.04 | **31.76** ±0.06 | **46.14** ±0.10 | **63.60** ±0.14 | **15.60** ±0.04 | **42.17** ±0.02 | **46.77** ±0.06 | **38.08** ±0.12 | **30.21** ±0.08 |
| 0.05 | Tent | 23.31 ±0.37 | 27.08 ±1.13 | 22.71 ±2.50 | 9.72 ±3.35 | 4.14 ±3.00 | 2.03 ±1.53 | 1.16 ±0.75 | 0.66 ±0.22 | 0.45 ±0.12 | 0.47 ±0.09 | 0.61 ±0.16 | 0.33 ±0.09 | 0.47 ±0.08 | 0.47 ±0.08 | 0.46 ±0.07 | 6.27 ±0.90 |
| | + SNAP | **27.10** ±0.23 | **33.41** ±0.10 | **31.78** ±0.62 | **19.85** ±0.79 | **16.94** ±1.50 | **14.75** ±2.53 | **12.46** ±4.27 | **5.53** ±2.30 | **2.69** ±1.18 | **1.47** ±0.49 | **1.52** ±0.40 | **0.67** ±0.09 | **0.88** ±0.10 | **0.89** ±0.10 | **0.84** ±0.07 | **11.39** ±0.98 |
| | CoTTA | 11.04 ±0.38 | 12.25 ±0.39 | 11.73 ±0.42 | 11.62 ±0.10 | 11.25 ±0.59 | 22.05 ±0.13 | 34.89 ±0.13 | 30.73 ±0.17 | 29.50 ±0.13 | 44.09 ±0.20 | 61.87 ±0.18 | 12.87 ±0.09 | 40.15 ±0.18 | 45.06 ±0.17 | 36.53 ±0.14 | 27.71 ±0.23 |
| | + SNAP | **15.20** ±0.15 | **15.89** ±0.02 | **15.93** ±0.10 | **13.81** ±0.04 | **14.15** ±0.03 | **24.74** ±0.16 | **36.68** ±0.27 | **32.51** ±0.04 | **31.71** ±0.20 | **46.11** ±0.05 | **63.48** ±0.09 | **15.73** ±0.19 | **42.20** ±0.12 | **46.69** ±0.10 | **38.05** ±0.04 | **30.19** ±0.10 |

## B.14 Robustness under persistent distribution shifts

To evaluate the long-term stability of SNAP under temporally correlated and recurring domain shifts, we adopt the evaluation setup (persist-TTA) of work [11], which repeatedly cycles through non-i.i.d. CIFAR10-C corruptions across 10 rounds. This scenario emulates continual adaptation in environments where domain drift is both persistent and revisited.

We apply SNAP on top of CoTTA [50], a method specifically designed for continual test-time adaptation. While CoTTA alone initially performs well, we observe a steady degradation across rounds as it accumulates shift-induced bias and overfits to recent domains. In contrast, combining

CoTTA with SNAP enables the model to preserve robust performance even after multiple adaptation cycles.

The key to this stability lies in SNAP's architecture: (1) class-balanced memory to prevent label bias, (2) sparse but confident updates to mitigate overfitting, (3) IoBMN for adapting normalization statistics to each incoming sample, and (4) exponential moving average (EMA) domain centroids to smooth domain shift tracking. These collectively stabilize long-term adaptation dynamics without the need for explicit shift detection.

Table 15: Long-term TTA accuracy, cycling through all CIFAR10-C corruptions each round (R).

| Method | R1 $\Longrightarrow$ | R2 $\Longrightarrow$ | R3 $\Longrightarrow$ | R4 $\Longrightarrow$ | R5 $\Longrightarrow$ | R6 $\Longrightarrow$ | R7 $\Longrightarrow$ | R8 $\Longrightarrow$ | R9 $\Longrightarrow$ | R10 |
|---|---|---|---|---|---|---|---|---|---|---|
| CoTTA (Full adapt) | 58.49 ±0.12 | 50.32 ±0.09 | 48.71 ±0.15 | 44.92 ±0.10 | 31.86 ±0.17 | 29.81 ±0.13 | 25.39 ±0.16 | 21.43 ±0.11 | 28.46 ±0.18 | 16.97 ±0.14 |
| + SNAP (AR=0.1) | 50.50 ±0.10 | 49.32 ±0.08 | 52.57 ±0.11 | 49.90 ±0.09 | 50.17 ±0.12 | 50.58 ±0.14 | 49.01 ±0.07 | 49.56 ±0.13 | 54.16 ±0.10 | 52.89 ±0.09 |

**Experimental results.** Table 15 and Figure 9 shows adaptation accuracy over 10 rounds. While CoTTA gradually collapses after the 5th round, integration with SNAP maintains accuracy above 50%, showing clear stability under persistent shifts.

These results confirm that SNAP enhances long-term TTA robustness by stabilizing both parameter and feature statistics over time. This makes it a reliable plug-in module for continual test-time adaptation pipelines.

### B.15 Robustness in single-sample (BS=1) adaptation scenario

To investigate the robustness of SNAP when adaptation is performed on a per-sample basis, we evaluate its performance in a single-sample adaptation setting, where the adaptation batch size is limited to 1. This scenario reflects highly constrained edge environments with limited memory or streaming inputs, where adaptation must occur with minimal latency and granularity.

To support this setup, we adopt the SAR [31] architecture, which natively supports a batch size of 1 and sparse update routines. SAR allows us to test SNAP with an adaptation rate of 0.1, meaning only one in every ten test samples is used for weight updates, using a single memory sample.

Table 16: Evaluation of SNAP (AR=0.1) with SAR on a single-sample (BS=1) adaptation scenario on ImageNet-C. Results are averaged over 3 random seeds (0, 1, 2).

| Method | Accuracy (%) |
|---|---|
| SAR (single-sample) | $52.21 \pm 0.28$ |
| + STTA | $8.06 \pm 0.12$ |
| **+ SNAP** | $\mathbf{51.80 \pm 0.25}$ |

SNAP achieves strong gains even when adapting from just one memory sample. When only one sample is used for adaptation, our method still selects the representative sample with high prediction confidence and low Wasserstein distance to the domain centroid, enabling stable model updates.

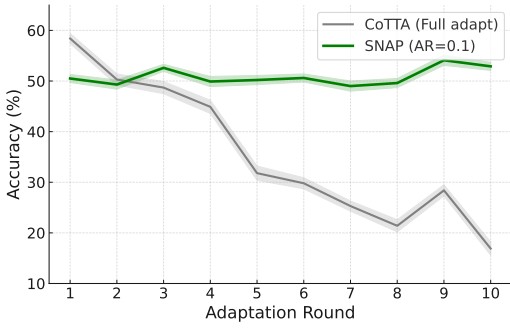

Figure 9: Long-term TTA accuracy over 10 adaptation rounds. Shaded regions indicate std over 3 seeds. SNAP maintains stable performance, while original CoTTA alone degrades over time.

Meanwhile, IoBMN continues to apply adaptive normalization to every incoming test sample, mitigating covariate shift even when parameter updates are sparse. Although small batches can slow the class distribution balancing process and moving domain centroid updates under skewed domains, we found that this effect has limited influence on overall adaptation performance and stability.

**Evaluation protocol.** We average performance over three random seeds (0, 1, 2) to ensure stability across different data orderings. Table 16 reports both the mean and standard deviation of accuracy.

These results demonstrate that SNAP preserves its robustness and sample-wise adaptability even under minimal adaptation frequency and granularity. CnDRM identifies meaningful memory samples, while IoBMN provides per-sample normalization, together enabling consistent performance in BS=1 settings.

## B.16 Impact of memory size on SNAP performance

The memory size of the Class and Domain Representative Memory (CnDRM) in SNAP has implications for both performance and privacy. Increasing memory size allows storing more samples, which intuitively could improve adaptation. However, such an approach raises privacy concerns and needs additional memory and latency when storing sensitive samples. To evaluate the trade-off, we conducted experiments on ImageNet-C under Gaussian noise corruption, using Tent + SNAP(adaptation rate 0.3) with a batch size of 16 and varying the memory size.

As shown in Table 17, increasing the memory size beyond the base configuration of 16 does not lead to significant performance gains. This observation highlights the efficiency of SNAP's representative sampling strategy, which prioritizes storing samples based on proximity to class and domain centroids. The saturation in accuracy suggests that a carefully aligned memory size to the batch size is sufficient to balance computational efficiency, performance, and privacy considerations.

Table 17: Performance comparison with varying memory sizes on ImageNet-C.

| Memory Size | Accuracy (%) |
|---|---|
| 16 (Base) | 26.60 $\pm0.11$ |
| 32 | 28.44 $\pm0.17$ |
| 64 | 28.89 $\pm0.06$ |
| 128 | 28.60 $\pm0.09$ |

In conclusion, to minimize computational overhead while ensuring robust test-time adaptation, the memory size in SNAP is designed to align with the batch size. This configuration addresses privacy and memory overhead risks by limiting the number of stored samples without compromising adaptation effectiveness.

## B.17 Effect of learning rate on sparse and full adaptation

To investigate the impact of learning rates on the performance of SNAP and baseline methods, we conducted experiments under sparse adaptation settings. Initially, the same learning rate was applied for each SOTA TTA algorithms across all adaptation rates to ensure fair comparisons (Table 26, 27, 22, 23, 24,and 25). However, as sparse adaptation inherently limits the number of updates, the updates might be insufficient at lower adaptation rates and explored the effect of increasing the learning rate.

The results, summarized in Table 18, 19, and 20, reveal that higher learning rates improve the accuracy of both the naive baseline and SNAP under sparse settings. Notably, while the naive TTA baseline benefits from a higher learning rate, its performance still falls short of that achieved with full adaptation. In contrast, SNAP surpasses the performance of full adaptation at optimal learning rates, demonstrating its ability to leverage sparse adaptation effectively. At the same time, applying these higher learning rates to full adaptation results in model instability and collapse, underscoring the need to carefully tune learning rates based on adaptation frequency. Therefore, we selected a stable learning rate of $1 \times 10^{-4}$ for the evaluations in our work that balances model convergence and performance across all adaptation rates. These findings suggest that SNAP not only adapts effectively under sparse settings but also maintains robustness under optimized learning rates.

Table 18: Accuracy with varying learning rates on ImageNet-C Gaussian noise adaptation rate 0.5. **Bold** numbers are the highest accuracy.

| Learning rate | Tent | | | CoTTA | | | EATA | | |
|---|---|---|---|---|---|---|---|---|---|
| | Full | naïve STTA | SNAP | Full | naïve STTA | SNAP | Full | naïve STTA | SNAP |
| $2 \times 10^{-3}$ | 2.31 ±0.08 | 4.16 ±0.12 | 6.68 ±0.14 | 13.31 ±0.10 | 12.03 ±0.09 | 14.58 ±0.11 | 0.36 ±0.05 | 0.48 ±0.07 | 0.69 ±0.06 |
| $1 \times 10^{-3}$ | 4.54 ±0.11 | 10.19 ±0.14 | 16.37 ±0.09 | 13.18 ±0.13 | 11.98 ±0.08 | 14.63 ±0.12 | 1.31 ±0.06 | 1.36 ±0.05 | 22.11 ±0.15 |
| $5 \times 10^{-4}$ | 10.22 ±0.12 | 18.43 ±0.13 | **28.36** ±0.10 | 13.15 ±0.09 | 11.95 ±0.07 | **15.17** ±0.11 | 21.96 ±0.14 | 13.97 ±0.12 | 25.42 ±0.13 |
| $1 \times 10^{-4}$ | **27.03** ±0.13 | **25.24** ±0.10 | 28.05 ±0.09 | 13.12 ±0.10 | 11.99 ±0.08 | 15.16 ±0.07 | **29.42** ±0.11 | **28.62** ±0.13 | **30.00** ±0.12 |
| $5 \times 10^{-5}$ | 26.34 ±0.09 | 22.62 ±0.10 | 26.32 ±0.11 | **13.34** ±0.10 | **12.10** ±0.07 | 14.93 ±0.09 | 29.37 ±0.13 | 27.30 ±0.11 | 28.76 ±0.12 |

Table 19: Accuracy with varying learning rates on ImageNet-C Gaussian noise adaptation rate 0.3. **Bold** numbers are the highest accuracy.

| Learning rate | Tent | | | CoTTA | | | EATA | | |
|---|---|---|---|---|---|---|---|---|---|
| | Full | naïve STTA | SNAP | Full | naïve STTA | SNAP | Full | naïve STTA | SNAP |
| $2 \times 10^{-3}$ | 2.31 ±0.09 | 7.04 ±0.13 | 13.69 ±0.15 | 13.31 ±0.10 | 11.88 ±0.08 | 14.67 ±0.11 | 0.36 ±0.04 | 0.59 ±0.05 | 0.75 ±0.06 |
| $1 \times 10^{-3}$ | 4.54 ±0.12 | 16.13 ±0.14 | 27.63 ±0.13 | 13.18 ±0.09 | 11.86 ±0.07 | 14.68 ±0.10 | 1.31 ±0.06 | 0.95 ±0.05 | 24.35 ±0.14 |
| $5 \times 10^{-4}$ | 10.22 ±0.13 | **24.96** ±0.15 | **29.95** ±0.12 | 13.15 ±0.08 | 11.85 ±0.06 | 15.11 ±0.10 | 21.96 ±0.13 | 20.96 ±0.12 | 27.72 ±0.11 |
| $1 \times 10^{-4}$ | **27.03** ±0.10 | 23.63 ±0.11 | 26.60 ±0.12 | 13.12 ±0.07 | 11.74 ±0.06 | **15.26** ±0.08 | **29.42** ±0.10 | **27.35** ±0.09 | **29.48** ±0.10 |
| $5 \times 10^{-5}$ | 26.34 ±0.09 | 20.94 ±0.12 | 24.87 ±0.13 | **13.34** ±0.07 | **11.92** ±0.06 | 14.85 ±0.09 | 29.37 ±0.11 | 26.07 ±0.10 | 27.90 ±0.11 |

Table 20: Accuracy with varying learning rates on ImageNet-C Gaussian noise adaptation rate 0.1. **Bold** numbers are the highest accuracy.

| Learning rate | Tent | | | CoTTA | | | EATA | | |
|---|---|---|---|---|---|---|---|---|---|
| | Full | naïve STTA | SNAP | Full | naïve STTA | SNAP | Full | naïve STTA | SNAP |
| $2 \times 10^{-3}$ | 2.31 ±0.10 | 18.06 ±0.14 | 27.41 ±0.12 | 13.31 ±0.08 | 10.93 ±0.07 | 14.80 ±0.11 | 0.36 ±0.03 | 1.86 ±0.06 | 9.59 ±0.15 |
| $1 \times 10^{-3}$ | 4.54 ±0.11 | **25.46** ±0.13 | **31.12** ±0.14 | 13.18 ±0.09 | 10.93 ±0.07 | 14.73 ±0.10 | 1.31 ±0.05 | 2.86 ±0.08 | 24.95 ±0.13 |
| $5 \times 10^{-4}$ | 10.22 ±0.12 | 24.71 ±0.14 | 28.01 ±0.11 | 13.15 ±0.07 | 10.92 ±0.06 | **15.18** ±0.09 | 21.96 ±0.12 | 18.76 ±0.10 | **28.09** ±0.11 |
| $1 \times 10^{-4}$ | **27.03** ±0.10 | 22.00 ±0.12 | 26.21 ±0.13 | 13.12 ±0.08 | **11.74** ±0.06 | 15.13 ±0.09 | **29.42** ±0.11 | **22.43** ±0.10 | 26.10 ±0.12 |
| $5 \times 10^{-5}$ | 26.34 ±0.09 | 16.72 ±0.13 | 19.31 ±0.12 | **13.34** ±0.08 | 10.92 ±0.07 | 14.76 ±0.09 | 29.37 ±0.11 | 20.32 ±0.10 | 23.28 ±0.10 |

In conclusion, selecting an appropriately high learning rate for sparse adaptation significantly enhances performance while ensuring model stability. This strategy is particularly useful for real-world deployment of SNAP, where computational efficiency and robust performance are paramount.

## B.18 Evaluation on real-world sensor data

To validate SNAP's generalizability to other real-world domains, we further test SNAP on HARTH [23], a human activity recognition dataset that collects data from two three-axial accelerometers attached to participants' thigh and lower back. Unlike our main evaluations, which focus on 2D vision and corruption-based domain shifts, HARTH introduces a distinct domain shift caused by *sensor positioning* and *user variation*.

We evaluate SNAP with an adaptation rate (AR) of 0.1 on Tent [48] and SAR [31]. The base model is composed of four one-dimensional convolutional layers followed by a fully-connected layer, and is trained on the source domain, composing of data collected from the back of 15 participants. The target domain is the data collected from the thigh of the remaining 7 participants. As shown in Table 21, SNAP improves accuracy even with sparse updates, demonstrating its effectiveness under realistic shifts

Table 21: Performance of SNAP (AR=0.1) on HARTH. Accuracy is averaged over all target domain users.

| Method | Average Accuracy (%) |
|---|---|
| Tent (Naïve STTA) | 19.64 |
| **+ SNAP** | **30.67** |
| SAR (Naïve STTA) | 21.10 |
| **+ SNAP** | **26.63** |

# C   Detailed experiment results

In this section, we provide detailed experimental results for the performance comparison of SNAP across a wide range of adaptation rates. We evaluated the performance on CIFAR10-C, CIFAR100-C, and ImageNet-C datasets with adaptation rates of 0.01, 0.03, 0.05, 0.1, 0.3, and 0.5, and across five state-of-the-art (SOTA) TTA algorithms: Tent [48], EATA [30], SAR [31], CoTTA [50], and RoTTA [53]. This comprehensive evaluation resulted in a total of 150 combinations (3 datasets, 6 adaptation rates, 5 algorithms).

The results demonstrate that, regardless of the adaptation rate, dataset, or the TTA algorithm, integrating SNAP consistently outperforms the baseline methods. Specifically, SNAP achieved the highest accuracy across nearly all of these 150 combinations, effectively demonstrating its robustness in both high and low adaptation settings. For CIFAR10-C and CIFAR100-C, SNAP showed substantial performance improvements compared to the baseline, even at very low adaptation rates (e.g., 0.01 and 0.05). Similarly, for ImageNet-C, SNAP maintained superior accuracy across diverse corruption types.

These results highlight that SNAP effectively balances adaptation and latency, ensuring optimal performance even when the adaptation rate is sparse and regardless of the underlying TTA algorithm. This consistent superiority across all 150 combinations underscores SNAP's suitability for practical, real-world applications on resource-constrained devices.

## C.1 CIFAR10-C

Table 22: STTA classification accuracy (%) comparing with and without SNAP on CIFAR10-C through Adaptation Rates(AR) (0.5, 0.3, and 0.1), including results for full adaptation (AR=1). **Bold** numbers are the highest accuracy.

| AR | Methods | Gau. | Shot | Imp. | Def. | Gla. | Mot. | Zoom | Snow | Fro. | Fog | Brit. | Cont. | Elas. | Pix. | JPEG | Avg. |
|---|---|---|---|---|---|---|---|---|---|---|---|---|---|---|---|---|---|
| | Source | 22.13 ±0.00 | 29.25 ±0.00 | 22.53 ±0.00 | 54.54 ±0.00 | 55.10 ±0.00 | 67.45 ±0.00 | 64.37 ±0.00 | 78.25 ±0.00 | 69.93 ±0.00 | 74.26 ±0.00 | 91.29 ±0.00 | 35.45 ±0.00 | 77.20 ±0.00 | 46.56 ±0.00 | 73.38 ±0.00 | 57.45 ±0.00 |
| | BN stats | 63.72 ±0.48 | 65.67 ±0.12 | 57.14 ±0.25 | 84.99 ±0.31 | 62.72 ±0.23 | 83.86 ±0.48 | 84.26 ±0.30 | 78.98 ±0.30 | 76.95 ±0.08 | 83.32 ±0.17 | 88.46 ±0.16 | 84.60 ±0.17 | 73.96 ±0.18 | 76.61 ±0.02 | 68.79 ±0.42 | 75.60 ±0.24 |
| | Tent | 73.66 ±0.88 | 76.18 ±0.94 | 68.04 ±1.32 | 86.61 ±0.50 | 67.12 ±0.76 | 85.73 ±0.38 | 86.24 ±0.09 | 82.34 ±0.94 | 81.56 ±0.64 | 86.02 ±0.18 | 89.99 ±0.16 | 87.16 ±2.50 | 76.40 ±0.82 | 82.95 ±0.15 | 76.45 ±0.46 | 80.43 ±0.71 |
| 1 | CoTTA | 71.95 ±0.32 | 73.97 ±0.48 | 67.03 ±0.66 | 83.91 ±0.20 | 66.75 ±0.08 | 82.64 ±0.34 | 83.34 ±0.19 | 79.92 ±0.09 | 79.49 ±0.13 | 82.41 ±0.23 | 88.39 ±0.18 | 80.14 ±0.17 | 75.38 ±0.09 | 79.24 ±0.07 | 75.42 ±0.25 | 78.00 ±0.23 |
| | EATA | 75.82 ±0.50 | 77.61 ±0.27 | 69.63 ±0.87 | 87.14 ±0.29 | 69.41 ±0.68 | 85.96 ±0.39 | 87.08 ±0.27 | 83.42 ±0.38 | 82.28 ±0.29 | 86.58 ±0.41 | 90.40 ±0.17 | 89.26 ±0.39 | 77.62 ±0.28 | 83.35 ±0.32 | 77.77 ±0.20 | 81.56 ±0.38 |
| | SAR | 73.52 ±1.53 | 74.03 ±0.46 | 65.45 ±1.81 | 85.69 ±0.37 | 65.01 ±0.35 | 84.63 ±0.34 | 85.01 ±0.37 | 81.47 ±0.72 | 80.91 ±0.09 | 84.18 ±0.12 | 88.70 ±0.16 | 86.23 ±0.03 | 74.94 ±0.28 | 81.20 ±0.69 | 74.84 ±0.62 | 79.05 ±0.52 |
| | RoTTA | 66.54 ±0.46 | 68.60 ±0.23 | 60.27 ±0.46 | 85.73 ±0.35 | 64.84 ±0.63 | 84.68 ±0.36 | 85.01 ±0.45 | 80.15 ±0.56 | 78.02 ±0.06 | 84.13 ±0.09 | 89.00 ±0.27 | 84.91 ±0.19 | 75.06 ±0.15 | 77.96 ±0.16 | 70.12 ±0.36 | 77.00 ±0.32 |
| | Tent | 73.44 ±0.61 | 75.93 ±0.44 | 67.18 ±0.78 | 86.52 ±0.17 | 67.28 ±1.78 | 85.25 ±0.49 | 86.23 ±0.42 | 82.24 ±0.77 | 80.35 ±0.14 | 85.39 ±0.20 | 89.80 ±0.28 | 87.77 ±0.27 | 77.00 ±0.65 | 82.08 ±0.68 | 75.58 ±0.60 | 80.14 ±0.55 |
| | **+ SNAP** | **75.17** ±0.00 | **77.66** ±0.78 | **68.78** ±1.26 | **88.25** ±0.38 | **69.18** ±0.51 | **87.11** ±0.18 | **88.19** ±0.13 | **84.21** ±0.29 | **82.72** ±0.45 | **87.34** ±0.51 | **91.63** ±0.12 | **86.30** ±1.07 | **78.76** ±0.28 | **83.43** ±0.18 | **77.28** ±0.50 | **81.74** ±0.44 |
| | CoTTA | 65.08 ±0.26 | 66.67 ±0.21 | 61.30 ±0.16 | 77.50 ±0.48 | 61.36 ±0.15 | 77.70 ±0.37 | 77.37 ±0.37 | 74.05 ±0.22 | 72.86 ±0.44 | 77.43 ±0.19 | 82.69 ±0.30 | 72.44 ±0.72 | 70.52 ±0.07 | 70.94 ±0.27 | 69.79 ±0.10 | 71.85 ±0.29 |
| | **+ SNAP** | **71.89** ±0.45 | **74.18** ±0.33 | **66.92** ±0.19 | **85.46** ±0.32 | **67.57** ±0.26 | **84.27** ±0.22 | **84.91** ±0.09 | **81.10** ±0.46 | **80.62** ±0.24 | **84.06** ±0.17 | **90.16** ±0.33 | **82.14** ±0.16 | **76.75** ±0.38 | **80.23** ±0.50 | **75.98** ±0.28 | **79.08** ±0.28 |
| | EATA | 73.95 ±0.22 | 75.82 ±0.18 | 68.00 ±0.70 | 86.83 ±0.25 | 67.83 ±0.50 | 85.27 ±0.39 | 86.48 ±0.15 | 82.63 ±0.50 | 80.99 ±0.05 | 85.45 ±0.16 | 89.86 ±0.18 | 87.61 ±0.53 | 77.01 ±0.31 | 82.13 ±0.18 | 76.11 ±0.45 | 80.40 ±0.32 |
| 0.5 | **+ SNAP** | **74.85** ±0.51 | **77.63** ±0.46 | **68.43** ±0.43 | **88.53** ±0.17 | **69.70** ±0.69 | **87.19** ±0.35 | **88.16** ±0.18 | **83.87** ±0.42 | **82.84** ±0.33 | **87.18** ±0.15 | **91.54** ±0.12 | **89.62** ±0.38 | **78.91** ±0.48 | **83.76** ±0.14 | **77.36** ±0.22 | **81.97** ±0.33 |
| | SAR | 69.10 ±1.63 | 72.37 ±1.05 | 63.22 ±0.44 | 85.18 ±0.25 | 64.30 ±1.02 | 83.94 ±0.12 | 85.07 ±0.45 | 80.11 ±0.17 | 79.64 ±0.60 | 83.91 ±0.37 | 88.64 ±0.10 | 84.21 ±0.30 | 75.70 ±0.34 | 79.10 ±0.52 | 72.92 ±0.09 | 77.83 ±0.50 |
| | **+ SNAP** | **73.98** ±0.48 | **75.48** ±0.65 | **66.41** ±1.26 | **86.63** ±0.15 | **68.15** ±0.07 | **85.50** ±0.15 | **86.53** ±0.10 | **81.62** ±0.39 | **80.20** ±0.17 | **85.06** ±0.27 | **91.46** ±0.03 | **87.04** ±0.11 | **77.22** ±0.45 | **81.16** ±0.27 | **75.53** ±0.23 | **80.13** ±0.32 |
| | RoTTA | 65.02 ±0.04 | 66.84 ±0.52 | 58.38 ±0.33 | 85.26 ±0.42 | 63.51 ±0.18 | 83.81 ±0.15 | 84.66 ±0.20 | 79.26 ±0.29 | 76.76 ±0.49 | 83.46 ±0.21 | 88.27 ±0.04 | 83.47 ±0.05 | 74.43 ±0.16 | 77.39 ±0.29 | 69.13 ±0.41 | 75.98 ±0.25 |
| | **+ SNAP** | **66.03** ±0.14 | **68.09** ±0.15 | **58.88** ±0.06 | **87.09** ±0.27 | **64.55** ±0.07 | **85.70** ±0.03 | **86.48** ±0.13 | **80.97** ±0.10 | **78.87** ±0.22 | **85.29** ±0.22 | **90.28** ±0.13 | **86.22** ±0.10 | **76.05** ±0.22 | **78.76** ±0.22 | **70.51** ±0.35 | **77.58** ±0.16 |
| | Tent | 71.18 ±0.99 | 74.06 ±0.80 | 65.44 ±1.17 | 85.93 ±0.28 | 66.01 ±0.97 | 84.37 ±0.14 | 85.90 ±0.17 | 81.31 ±0.40 | 79.80 ±0.09 | 84.80 ±0.25 | 89.58 ±0.23 | 84.01 ±0.30 | 75.96 ±0.30 | 80.46 ±0.39 | 74.09 ±0.54 | 78.86 ±0.47 |
| | **+ SNAP** | **74.95** ±0.84 | **77.29** ±0.55 | **67.59** ±0.46 | **88.27** ±0.27 | **67.46** ±0.26 | **86.97** ±0.21 | **87.64** ±0.16 | **83.46** ±0.40 | **82.45** ±0.19 | **86.72** ±0.19 | **91.22** ±0.21 | **87.79** ±0.98 | **78.26** ±0.35 | **82.61** ±0.38 | **75.79** ±0.32 | **81.23** ±0.39 |
| | CoTTA | 63.01 ±0.12 | 64.38 ±0.64 | 58.95 ±0.74 | 75.43 ±0.61 | 59.65 ±0.48 | 76.08 ±0.58 | 75.47 ±0.16 | 71.75 ±0.55 | 70.33 ±0.48 | 75.52 ±0.32 | 80.94 ±0.49 | 70.53 ±0.51 | 68.75 ±0.65 | 67.87 ±0.30 | 67.55 ±0.37 | 69.75 ±0.47 |
| | **+ SNAP** | **71.39** ±0.31 | **73.57** ±0.27 | **66.29** ±0.10 | **85.22** ±0.22 | **66.71** ±0.19 | **84.20** ±0.18 | **84.64** ±0.13 | **80.77** ±0.21 | **80.56** ±0.32 | **84.06** ±0.15 | **89.85** ±0.07 | **81.86** ±0.08 | **76.48** ±0.24 | **79.94** ±0.27 | **75.69** ±0.48 | **78.75** ±0.19 |
| | EATA | 70.98 ±1.05 | 73.70 ±0.28 | 65.73 ±1.68 | 86.01 ±0.35 | 66.71 ±0.81 | 84.36 ±0.23 | 86.10 ±0.38 | 80.92 ±0.47 | 79.87 ±0.04 | 84.48 ±0.19 | 89.29 ±0.31 | 86.33 ±0.20 | 76.19 ±0.20 | 80.66 ±0.58 | 73.98 ±0.52 | 79.02 ±0.48 |
| 0.3 | **+ SNAP** | **74.19** ±0.38 | **76.64** ±0.68 | **67.89** ±0.19 | **87.93** ±0.25 | **68.56** ±0.20 | **87.08** ±0.05 | **87.89** ±0.34 | **83.56** ±0.30 | **82.20** ±0.25 | **86.60** ±0.23 | **91.11** ±0.22 | **88.94** ±0.61 | **78.10** ±0.14 | **83.03** ±0.20 | **75.83** ±0.43 | **81.30** ±0.30 |
| | SAR | 69.10 ±1.63 | 72.37 ±1.05 | 63.22 ±0.44 | 85.18 ±0.25 | 64.30 ±1.02 | 83.94 ±0.12 | 85.07 ±0.45 | 80.11 ±0.17 | 79.64 ±0.60 | 83.91 ±0.37 | 88.64 ±0.10 | 84.21 ±0.30 | 75.70 ±0.34 | 79.10 ±0.52 | 72.92 ±0.09 | 77.83 ±0.50 |
| | **+ SNAP** | **72.72** ±0.94 | **75.25** ±0.30 | **65.78** ±1.06 | **86.53** ±0.16 | **66.19** ±0.60 | **85.53** ±0.26 | **86.40** ±0.27 | **81.61** ±0.45 | **80.53** ±0.64 | **85.08** ±0.23 | **91.41** ±0.14 | **86.74** ±0.08 | **77.23** ±0.41 | **81.00** ±0.37 | **74.52** ±1.04 | **79.77** ±0.46 |
| | RoTTA | 64.09 ±0.44 | 66.07 ±0.13 | 57.58 ±0.63 | 84.97 ±0.20 | 62.66 ±0.15 | 83.06 ±0.18 | 84.08 ±0.17 | 78.60 ±0.34 | 76.40 ±0.36 | 82.86 ±0.05 | 88.03 ±0.22 | 83.21 ±0.24 | 74.14 ±0.58 | 76.35 ±0.47 | 68.70 ±0.17 | 75.39 ±0.29 |
| | **+ SNAP** | **65.83** ±0.18 | **67.57** ±0.19 | **58.39** ±0.29 | **86.97** ±0.33 | **64.22** ±0.16 | **85.63** ±0.16 | **86.39** ±0.09 | **80.75** ±0.15 | **78.90** ±0.08 | **85.21** ±0.21 | **90.19** ±0.09 | **85.92** ±0.05 | **75.92** ±0.20 | **78.91** ±0.05 | **70.42** ±0.37 | **77.41** ±0.18 |
| | Tent | 67.32 ±0.93 | 69.39 ±0.96 | 60.69 ±0.36 | 85.34 ±0.24 | 63.82 ±0.41 | 83.52 ±0.13 | 84.70 ±0.15 | 79.68 ±0.41 | 77.79 ±0.50 | 83.75 ±0.08 | 88.53 ±0.49 | 83.12 ±0.66 | 75.18 ±0.68 | 77.82 ±0.69 | 71.47 ±0.44 | 76.81 ±0.48 |
| | **+ SNAP** | **70.22** ±0.44 | **71.48** ±0.91 | **63.08** ±0.04 | **87.35** ±0.20 | **65.74** ±0.26 | **85.89** ±0.25 | **86.38** ±0.32 | **81.93** ±0.33 | **80.00** ±0.21 | **85.62** ±0.14 | **90.34** ±0.22 | **87.47** ±0.11 | **76.44** ±0.12 | **79.63** ±0.14 | **72.72** ±0.39 | **78.95** ±0.27 |
| | CoTTA | 59.11 ±0.43 | 60.26 ±0.56 | 56.07 ±0.65 | 72.23 ±0.69 | 56.77 ±0.64 | 73.55 ±0.68 | 72.20 ±0.94 | 68.05 ±0.63 | 66.68 ±0.52 | 72.88 ±0.56 | 77.66 ±1.15 | 65.95 ±1.17 | 65.67 ±0.83 | 64.12 ±0.95 | 65.16 ±0.58 | 66.42 ±0.73 |
| | **+ SNAP** | **71.70** ±0.40 | **73.54** ±0.21 | **66.70** ±0.02 | **85.16** ±0.19 | **66.83** ±0.39 | **84.30** ±0.08 | **84.88** ±0.20 | **81.02** ±0.25 | **80.61** ±0.23 | **84.20** ±0.08 | **89.84** ±0.20 | **81.71** ±0.20 | **76.60** ±0.14 | **79.66** ±0.25 | **75.71** ±0.20 | **78.83** ±0.20 |
| | EATA | 66.65 ±0.43 | 68.96 ±0.47 | 59.73 ±0.15 | 84.93 ±0.27 | 63.26 ±0.36 | 83.10 ±0.24 | 84.53 ±0.24 | 79.28 ±0.44 | 77.46 ±0.42 | 83.48 ±0.13 | 88.12 ±0.09 | 82.46 ±0.24 | 74.49 ±0.20 | 77.48 ±0.69 | 70.43 ±0.25 | 76.29 ±0.30 |
| 0.1 | **+ SNAP** | **69.29** ±0.39 | **70.49** ±0.57 | **61.71** ±0.37 | **87.32** ±0.42 | **65.48** ±0.38 | **85.96** ±0.21 | **86.64** ±0.34 | **81.44** ±0.47 | **79.56** ±0.23 | **85.47** ±0.38 | **90.50** ±0.36 | **86.84** ±0.21 | **76.32** ±0.12 | **79.64** ±0.32 | **72.51** ±0.32 | **78.61** ±0.34 |
| | SAR | 66.11 ±0.59 | 68.18 ±0.83 | 59.15 ±0.72 | 84.91 ±0.45 | 62.87 ±0.27 | 82.33 ±0.60 | 84.27 ±0.13 | 79.23 ±0.32 | 77.58 ±0.43 | 83.21 ±0.18 | 88.29 ±0.09 | 82.60 ±0.57 | 74.65 ±0.46 | 75.92 ±0.77 | 70.79 ±0.40 | 76.01 ±0.45 |
| | **+ SNAP** | **67.76** ±0.22 | **70.68** ±0.14 | **60.82** ±1.08 | **86.78** ±0.26 | **64.73** ±0.43 | **85.29** ±0.10 | **86.22** ±0.11 | **80.82** ±0.23 | **79.30** ±0.48 | **84.95** ±0.28 | **91.33** ±0.17 | **86.59** ±0.14 | **75.72** ±0.26 | **78.72** ±0.35 | **71.24** ±0.46 | **78.06** ±0.31 |
| | RoTTA | 63.12 ±0.33 | 64.84 ±0.21 | 56.72 ±0.30 | 84.49 ±0.04 | 62.15 ±0.17 | 82.53 ±0.30 | 83.84 ±0.02 | 78.03 ±0.29 | 76.13 ±0.71 | 82.88 ±0.16 | 87.48 ±0.08 | 81.49 ±0.11 | 73.75 ±0.14 | 76.04 ±0.29 | 68.24 ±0.27 | 74.78 ±0.23 |
| | **+ SNAP** | **65.35** ±0.20 | **66.99** ±0.15 | **58.09** ±0.18 | **86.77** ±0.18 | **63.63** ±0.18 | **85.47** ±0.13 | **86.01** ±0.21 | **80.54** ±0.11 | **78.38** ±0.24 | **84.99** ±0.43 | **90.00** ±0.23 | **85.99** ±0.03 | **75.67** ±0.17 | **78.14** ±0.06 | **70.09** ±0.23 | **77.07** ±0.18 |

Table 23: STTA classification accuracy (%) comparing with and without SNAP on CIFAR10-C through Adaptation Rates(AR) (0.05, 0.03, and 0.01). **Bold** numbers are the highest accuracy.

| AR | Methods | Gau. | Shot | Imp. | Def. | Gla. | Mot. | Zoom | Snow | Fro. | Fog | Brit. | Cont. | Elas. | Pix. | JPEG | Avg. |
|---|---|---|---|---|---|---|---|---|---|---|---|---|---|---|---|---|---|
| 0.05 | Tent | 64.65 | 67.08 | 58.48 | 85.00 | 62.61 | 82.76 | 84.63 | 79.01 | 77.66 | 83.32 | 88.00 | 82.34 | 74.16 | 77.11 | 69.40 | 75.75 |
| | | ±0.55 | ±0.58 | ±0.42 | ±0.60 | ±0.44 | ±0.70 | ±0.55 | ±0.74 | ±0.91 | ±0.48 | ±0.56 | ±0.93 | ±0.10 | ±0.60 | ±0.48 | ±0.57 |
| | **+ SNAP** | **67.71** | **69.84** | **59.53** | **87.10** | **64.66** | **85.73** | **86.35** | **80.68** | **78.92** | **85.60** | **90.19** | **86.72** | **76.16** | **78.86** | **70.95** | **77.93** |
| | | **±0.38** | **±0.82** | **±1.10** | **±0.15** | **±0.25** | **±0.20** | **±0.20** | **±0.23** | **±0.14** | **±0.08** | **±0.31** | **±0.20** | **±0.17** | **±0.42** | **±0.30** | **±0.33** |
| | CoTTA | 59.27 | 61.18 | 56.33 | 72.22 | 57.37 | 74.27 | 72.61 | 70.03 | 68.68 | 74.82 | 79.72 | 65.57 | 66.92 | 64.13 | 65.25 | 67.22 |
| | | ±0.66 | ±1.12 | ±0.06 | ±1.43 | ±1.10 | ±1.46 | ±1.11 | ±1.02 | ±0.92 | ±1.09 | ±1.07 | ±1.38 | ±1.14 | ±1.27 | ±0.98 | ±1.05 |
| | **+ SNAP** | **71.42** | **73.31** | **65.91** | **85.23** | **67.01** | **84.19** | **84.91** | **80.80** | **80.56** | **84.19** | **90.00** | **82.09** | **76.31** | **79.79** | **75.18** | **78.73** |
| | | **±0.29** | **±0.12** | **±0.13** | **±0.11** | **±0.21** | **±0.20** | **±0.14** | **±0.19** | **±0.34** | **±0.14** | **±0.23** | **±0.35** | **±0.05** | **±0.29** | **±0.21** | **±0.20** |
| | EATA | 64.68 | 67.01 | 58.07 | 84.90 | 62.56 | 82.64 | 84.57 | 78.77 | 77.16 | 83.09 | 87.80 | 81.62 | 74.05 | 76.99 | 69.31 | 75.55 |
| | | ±0.31 | ±0.37 | ±0.24 | ±0.54 | ±0.33 | ±0.67 | ±0.61 | ±0.71 | ±0.92 | ±0.44 | ±0.47 | ±0.59 | ±0.28 | ±0.41 | ±0.71 | ±0.51 |
| | **+ SNAP** | **67.36** | **68.73** | **59.35** | **87.05** | **64.36** | **85.62** | **86.48** | **81.31** | **78.73** | **85.33** | **90.03** | **86.31** | **76.04** | **78.79** | **70.90** | **77.76** |
| | | **±0.33** | **±0.26** | **±0.37** | **±0.22** | **±0.18** | **±0.25** | **±0.24** | **±0.22** | **±0.24** | **±0.22** | **±0.24** | **±0.07** | **±0.12** | **±0.27** | **±0.38** | **±0.23** |
| | SAR | 64.79 | 66.32 | 57.58 | 84.66 | 62.46 | 81.42 | 84.13 | 78.87 | 77.20 | 82.62 | 88.10 | 82.12 | 74.04 | 75.38 | 69.13 | 75.25 |
| | | ±0.13 | ±0.86 | ±0.69 | ±0.72 | ±0.26 | ±1.52 | ±0.34 | ±0.26 | ±0.81 | ±1.24 | ±0.41 | ±0.74 | ±0.05 | ±0.80 | ±0.52 | ±0.62 |
| | **+ SNAP** | **66.00** | **68.85** | **58.47** | **86.54** | **63.06** | **85.26** | **86.13** | **80.38** | **78.17** | **85.17** | **90.93** | **85.96** | **75.27** | **77.37** | **70.61** | **77.21** |
| | | **±0.17** | **±0.75** | **±0.42** | **±0.25** | **±0.28** | **±0.09** | **±0.38** | **±0.09** | **±0.27** | **±0.13** | **±0.36** | **±0.20** | **±0.31** | **±0.28** | **±0.30** | **±0.29** |
| | RoTTA | 63.21 | 64.87 | 56.60 | 84.64 | 62.16 | 82.31 | 84.13 | 78.16 | 76.39 | 82.90 | 87.44 | 81.47 | 73.59 | 76.02 | 68.09 | 74.80 |
| | | ±0.37 | ±0.62 | ±0.28 | ±0.52 | ±0.31 | ±0.63 | ±0.56 | ±0.71 | ±0.95 | ±0.62 | ±0.46 | ±0.65 | ±0.42 | ±0.40 | ±0.33 | ±0.52 |
| | **+ SNAP** | **65.28** | **66.91** | **57.88** | **86.75** | **63.51** | **85.48** | **86.17** | **80.46** | **78.38** | **85.24** | **89.99** | **85.82** | **75.66** | **77.98** | **70.15** | **77.05** |
| | | **±0.32** | **±0.22** | **±0.06** | **±0.25** | **±0.13** | **±0.13** | **±0.10** | **±0.23** | **±0.26** | **±0.13** | **±0.23** | **±0.03** | **±0.16** | **±0.19** | **±0.29** | **±0.18** |
| 0.03 | Tent | 64.36 | 66.21 | 57.65 | 84.73 | 62.95 | 83.07 | 84.50 | 78.46 | 76.99 | 83.00 | 88.07 | 82.62 | 73.93 | 76.50 | 68.82 | 75.46 |
| | | ±0.43 | ±0.16 | ±1.01 | ±0.48 | ±0.52 | ±0.50 | ±0.32 | ±0.82 | ±0.32 | ±0.36 | ±0.43 | ±0.34 | ±0.23 | ±0.46 | ±0.48 | ±0.46 |
| | **+ SNAP** | **66.32** | **68.38** | **59.00** | **86.93** | **64.04** | **85.58** | **86.35** | **80.78** | **78.68** | **85.34** | **90.08** | **86.19** | **75.77** | **78.37** | **70.49** | **77.49** |
| | | **±0.61** | **±0.71** | **±0.52** | **±0.19** | **±0.24** | **±0.34** | **±0.05** | **±0.10** | **±0.02** | **±0.05** | **±0.10** | **±0.31** | **±0.05** | **±0.06** | **±0.08** | **±0.23** |
| | CoTTA | 60.38 | 61.26 | 56.71 | 72.44 | 57.58 | 74.64 | 72.73 | 69.68 | 68.34 | 74.64 | 79.52 | 67.28 | 67.42 | 64.89 | 66.19 | 67.58 |
| | | ±1.71 | ±1.94 | ±2.47 | ±2.23 | ±1.85 | ±1.74 | ±2.61 | ±2.03 | ±2.02 | ±2.52 | ±2.37 | ±1.89 | ±1.77 | ±0.79 | ±1.73 | ±1.98 |
| | **+ SNAP** | **71.12** | **73.68** | **66.34** | **85.30** | **66.64** | **84.25** | **84.55** | **80.88** | **80.11** | **84.06** | **89.89** | **81.98** | **76.27** | **79.77** | **75.35** | **78.68** |
| | | **±0.47** | **±0.29** | **±0.24** | **±0.01** | **±0.12** | **±0.34** | **±0.13** | **±0.15** | **±0.15** | **±0.14** | **±0.14** | **±0.37** | **±0.19** | **±0.26** | **±0.08** | **±0.21** |
| | EATA | 63.99 | 65.95 | 57.39 | 84.71 | 62.66 | 83.11 | 84.44 | 78.42 | 76.63 | 82.97 | 88.00 | 82.55 | 73.85 | 76.46 | 68.91 | 75.34 |
| | | ±0.87 | ±0.44 | ±1.05 | ±0.48 | ±0.62 | ±0.52 | ±0.33 | ±0.75 | ±0.26 | ±0.26 | ±0.47 | ±0.34 | ±0.33 | ±0.29 | ±0.56 | ±0.50 |
| | **+ SNAP** | **66.16** | **67.60** | **58.81** | **86.95** | **64.06** | **85.49** | **86.34** | **80.79** | **78.65** | **85.24** | **90.09** | **86.23** | **75.88** | **78.48** | **70.56** | **77.42** |
| | | **±0.03** | **±0.41** | **±0.36** | **±0.13** | **±0.17** | **±0.36** | **±0.08** | **±0.01** | **±0.25** | **±0.13** | **±0.12** | **±0.08** | **±0.18** | **±0.10** | **±0.47** | **±0.19** |
| | SAR | 63.72 | 65.75 | 57.89 | 84.37 | 62.45 | 81.47 | 82.46 | 78.32 | 76.79 | 81.93 | 88.60 | 82.72 | 73.89 | 74.55 | 68.79 | 74.91 |
| | | ±0.46 | ±0.29 | ±0.65 | ±0.81 | ±0.69 | ±1.61 | ±2.95 | ±0.81 | ±0.24 | ±1.33 | ±0.68 | ±0.29 | ±0.43 | ±0.98 | ±0.61 | ±0.85 |
| | **+ SNAP** | **65.40** | **67.68** | **58.37** | **86.72** | **63.11** | **85.10** | **86.18** | **79.93** | **78.05** | **84.92** | **90.93** | **85.58** | **75.30** | **77.22** | **69.97** | **76.96** |
| | | **±0.33** | **±0.60** | **±0.45** | **±0.18** | **±0.16** | **±0.16** | **±0.29** | **±0.17** | **±0.31** | **±0.22** | **±0.35** | **±0.14** | **±0.14** | **±0.30** | **±0.30** | **±0.27** |
| | RoTTA | 63.36 | 65.10 | 56.64 | 84.62 | 62.41 | 82.96 | 84.35 | 78.10 | 76.42 | 82.69 | 87.90 | 82.34 | 73.56 | 76.09 | 68.39 | 75.00 |
| | | ±0.80 | ±0.55 | ±0.56 | ±0.49 | ±0.79 | ±0.67 | ±0.43 | ±0.80 | ±0.23 | ±0.25 | ±0.53 | ±0.32 | ±0.25 | ±0.44 | ±0.31 | ±0.50 |
| | **+ SNAP** | **65.27** | **67.05** | **58.05** | **86.79** | **63.48** | **85.46** | **86.25** | **80.39** | **78.34** | **85.19** | **90.10** | **85.94** | **75.67** | **78.04** | **69.75** | **77.05** |
| | | **±0.32** | **±0.19** | **±0.22** | **±0.21** | **±0.18** | **±0.33** | **±0.09** | **±0.08** | **±0.15** | **±0.10** | **±0.16** | **±0.08** | **±0.12** | **±0.09** | **±0.27** | **±0.17** |
| 0.01 | Tent | 62.43 | 64.13 | 55.85 | 84.03 | 62.21 | 82.47 | 83.87 | 77.71 | 76.55 | 82.75 | 87.35 | 81.83 | 73.24 | 75.34 | 67.73 | 74.50 |
| | | ±1.70 | ±1.51 | ±1.35 | ±1.07 | ±1.20 | ±0.88 | ±0.93 | ±0.66 | ±0.18 | ±0.14 | ±1.11 | ±1.81 | ±1.33 | ±1.18 | ±1.50 | ±1.10 |
| | **+ SNAP** | **65.51** | **67.26** | **58.05** | **86.89** | **63.53** | **85.44** | **85.97** | **80.58** | **78.35** | **85.12** | **90.09** | **85.86** | **75.66** | **78.38** | **70.12** | **77.12** |
| | | **±0.24** | **±0.31** | **±0.34** | **±0.28** | **±0.07** | **±0.33** | **±0.20** | **±0.12** | **±0.12** | **±0.16** | **±0.21** | **±0.11** | **±0.08** | **±0.21** | **±0.33** | **±0.21** |
| | CoTTA | 59.75 | 59.44 | 54.47 | 71.12 | 57.11 | 72.47 | 72.83 | 66.05 | 65.14 | 69.75 | 75.12 | 64.31 | 66.22 | 62.65 | 64.76 | 65.41 |
| | | ±4.69 | ±6.21 | ±5.57 | ±5.10 | ±4.35 | ±4.52 | ±4.80 | ±7.60 | ±7.65 | ±9.79 | ±6.79 | ±6.46 | ±4.50 | ±5.27 | ±5.36 | ±5.91 |
| | **+ SNAP** | **71.79** | **73.61** | **65.98** | **85.34** | **66.76** | **84.26** | **84.93** | **80.64** | **80.38** | **83.94** | **89.98** | **82.47** | **76.48** | **79.61** | **75.60** | **78.79** |
| | | **±0.22** | **±0.29** | **±0.58** | **±0.36** | **±0.26** | **±0.12** | **±0.21** | **±0.45** | **±0.30** | **±0.42** | **±0.08** | **±0.64** | **±0.26** | **±0.24** | **±0.29** | **±0.31** |
| | EATA | 62.36 | 63.92 | 55.73 | 84.05 | 62.24 | 82.38 | 83.90 | 77.66 | 76.48 | 82.67 | 87.34 | 81.82 | 73.30 | 75.31 | 67.76 | 74.46 |
| | | ±1.73 | ±1.66 | ±1.39 | ±1.10 | ±1.18 | ±0.85 | ±0.93 | ±0.72 | ±0.15 | ±0.17 | ±1.12 | ±1.81 | ±1.24 | ±1.20 | ±1.52 | ±1.12 |
| | **+ SNAP** | **65.49** | **67.19** | **57.93** | **86.92** | **63.65** | **85.42** | **85.97** | **80.46** | **78.13** | **85.07** | **90.03** | **85.87** | **75.69** | **78.20** | **70.03** | **77.07** |
| | | **±0.29** | **±0.04** | **±0.40** | **±0.41** | **±0.18** | **±0.28** | **±0.10** | **±0.24** | **±0.18** | **±0.27** | **±0.13** | **±0.20** | **±0.11** | **±0.13** | **±0.46** | **±0.23** |
| | SAR | 62.50 | 64.13 | 55.65 | 82.30 | 62.22 | 77.21 | 80.11 | 77.66 | 76.75 | 79.12 | 89.45 | 81.97 | 73.39 | 69.39 | 67.83 | 73.31 |
| | | ±1.69 | ±1.83 | ±1.38 | ±3.37 | ±1.21 | ±6.27 | ±6.19 | ±0.80 | ±0.34 | ±3.28 | ±1.79 | ±1.97 | ±1.21 | ±5.48 | ±1.65 | ±2.57 |
| | **+ SNAP** | **65.06** | **66.93** | **57.66** | **86.76** | **62.78** | **85.05** | **85.94** | **79.95** | **77.62** | **84.65** | **90.72** | **85.48** | **75.34** | **75.72** | **69.61** | **76.62** |
| | | **±0.17** | **±0.11** | **±0.51** | **±0.29** | **±0.24** | **±0.21** | **±0.48** | **±0.18** | **±0.37** | **±0.21** | **±0.62** | **±0.35** | **±0.13** | **±0.25** | **±0.25** | **±0.36** |
| | RoTTA | 62.25 | 63.71 | 55.59 | 84.05 | 62.17 | 82.32 | 83.86 | 77.56 | 76.39 | 82.64 | 87.27 | 81.75 | 73.21 | 75.15 | 67.75 | 74.38 |
| | | ±1.65 | ±1.68 | ±1.46 | ±1.12 | ±1.37 | ±0.83 | ±0.90 | ±0.75 | ±0.24 | ±0.10 | ±1.12 | ±1.82 | ±1.21 | ±1.27 | ±1.48 | ±1.13 |
| | **+ SNAP** | **65.32** | **66.94** | **57.85** | **86.91** | **63.44** | **85.32** | **85.98** | **80.49** | **78.22** | **85.04** | **90.01** | **85.77** | **75.75** | **78.15** | **70.06** | **77.02** |
| | | **±0.25** | **±0.12** | **±0.29** | **±0.31** | **±0.24** | **±0.22** | **±0.14** | **±0.24** | **±0.20** | **±0.15** | **±0.06** | **±0.24** | **±0.11** | **±0.07** | **±0.47** | **±0.21** |

## C.2 CIFAR100-C

Table 24: STTA classification accuracy (%) comparing with and without SNAP on CIAFR100-C through Adaptation Rates(AR) (0.5, 0.3, and 0.1), including results for full adaptation (AR=1). **Bold** numbers are the highest accuracy.

| AR | Methods | Gau. | Shot | Imp. | Def. | Gla. | Mot. | Zoom | Snow | Fro. | Fog | Brit. | Cont. | Elas. | Pix. | JPEG | Avg. |
|---|---|---|---|---|---|---|---|---|---|---|---|---|---|---|---|---|---|
| 1 | Source | 10.26 ±0.00 | 11.87 ±0.00 | 6.48 ±0.00 | 35.16 ±0.00 | 20.33 ±0.00 | 44.42 ±0.00 | 42.13 ±0.00 | 45.99 ±0.00 | 34.84 ±0.00 | 41.12 ±0.00 | 66.37 ±0.00 | 19.54 ±0.00 | 50.59 ±0.00 | 22.68 ±0.00 | 45.48 ±0.00 | 33.15 ±0.00 |
| | BN stats | 36.90 ±0.10 | 37.96 ±0.24 | 32.13 ±0.44 | 62.65 ±0.26 | 39.14 ±0.19 | 60.05 ±0.42 | 61.16 ±0.05 | 50.68 ±0.13 | 50.38 ±0.09 | 54.81 ±0.24 | 64.40 ±0.05 | 60.33 ±0.12 | 50.48 ±0.24 | 53.49 ±0.11 | 41.98 ±0.49 | 50.44 ±0.21 |
| | Tent | 46.71 ±0.29 | 48.06 ±0.47 | 40.98 ±0.13 | 65.19 ±0.40 | 44.10 ±0.41 | 62.78 ±0.24 | 63.95 ±0.23 | 55.43 ±0.36 | 55.46 ±0.49 | 59.32 ±0.30 | 67.43 ±0.17 | 63.83 ±0.42 | 53.89 ±0.15 | 59.40 ±0.32 | 49.91 ±0.66 | 55.76 ±0.33 |
| | CoTTA | 42.14 ±0.34 | 42.92 ±0.44 | 37.92 ±0.18 | 55.40 ±0.12 | 41.01 ±0.39 | 55.18 ±0.10 | 55.39 ±0.58 | 49.46 ±0.23 | 50.61 ±0.63 | 50.86 ±0.31 | 61.35 ±0.27 | 47.44 ±0.37 | 48.69 ±0.18 | 54.38 ±0.16 | 48.11 ±0.65 | 49.39 ±0.33 |
| | EATA | 38.42 ±0.41 | 39.96 ±0.47 | 32.64 ±0.71 | 62.35 ±0.41 | 38.73 ±0.33 | 59.93 ±0.17 | 61.07 ±0.19 | 50.50 ±0.36 | 50.79 ±0.34 | 55.30 ±0.23 | 64.38 ±0.12 | 60.63 ±0.13 | 49.66 ±0.32 | 53.63 ±0.41 | 43.02 ±0.20 | 50.74 ±0.32 |
| | SAR | 50.75 ±0.44 | 52.00 ±0.22 | 43.87 ±0.40 | 65.44 ±0.39 | 46.30 ±0.22 | 63.60 ±0.15 | 64.68 ±0.48 | 58.41 ±0.09 | 58.26 ±0.40 | 61.34 ±0.15 | 68.03 ±0.31 | 67.68 ±0.25 | 54.53 ±0.21 | 61.52 ±0.23 | 52.72 ±0.27 | 57.94 ±0.27 |
| | RoTTA | 38.54 ±0.22 | 39.85 ±0.24 | 33.73 ±0.37 | 63.45 ±0.17 | 40.74 ±0.32 | 60.54 ±0.19 | 62.03 ±0.26 | 51.61 ±0.09 | 51.75 ±0.14 | 56.20 ±0.31 | 65.14 ±0.10 | 61.55 ±0.14 | 51.22 ±0.14 | 54.42 ±0.22 | 42.50 ±0.35 | 51.55 ±0.22 |
| 0.5 | Tent | 43.96 ±0.85 | 45.42 ±1.34 | 36.57 ±1.57 | 62.28 ±0.13 | 36.57 ±2.97 | 59.96 ±0.59 | 61.90 ±0.48 | 53.25 ±0.72 | 53.14 ±1.70 | 57.36 ±0.22 | 65.20 ±0.20 | 60.14 ±2.77 | 49.72 ±0.08 | 57.62 ±0.61 | 46.83 ±0.52 | 52.66 ±0.98 |
| | + SNAP | **49.06** ±0.00 | **50.43** ±0.13 | **41.49** ±0.80 | **65.55** ±0.24 | **44.09** ±0.06 | **63.31** ±0.53 | **65.62** ±0.37 | **57.62** ±0.09 | **56.81** ±0.31 | **60.75** ±0.48 | **68.72** ±0.31 | **67.52** ±0.64 | **54.08** ±0.19 | **61.15** ±0.14 | **51.54** ±0.11 | **57.18** ±0.29 |
| | CoTTA | 34.31 ±0.09 | 35.16 ±0.46 | 31.42 ±0.28 | 47.78 ±0.45 | 34.99 ±0.40 | 48.91 ±0.48 | 47.79 ±0.46 | 41.27 ±0.37 | 41.42 ±0.57 | 43.77 ±0.86 | 52.16 ±0.27 | 38.30 ±0.46 | 42.25 ±0.49 | 44.12 ±0.41 | 41.58 ±0.22 | 41.68 ±0.42 |
| | + SNAP | **41.28** ±0.46 | **42.23** ±0.16 | **37.17** ±0.19 | **58.29** ±0.21 | **40.70** ±0.08 | **57.32** ±0.12 | **57.78** ±0.09 | **49.85** ±0.38 | **50.82** ±0.11 | **52.21** ±0.28 | **63.69** ±0.18 | **51.30** ±0.23 | **49.41** ±0.14 | **55.15** ±0.09 | **47.92** ±0.25 | **50.34** ±0.20 |
| | EATA | 38.02 ±0.22 | 39.48 ±0.15 | 32.77 ±0.17 | 61.68 ±0.38 | 38.42 ±0.07 | 59.11 ±0.09 | 60.63 ±0.18 | 50.15 ±0.67 | 49.92 ±0.13 | 54.60 ±0.21 | 63.43 ±0.44 | 58.70 ±0.22 | 49.42 ±0.20 | 53.08 ±0.21 | 42.62 ±0.24 | 50.13 ±0.24 |
| | + SNAP | **39.75** ±0.11 | **41.14** ±0.26 | **34.15** ±0.10 | **63.75** ±0.23 | **40.55** ±0.21 | **61.09** ±0.08 | **62.81** ±0.19 | **52.12** ±0.08 | **52.12** ±0.30 | **56.47** ±0.18 | **65.73** ±0.23 | **61.85** ±0.34 | **51.14** ±0.28 | **55.75** ±0.15 | **44.86** ±0.51 | **52.22** ±0.22 |
| | SAR | 49.00 ±0.61 | 50.00 ±0.42 | 42.99 ±0.30 | 65.10 ±0.44 | 45.21 ±0.41 | 62.51 ±0.20 | 64.43 ±0.43 | 55.78 ±0.27 | 56.59 ±0.46 | 60.21 ±0.48 | 67.33 ±0.44 | 65.17 ±0.46 | 53.90 ±0.50 | 60.22 ±0.29 | 51.28 ±0.23 | 56.65 ±0.40 |
| | + SNAP | **51.71** ±0.46 | **52.79** ±0.08 | **44.95** ±0.54 | **66.59** ±0.10 | **47.84** ±0.01 | **64.40** ±0.18 | **66.15** ±0.28 | **59.02** ±0.20 | **59.12** ±0.37 | **62.62** ±0.16 | **69.15** ±0.06 | **68.20** ±0.16 | **55.89** ±0.26 | **62.66** ±0.31 | **53.77** ±0.23 | **58.99** ±0.23 |
| | RoTTA | 37.12 ±0.09 | 38.34 ±0.20 | 32.54 ±0.22 | 62.25 ±0.09 | 38.91 ±0.13 | 59.52 ±0.19 | 61.19 ±0.21 | 50.22 ±0.56 | 49.91 ±0.15 | 54.69 ±0.10 | 63.74 ±0.47 | 59.40 ±0.29 | 50.32 ±0.29 | 53.29 ±0.15 | 41.94 ±0.23 | 50.22 ±0.23 |
| | + SNAP | **38.33** ±0.30 | **39.12** ±0.24 | **32.93** ±0.28 | **64.01** ±0.15 | **40.36** ±0.44 | **61.30** ±0.38 | **62.96** ±0.28 | **51.77** ±0.22 | **51.54** ±0.19 | **56.15** ±0.28 | **66.13** ±0.17 | **61.67** ±0.24 | **51.60** ±0.23 | **54.90** ±0.36 | **43.14** ±0.36 | **51.73** ±0.25 |
| 0.3 | Tent | 44.41 ±0.80 | 46.79 ±0.72 | 38.72 ±1.17 | 62.98 ±0.28 | 39.79 ±0.92 | 60.38 ±0.53 | 62.25 ±0.33 | 52.47 ±0.76 | 53.69 ±0.65 | 57.47 ±0.63 | 65.80 ±0.28 | 60.13 ±2.70 | 50.03 ±0.60 | 58.21 ±0.81 | 47.23 ±0.43 | 53.36 ±0.77 |
| | + SNAP | **49.23** ±0.04 | **50.15** ±0.48 | **42.19** ±0.75 | **65.85** ±0.15 | **45.12** ±1.15 | **63.39** ±0.28 | **64.91** ±0.26 | **57.45** ±0.51 | **57.13** ±0.37 | **60.72** ±0.17 | **68.86** ±0.31 | **66.65** ±1.52 | **54.25** ±0.41 | **61.38** ±0.54 | **51.80** ±0.68 | **57.27** ±0.51 |
| | CoTTA | 31.74 ±0.43 | 32.66 ±0.38 | 29.28 ±0.15 | 44.98 ±0.45 | 32.96 ±0.56 | 46.51 ±0.48 | 44.96 ±0.37 | 38.57 ±0.90 | 38.16 ±0.78 | 41.91 ±0.39 | 49.38 ±0.86 | 35.53 ±0.33 | 40.04 ±0.61 | 40.77 ±0.67 | 39.12 ±0.43 | 39.11 ±0.52 |
| | + SNAP | **41.44** ±0.38 | **42.49** ±0.09 | **37.08** ±0.13 | **58.27** ±0.24 | **40.99** ±0.37 | **57.24** ±0.37 | **57.68** ±0.17 | **50.36** ±0.22 | **51.09** ±0.18 | **51.66** ±0.52 | **63.50** ±0.26 | **50.90** ±0.42 | **49.49** ±0.13 | **54.75** ±0.26 | **47.81** ±0.42 | **50.32** ±0.26 |
| | EATA | 37.97 ±0.04 | 39.47 ±0.34 | 32.69 ±0.12 | 61.45 ±0.19 | 37.96 ±0.17 | 59.02 ±0.28 | 60.79 ±0.12 | 49.73 ±0.05 | 49.55 ±0.38 | 54.63 ±0.07 | 63.38 ±0.21 | 58.16 ±0.24 | 49.07 ±0.24 | 53.17 ±0.41 | 42.49 ±0.44 | 49.97 ±0.23 |
| | + SNAP | **40.03** ±0.26 | **41.39** ±0.29 | **34.91** ±0.58 | **63.58** ±0.15 | **40.29** ±0.28 | **61.58** ±0.12 | **62.56** ±0.25 | **51.85** ±0.25 | **51.78** ±0.21 | **56.13** ±0.01 | **65.70** ±0.22 | **61.68** ±0.29 | **51.25** ±0.35 | **55.28** ±0.23 | **44.80** ±0.17 | **52.19** ±0.24 |
| | SAR | 49.00 ±0.61 | 50.00 ±0.42 | 42.99 ±0.30 | 65.10 ±0.44 | 45.21 ±0.41 | 62.51 ±0.20 | 64.43 ±0.43 | 55.78 ±0.27 | 56.59 ±0.46 | 60.21 ±0.48 | 67.33 ±0.44 | 65.17 ±0.46 | 53.90 ±0.50 | 60.22 ±0.29 | 51.28 ±0.23 | 56.65 ±0.40 |
| | + SNAP | **50.63** ±0.31 | **52.03** ±0.32 | **44.89** ±0.54 | **66.28** ±0.13 | **47.08** ±0.26 | **64.32** ±0.09 | **65.90** ±0.21 | **57.98** ±0.27 | **58.09** ±0.49 | **61.88** ±0.24 | **69.17** ±0.42 | **67.82** ±0.29 | **55.47** ±0.29 | **62.02** ±0.31 | **53.09** ±0.15 | **58.44** ±0.29 |
| | RoTTA | 36.83 ±0.18 | 37.94 ±0.22 | 32.00 ±0.05 | 61.90 ±0.20 | 38.67 ±0.10 | 59.15 ±0.14 | 60.97 ±0.24 | 49.92 ±0.23 | 49.32 ±0.38 | 54.62 ±0.21 | 63.71 ±0.18 | 58.31 ±0.11 | 49.79 ±0.22 | 52.88 ±0.34 | 41.59 ±0.27 | 49.84 ±0.21 |
| | + SNAP | **38.11** ±0.13 | **39.21** ±0.23 | **32.80** ±0.14 | **63.72** ±0.13 | **40.01** ±0.23 | **61.51** ±0.13 | **62.74** ±0.16 | **51.37** ±0.15 | **51.49** ±0.30 | **55.68** ±0.25 | **65.90** ±0.13 | **61.56** ±0.29 | **51.50** ±0.08 | **54.67** ±0.19 | **43.01** ±0.19 | **51.55** ±0.18 |
| 0.1 | Tent | 43.55 ±0.66 | 44.25 ±0.54 | 37.95 ±0.72 | 62.56 ±0.47 | 41.80 ±0.04 | 59.45 ±0.20 | 62.13 ±0.21 | 53.04 ±0.84 | 51.60 ±0.39 | 56.76 ±0.15 | 64.60 ±0.56 | 61.19 ±1.68 | 51.01 ±0.39 | 56.42 ±0.27 | 46.28 ±0.49 | 52.84 ±0.51 |
| | + SNAP | **46.51** ±0.35 | **47.68** ±0.23 | **39.92** ±0.48 | **65.39** ±0.11 | **44.14** ±0.60 | **63.29** ±0.18 | **64.53** ±0.38 | **55.20** ±0.47 | **55.55** ±0.11 | **59.71** ±0.33 | **68.05** ±0.17 | **64.90** ±0.90 | **53.91** ±0.30 | **59.28** ±0.16 | **49.58** ±0.75 | **55.84** ±0.37 |
| | CoTTA | 28.53 ±0.90 | 29.53 ±0.86 | 26.45 ±0.60 | 42.19 ±1.19 | 30.34 ±0.77 | 44.69 ±1.07 | 41.88 ±0.62 | 34.44 ±0.84 | 33.93 ±1.07 | 39.03 ±0.89 | 49.34 ±1.36 | 31.17 ±0.60 | 37.25 ±0.80 | 36.17 ±1.20 | 36.84 ±0.71 | 35.86 ±0.90 |
| | + SNAP | **41.72** ±0.25 | **42.62** ±0.60 | **37.46** ±0.13 | **58.43** ±0.13 | **41.24** ±0.21 | **57.33** ±0.07 | **57.96** ±0.30 | **50.34** ±0.38 | **51.17** ±0.18 | **52.29** ±0.16 | **63.59** ±0.20 | **51.32** ±0.36 | **49.68** ±0.21 | **54.78** ±0.28 | **47.89** ±0.35 | **50.52** ±0.25 |
| | EATA | 38.41 ±0.53 | 39.03 ±0.45 | 32.29 ±0.32 | 61.07 ±0.36 | 38.45 ±0.29 | 58.21 ±0.47 | 60.62 ±0.30 | 49.59 ±0.34 | 49.19 ±0.28 | 54.23 ±0.30 | 62.88 ±0.50 | 57.39 ±0.62 | 49.00 ±0.65 | 53.01 ±0.60 | 42.05 ±0.15 | 49.70 ±0.42 |
| | + SNAP | **40.62** ±0.26 | **41.53** ±0.49 | **34.31** ±0.24 | **64.08** ±0.30 | **40.29** ±0.21 | **61.32** ±0.24 | **63.04** ±0.16 | **52.00** ±0.53 | **51.77** ±0.40 | **56.85** ±0.43 | **65.98** ±0.09 | **61.96** ±0.34 | **51.05** ±0.09 | **55.67** ±0.28 | **44.80** ±0.15 | **52.35** ±0.28 |
| | SAR | 43.92 ±0.52 | 45.28 ±0.55 | 38.64 ±0.28 | 63.36 ±0.25 | 42.58 ±0.44 | 60.36 ±0.42 | 62.78 ±0.23 | 53.39 ±0.86 | 52.23 ±0.28 | 57.54 ±0.32 | 65.41 ±0.41 | 60.88 ±0.88 | 52.07 ±0.59 | 56.80 ±0.13 | 47.16 ±0.20 | 53.49 ±0.43 |
| | + SNAP | **46.29** ±0.68 | **47.60** ±0.06 | **39.95** ±0.21 | **65.26** ±0.18 | **44.00** ±0.22 | **63.09** ±0.25 | **64.97** ±0.36 | **55.08** ±0.24 | **55.17** ±0.17 | **59.73** ±0.24 | **68.13** ±0.09 | **64.72** ±0.44 | **53.84** ±0.31 | **58.98** ±0.35 | **49.54** ±0.65 | **55.76** ±0.30 |
| | RoTTA | 36.28 ±0.15 | 37.12 ±0.41 | 31.38 ±0.27 | 61.20 ±0.07 | 38.36 ±0.15 | 58.26 ±0.24 | 60.30 ±0.47 | 49.20 ±0.23 | 48.21 ±0.14 | 53.54 ±0.23 | 62.80 ±0.40 | 56.74 ±0.51 | 49.61 ±0.24 | 52.28 ±0.41 | 41.26 ±0.11 | 49.11 ±0.27 |
| | + SNAP | **37.83** ±0.13 | **38.42** ±0.36 | **32.38** ±0.20 | **63.73** ±0.09 | **39.72** ±0.38 | **61.32** ±0.18 | **62.58** ±0.19 | **51.38** ±0.18 | **51.18** ±0.13 | **55.61** ±0.07 | **65.70** ±0.29 | **61.39** ±0.21 | **51.36** ±0.09 | **54.51** ±0.24 | **42.85** ±0.33 | **51.33** ±0.21 |

Table 25: STTA classification accuracy (%) comparing with and without SNAP on CIFAR100-C through Adaptation Rates(AR) (0.05, 0.03, and 0.01). **Bold** numbers are the highest accuracy.

| AR | Methods | Gau. | Shot | Imp. | Def. | Gla. | Mot. | Zoom | Snow | Fro. | Fog | Brit. | Cont. | Elas. | Pix. | JPEG | Avg. |
|---|---|---|---|---|---|---|---|---|---|---|---|---|---|---|---|---|---|
| 0.05 | Tent | 40.69 ±0.35 | 41.55 ±0.62 | 35.14 ±0.38 | 62.26 ±0.52 | 40.26 ±0.23 | 58.92 ±0.60 | 61.06 ±0.43 | 51.21 ±0.88 | 50.00 ±0.31 | 55.52 ±0.33 | 64.05 ±0.62 | 58.45 ±1.06 | 50.50 ±0.80 | 54.68 ±0.26 | 44.36 ±0.69 | 51.24 ±0.54 |
|  | **+ SNAP** | **42.87 ±0.37** | **44.87 ±0.70** | **37.60 ±0.08** | **65.01 ±0.01** | **42.22 ±0.35** | **62.22 ±0.31** | **63.72 ±0.45** | **54.03 ±0.46** | **53.68 ±0.39** | **58.03 ±0.47** | **67.05 ±0.50** | **63.08 ±0.10** | **52.97 ±0.15** | **57.67 ±0.12** | **46.94 ±0.13** | **54.13 ±0.31** |
|  | CoTTA | 26.15 ±0.60 | 26.89 ±0.32 | 25.26 ±0.44 | 39.48 ±0.71 | 28.34 ±0.74 | 41.41 ±0.76 | 38.77 ±1.14 | 32.06 ±0.85 | 30.84 ±0.65 | 35.56 ±1.12 | 41.60 ±1.36 | 28.52 ±0.79 | 34.99 ±0.45 | 33.60 ±0.82 | 34.54 ±0.54 | 33.20 ±0.75 |
|  | **+ SNAP** | **42.02 ±0.21** | **42.70 ±0.13** | **37.67 ±0.31** | **58.30 ±0.26** | **41.57 ±0.37** | **57.47 ±0.14** | **58.02 ±0.18** | **50.55 ±0.27** | **51.31 ±0.32** | **52.34 ±0.17** | **63.63 ±0.16** | **51.25 ±0.49** | **49.76 ±0.18** | **54.94 ±0.05** | **47.98 ±0.12** | **50.63 ±0.22** |
|  | EATA | 38.46 ±0.14 | 39.05 ±0.58 | 33.47 ±0.23 | 61.07 ±0.63 | 38.52 ±0.29 | 58.16 ±0.46 | 60.59 ±0.48 | 49.60 ±0.55 | 49.18 ±0.47 | 54.41 ±0.24 | 63.15 ±0.43 | 57.06 ±1.37 | 49.09 ±0.88 | 52.87 ±0.42 | 42.49 ±0.34 | 49.81 ±0.50 |
|  | **+ SNAP** | **40.49 ±0.21** | **41.64 ±0.43** | **34.37 ±0.15** | **64.28 ±0.20** | **40.38 ±0.51** | **61.52 ±0.30** | **63.17 ±0.18** | **51.66 ±0.53** | **52.12 ±0.52** | **56.50 ±0.21** | **66.03 ±0.36** | **62.01 ±0.12** | **51.76 ±0.12** | **55.66 ±0.23** | **44.83 ±0.32** | **52.43 ±0.29** |
|  | SAR | 40.28 ±0.07 | 41.62 ±0.62 | 35.35 ±0.04 | 62.84 ±0.26 | 40.37 ±0.41 | 59.51 ±0.38 | 61.68 ±0.28 | 51.29 ±0.81 | 50.66 ±0.38 | 55.60 ±0.40 | 64.43 ±0.62 | 58.49 ±0.82 | 50.90 ±0.64 | 54.82 ±0.27 | 44.64 ±0.43 | 51.50 ±0.43 |
|  | **+ SNAP** | **41.76 ±0.29** | **44.24 ±0.44** | **36.89 ±0.21** | **64.34 ±0.38** | **41.54 ±0.37** | **62.13 ±0.15** | **63.39 ±0.24** | **53.24 ±0.33** | **52.91 ±0.02** | **57.54 ±0.22** | **66.89 ±0.60** | **62.41 ±0.50** | **52.70 ±0.15** | **57.23 ±0.47** | **46.63 ±0.57** | **53.59 ±0.33** |
|  | RoTTA | 36.38 ±0.12 | 37.38 ±0.42 | 31.78 ±0.45 | 61.44 ±0.06 | 38.26 ±0.20 | 58.18 ±0.42 | 60.19 ±0.53 | 48.98 ±0.18 | 48.30 ±0.28 | 53.50 ±0.17 | 62.73 ±0.42 | 56.52 ±0.90 | 49.37 ±0.49 | 52.19 ±0.19 | 41.60 ±0.28 | 49.12 ±0.34 |
|  | **+ SNAP** | **37.67 ±0.12** | **38.66 ±0.21** | **32.47 ±0.12** | **63.95 ±0.16** | **40.18 ±0.20** | **61.33 ±0.47** | **62.52 ±0.35** | **51.47 ±0.14** | **51.32 ±0.36** | **55.67 ±0.21** | **65.89 ±0.24** | **61.24 ±0.15** | **51.47 ±0.14** | **54.52 ±0.15** | **42.84 ±0.38** | **51.41 ±0.23** |
| 0.03 | Tent | 38.55 ±0.17 | 39.28 ±0.15 | 33.77 ±0.16 | 61.64 ±0.25 | 39.66 ±0.39 | 58.83 ±0.48 | 60.89 ±0.29 | 49.45 ±0.51 | 49.51 ±0.78 | 54.64 ±0.42 | 63.48 ±0.58 | 57.29 ±0.33 | 50.34 ±0.34 | 53.44 ±0.38 | 43.28 ±0.26 | 50.27 ±0.37 |
|  | **+ SNAP** | **41.22 ±0.33** | **42.20 ±0.27** | **35.31 ±0.36** | **64.48 ±0.06** | **40.82 ±0.60** | **61.96 ±0.02** | **63.50 ±0.30** | **52.84 ±0.40** | **52.36 ±0.40** | **57.18 ±0.33** | **66.50 ±0.02** | **62.17 ±0.41** | **52.12 ±0.17** | **56.48 ±0.18** | **45.72 ±0.40** | **52.99 ±0.28** |
|  | CoTTA | 27.11 ±1.11 | 27.73 ±2.05 | 25.87 ±1.41 | 40.25 ±2.62 | 29.52 ±1.49 | 42.16 ±2.21 | 39.60 ±2.51 | 32.74 ±2.42 | 32.23 ±1.71 | 36.60 ±2.75 | 43.33 ±2.80 | 29.13 ±2.42 | 36.45 ±1.82 | 34.51 ±1.66 | 35.96 ±1.75 | 34.21 ±2.05 |
|  | **+ SNAP** | **41.77 ±0.24** | **42.85 ±0.19** | **37.50 ±0.08** | **58.61 ±0.22** | **41.15 ±0.16** | **57.65 ±0.22** | **58.05 ±0.32** | **50.45 ±0.65** | **51.34 ±0.20** | **52.72 ±0.35** | **63.49 ±0.07** | **51.63 ±0.61** | **49.87 ±0.17** | **55.24 ±0.13** | **48.14 ±0.36** | **50.70 ±0.26** |
|  | EATA | 37.94 ±0.32 | 38.63 ±0.21 | 32.00 ±0.91 | 61.02 ±0.33 | 39.08 ±0.30 | 58.52 ±0.66 | 60.28 ±0.42 | 48.73 ±0.32 | 49.15 ±0.97 | 53.89 ±0.53 | 63.03 ±0.34 | 56.64 ±0.49 | 49.45 ±0.47 | 52.93 ±0.35 | 42.11 ±0.44 | 49.56 ±0.47 |
|  | **+ SNAP** | **39.87 ±0.89** | **41.12 ±0.20** | **34.48 ±0.08** | **64.14 ±0.23** | **40.27 ±0.09** | **61.91 ±0.00** | **63.09 ±0.43** | **52.37 ±0.42** | **51.93 ±0.44** | **56.36 ±0.26** | **66.02 ±0.05** | **61.88 ±0.04** | **51.83 ±0.11** | **55.60 ±0.45** | **44.59 ±0.26** | **52.36 ±0.26** |
|  | SAR | 38.33 ±0.25 | 39.19 ±0.26 | 33.15 ±0.43 | 61.77 ±0.21 | 39.78 ±0.06 | 59.09 ±0.33 | 61.02 ±0.25 | 49.67 ±0.54 | 49.86 ±0.65 | 54.71 ±0.31 | 63.59 ±0.49 | 57.45 ±0.18 | 50.37 ±0.39 | 53.67 ±0.32 | 42.88 ±0.51 | 50.30 ±0.35 |
|  | **+ SNAP** | **39.84 ±0.07** | **41.83 ±0.78** | **34.94 ±0.28** | **63.70 ±0.26** | **40.49 ±0.16** | **61.45 ±0.28** | **63.17 ±0.07** | **52.27 ±0.51** | **51.91 ±0.17** | **56.69 ±0.25** | **65.91 ±0.27** | **61.31 ±0.52** | **51.68 ±0.22** | **56.06 ±0.18** | **44.95 ±0.16** | **52.41 ±0.28** |
|  | RoTTA | 36.24 ±0.03 | 36.94 ±0.21 | 31.15 ±0.09 | 60.87 ±0.17 | 38.28 ±0.14 | 58.25 ±0.53 | 59.88 ±0.36 | 48.43 ±0.52 | 48.17 ±0.61 | 53.32 ±0.47 | 62.73 ±0.46 | 56.18 ±0.34 | 49.23 ±0.39 | 52.12 ±0.31 | 41.28 ±0.61 | 48.87 ±0.35 |
|  | **+ SNAP** | **37.85 ±0.20** | **38.68 ±0.20** | **32.78 ±0.31** | **63.97 ±0.24** | **39.75 ±0.17** | **61.41 ±0.16** | **62.57 ±0.52** | **51.53 ±0.27** | **51.38 ±0.28** | **55.68 ±0.37** | **65.56 ±0.20** | **61.25 ±0.13** | **51.53 ±0.19** | **54.84 ±0.26** | **42.96 ±0.33** | **51.45 ±0.25** |
| 0.01 | Tent | 36.08 ±0.42 | 36.95 ±0.21 | 31.31 ±0.47 | 61.03 ±0.51 | 38.09 ±0.56 | 57.63 ±0.53 | 58.76 ±0.31 | 48.24 ±0.47 | 48.65 ±0.87 | 53.45 ±0.19 | 62.14 ±0.49 | 55.07 ±2.13 | 48.59 ±0.25 | 51.82 ±0.58 | 40.68 ±0.04 | 48.57 ±0.54 |
|  | **+ SNAP** | **38.40 ±0.06** | **39.40 ±0.16** | **33.26 ±0.10** | **63.85 ±0.11** | **40.36 ±0.36** | **61.23 ±0.34** | **62.79 ±0.24** | **51.92 ±0.06** | **51.73 ±0.00** | **56.20 ±0.17** | **65.83 ±0.29** | **60.95 ±0.00** | **51.82 ±0.30** | **54.75 ±0.16** | **43.53 ±0.16** | **51.73 ±0.18** |
|  | CoTTA | 26.59 ±1.64 | 27.92 ±1.79 | 24.86 ±1.51 | 41.34 ±2.21 | 28.91 ±1.96 | 43.09 ±2.85 | 40.11 ±2.87 | 34.33 ±1.61 | 33.32 ±2.67 | 37.99 ±2.03 | 44.78 ±3.61 | 28.80 ±2.18 | 36.26 ±1.90 | 34.70 ±1.66 | 35.67 ±1.47 | 34.58 ±2.13 |
|  | **+ SNAP** | **42.05 ±0.05** | **42.91 ±0.17** | **37.50 ±0.08** | **58.70 ±0.12** | **41.22 ±0.36** | **57.38 ±0.17** | **58.14 ±0.33** | **50.39 ±0.68** | **51.13 ±0.43** | **52.23 ±0.12** | **63.42 ±0.35** | **51.74 ±0.17** | **49.87 ±0.50** | **54.84 ±0.09** | **47.72 ±0.25** | **50.62 ±0.26** |
|  | EATA | 36.10 ±0.27 | 37.05 ±0.59 | 31.03 ±0.34 | 60.86 ±0.50 | 37.83 ±0.37 | 57.64 ±0.57 | 58.77 ±0.32 | 48.02 ±0.50 | 48.75 ±1.26 | 53.37 ±0.09 | 62.18 ±0.43 | 54.95 ±2.22 | 48.55 ±0.15 | 51.89 ±0.65 | 40.75 ±0.02 | 48.51 ±0.55 |
|  | **+ SNAP** | **38.54 ±0.14** | **39.78 ±0.15** | **33.11 ±0.22** | **63.82 ±0.10** | **39.98 ±0.53** | **61.33 ±0.20** | **62.53 ±0.24** | **51.76 ±0.12** | **51.50 ±0.32** | **56.03 ±0.44** | **65.94 ±0.19** | **61.16 ±0.11** | **51.47 ±0.04** | **54.52 ±0.27** | **43.67 ±0.04** | **51.68 ±0.21** |
|  | SAR | 36.04 ±0.00 | 37.02 ±0.26 | 31.38 ±0.30 | 61.13 ±0.35 | 38.07 ±0.44 | 58.00 ±0.59 | 59.08 ±0.36 | 48.44 ±0.47 | 48.84 ±0.92 | 53.52 ±0.16 | 62.57 ±0.47 | 55.19 ±2.20 | 48.87 ±0.15 | 52.01 ±0.57 | 40.71 ±0.19 | 48.72 ±0.50 |
|  | **+ SNAP** | **37.91 ±0.39** | **38.85 ±0.25** | **32.92 ±0.38** | **63.17 ±0.23** | **39.35 ±0.45** | **60.51 ±0.26** | **62.01 ±0.11** | **51.11 ±0.26** | **50.48 ±0.11** | **55.47 ±0.38** | **65.07 ±0.15** | **59.69 ±0.15** | **51.24 ±0.47** | **54.10 ±0.06** | **42.80 ±0.28** | **50.98 ±0.28** |
|  | RoTTA | 35.55 ±0.33 | 36.34 ±0.31 | 30.55 ±0.45 | 60.76 ±0.50 | 37.42 ±0.50 | 57.50 ±0.56 | 58.57 ±0.30 | 47.87 ±0.28 | 48.31 ±0.97 | 53.11 ±0.23 | 61.90 ±0.62 | 54.70 ±1.98 | 48.25 ±0.08 | 51.37 ±0.62 | 40.29 ±0.11 | 48.16 ±0.52 |
|  | **+ SNAP** | **37.82 ±0.16** | **38.72 ±0.05** | **32.60 ±0.10** | **63.53 ±0.01** | **39.80 ±0.49** | **61.00 ±0.37** | **62.27 ±0.23** | **51.42 ±0.06** | **51.33 ±0.12** | **55.71 ±0.42** | **65.64 ±0.14** | **60.89 ±0.18** | **51.50 ±0.18** | **54.27 ±0.19** | **42.92 ±0.47** | **51.30 ±0.21** |

## C.3 ImageNet-C

Table 26: STTA classification accuracy (%) comparing with and without SNAP on ImageNet-C through Adaptation Rates(AR) (0.5, 0.3, and 0.1), including results for full adaptation (AR=1). **Bold** numbers are the highest accuracy.

| AR | Methods | Gau. | Shot | Imp. | Def. | Gla. | Mot. | Zoom | Snow | Fro. | Fog | Brit. | Cont. | Elas. | Pix. | JPEG | Avg. |
|---|---|---|---|---|---|---|---|---|---|---|---|---|---|---|---|---|---|
| 1 | Source | 3.00 ±0.00 | 3.70 ±0.00 | 2.64 ±0.00 | 17.90 ±0.00 | 9.74 ±0.00 | 14.72 ±0.00 | 22.45 ±0.00 | 16.60 ±0.00 | 23.06 ±0.00 | 24.00 ±0.00 | 59.11 ±0.00 | 5.37 ±0.00 | 16.50 ±0.00 | 20.88 ±0.00 | 32.63 ±0.00 | 18.15 ±0.00 |
| | BN stats | 14.29 ±0.05 | 15.06 ±0.02 | 14.89 ±0.08 | 13.30 ±0.08 | 13.38 ±0.08 | 23.78 ±0.05 | 35.22 ±0.06 | 31.78 ±0.04 | 30.26 ±0.07 | 44.40 ±0.14 | 62.39 ±0.11 | 15.14 ±0.05 | 40.42 ±0.10 | 45.25 ±0.04 | 36.53 ±0.16 | 29.07 ±0.07 |
| | Tent | 27.03 ±0.05 | 28.98 ±0.08 | 28.64 ±0.29 | 24.66 ±0.27 | 23.63 ±0.25 | 38.70 ±0.10 | 45.77 ±0.12 | 44.82 ±0.08 | 38.06 ±0.35 | 54.59 ±0.08 | 64.61 ±0.10 | 16.84 ±1.51 | 51.64 ±0.10 | 55.54 ±0.15 | 49.38 ±0.07 | 39.53 ±0.24 |
| | CoTTA | 13.12 ±0.08 | 13.98 ±0.07 | 13.94 ±0.01 | 12.44 ±0.10 | 12.18 ±0.04 | 23.74 ±0.04 | 35.22 ±0.06 | 31.78 ±0.05 | 30.26 ±0.06 | 44.40 ±0.14 | 62.40 ±0.11 | 15.13 ±0.03 | 40.42 ±0.10 | 45.26 ±0.04 | 36.53 ±0.16 | 28.72 ±0.07 |
| | EATA | 29.62 ±0.02 | 31.79 ±0.09 | 31.17 ±0.19 | 26.89 ±0.03 | 26.30 ±0.15 | 40.65 ±0.12 | 47.44 ±0.06 | 46.29 ±0.09 | 40.78 ±0.05 | 55.57 ±0.08 | 64.97 ±0.08 | 38.02 ±0.08 | 52.66 ±0.20 | 56.03 ±0.04 | 50.26 ±0.16 | 42.56 ±0.10 |
| | SAR | 29.23 ±0.40 | 31.14 ±1.44 | 29.88 ±0.96 | 29.29 ±0.72 | 27.39 ±0.97 | 39.76 ±0.63 | 44.13 ±0.11 | 45.98 ±0.23 | 29.39 ±0.30 | 55.13 ±0.20 | 63.71 ±0.08 | 17.34 ±0.61 | 52.31 ±0.08 | 56.09 ±0.18 | 49.35 ±0.13 | 39.34 ±0.47 |
| | RoTTA | 20.60 ±0.07 | 22.83 ±0.09 | 19.81 ±0.24 | 10.46 ±0.04 | 10.10 ±0.26 | 21.31 ±0.27 | 31.83 ±0.23 | 39.66 ±0.18 | 32.09 ±0.18 | 46.08 ±0.23 | 62.22 ±0.27 | 20.27 ±0.49 | 42.54 ±0.29 | 47.47 ±0.23 | 40.67 ±0.10 | 31.20 ±0.21 |
| 0.5 | Tent | 25.24 ±0.10 | 26.86 ±0.27 | 26.35 ±0.08 | 23.26 ±0.06 | 22.41 ±0.05 | 35.99 ±0.09 | 44.60 ±0.10 | 42.96 ±0.13 | 37.68 ±0.17 | 53.60 ±0.15 | 64.40 ±0.12 | 21.35 ±0.94 | 50.23 ±0.12 | 54.32 ±0.15 | 47.93 ±0.04 | 38.48 ±0.17 |
| | + SNAP | **28.05** ±0.00 | **29.97** ±0.04 | **29.39** ±0.19 | **25.73** ±0.15 | **23.39** ±0.06 | **38.49** ±0.17 | **45.65** ±0.03 | **44.21** ±0.09 | **39.57** ±0.10 | **53.90** ±0.10 | **64.52** ±0.09 | **34.39** ±1.83 | **49.99** ±0.14 | **54.88** ±0.07 | **48.72** ±0.09 | **40.72** ±0.21 |
| | CoTTA | 11.99 ±0.13 | 13.04 ±0.20 | 12.86 ±0.10 | 11.90 ±0.07 | 11.64 ±0.07 | 22.92 ±0.02 | 35.06 ±0.06 | 31.20 ±0.09 | 29.97 ±0.07 | 44.28 ±0.07 | 62.16 ±0.05 | 14.02 ±0.09 | 40.39 ±0.05 | 45.29 ±0.09 | 36.58 ±0.12 | 28.22 ±0.09 |
| | + SNAP | **15.16** ±0.14 | **15.96** ±0.02 | **15.86** ±0.14 | **13.98** ±0.04 | **14.13** ±0.00 | **24.69** ±0.09 | **36.51** ±0.07 | **32.59** ±0.16 | **31.71** ±0.06 | **45.98** ±0.09 | **63.62** ±0.05 | **15.72** ±0.04 | **42.05** ±0.09 | **46.71** ±0.24 | **37.93** ±0.14 | **30.17** ±0.09 |
| | EATA | 28.62 ±0.10 | 30.12 ±0.10 | 29.94 ±0.14 | 25.34 ±0.20 | 24.48 ±0.44 | 38.94 ±0.10 | 46.85 ±0.25 | 45.20 ±0.12 | **40.03** ±0.01 | 55.04 ±0.07 | 64.84 ±0.41 | 34.48 ±0.41 | 52.06 ±0.24 | 55.57 ±0.13 | 49.85 ±0.05 | 41.42 ±0.16 |
| | + SNAP | **30.00** ±0.29 | **31.88** ±0.17 | **31.47** ±0.13 | **26.93** ±0.21 | **26.64** ±0.28 | **39.16** ±0.15 | **47.23** ±0.07 | **45.36** ±0.13 | 39.75 ±0.14 | **55.30** ±0.14 | **64.52** ±0.10 | 33.75 ±0.07 | **52.29** ±0.09 | **55.66** ±0.18 | **50.48** ±0.08 | **42.03** ±0.15 |
| | SAR | 26.74 ±0.25 | 28.56 ±1.75 | 28.77 ±0.13 | 19.90 ±0.21 | 21.50 ±0.38 | 39.97 ±0.10 | 44.98 ±0.12 | 45.95 ±0.17 | 34.22 ±0.80 | 55.04 ±0.05 | 63.93 ±0.03 | 6.58 ±0.64 | 52.50 ±0.10 | 55.98 ±0.19 | 49.71 ±0.09 | 38.29 ±0.33 |
| | + SNAP | **31.58** ±0.38 | **33.22** ±2.44 | **33.77** ±0.56 | **26.47** ±1.69 | **26.26** ±0.94 | **44.01** ±0.10 | **47.94** ±0.04 | **48.77** ±0.12 | **42.51** ±0.09 | **56.96** ±0.13 | **64.86** ±0.10 | **28.31** ±10.99 | **54.23** ±0.08 | **57.55** ±0.16 | **51.90** ±0.19 | **43.22** ±1.20 |
| | RoTTA | 18.17 ±0.05 | 19.59 ±0.03 | 18.49 ±0.10 | 12.32 ±0.11 | 11.79 ±0.13 | 23.56 ±0.15 | 34.62 ±0.14 | 37.84 ±0.11 | 32.91 ±0.06 | 47.86 ±0.05 | 63.94 ±0.16 | 18.68 ±0.42 | 43.21 ±0.08 | 48.54 ±0.23 | 40.20 ±0.23 | 31.45 ±0.14 |
| | + SNAP | **20.43** ±0.03 | **22.03** ±0.08 | **21.05** ±0.11 | **15.47** ±0.11 | **14.49** ±0.07 | **26.36** ±0.06 | **36.46** ±0.13 | **38.98** ±0.09 | **34.15** ±0.12 | **48.41** ±0.13 | **64.02** ±0.23 | **20.74** ±0.10 | **43.66** ±0.10 | **49.16** ±0.15 | **41.05** ±0.11 | **33.10** ±0.11 |
| 0.3 | Tent | 23.63 ±0.08 | 25.18 ±0.37 | 24.80 ±0.28 | 21.81 ±0.02 | 20.97 ±0.18 | 34.11 ±0.07 | 43.60 ±0.04 | 41.44 ±0.05 | 36.98 ±0.04 | 52.66 ±0.15 | 64.21 ±0.13 | 22.74 ±0.04 | 48.96 ±0.16 | 53.46 ±0.07 | 46.80 ±0.09 | 37.42 ±0.12 |
| | + SNAP | **26.60** ±0.20 | **28.21** ±0.19 | **27.94** ±0.33 | **24.37** ±0.36 | **22.39** ±0.12 | **36.45** ±0.07 | **44.36** ±0.13 | **42.64** ±0.07 | **38.54** ±0.15 | **52.91** ±0.06 | **64.26** ±0.10 | **33.47** ±0.44 | 48.58 ±0.10 | **53.90** ±0.14 | **47.41** ±0.11 | **39.47** ±0.17 |
| | CoTTA | 11.74 ±0.09 | 12.74 ±0.06 | 12.68 ±0.07 | 11.77 ±0.17 | 11.62 ±0.14 | 22.64 ±0.14 | 34.97 ±0.07 | 31.05 ±0.07 | 29.81 ±0.01 | 44.24 ±0.13 | 62.12 ±0.05 | 13.73 ±0.06 | 40.31 ±0.15 | 45.19 ±0.08 | 36.71 ±0.09 | 28.09 ±0.09 |
| | + SNAP | **15.26** ±0.16 | **16.00** ±0.09 | **15.83** ±0.06 | **13.81** ±0.04 | **14.13** ±0.01 | **24.84** ±0.03 | **36.46** ±0.13 | **32.58** ±0.03 | **31.73** ±0.08 | **46.04** ±0.08 | **63.52** ±0.21 | **15.69** ±0.06 | **42.18** ±0.07 | **46.74** ±0.05 | **38.00** ±0.14 | **30.19** ±0.08 |
| | EATA | 27.35 ±0.04 | 29.03 ±0.15 | 28.62 ±0.27 | 23.94 ±0.06 | 23.45 ±0.60 | 37.21 ±0.30 | 46.18 ±0.13 | 44.05 ±0.20 | 39.19 ±0.22 | 54.52 ±0.01 | 64.54 ±0.62 | 32.20 ±0.16 | 51.22 ±0.16 | 55.00 ±0.10 | 49.27 ±0.21 | 40.38 ±0.21 |
| | + SNAP | **29.48** ±0.14 | **31.20** ±0.04 | **30.69** ±0.11 | **26.68** ±0.14 | **25.90** ±0.25 | **38.24** ±0.01 | **46.60** ±0.22 | **44.62** ±0.06 | **39.31** ±0.19 | **54.82** ±0.06 | 64.44 ±0.13 | **32.87** ±0.29 | **51.41** ±0.25 | **55.41** ±0.06 | **49.78** ±0.14 | **41.43** ±0.14 |
| | SAR | 28.12 ±0.13 | 29.30 ±0.89 | 29.63 ±0.17 | 22.37 ±0.47 | 23.88 ±0.33 | 39.34 ±0.18 | 45.36 ±0.11 | 45.69 ±0.18 | 36.73 ±0.79 | 54.91 ±0.07 | 64.11 ±0.02 | 10.96 ±1.33 | 52.22 ±0.19 | 55.76 ±0.13 | 49.60 ±0.08 | 39.20 ±0.34 |
| | + SNAP | **32.63** ±0.11 | **34.69** ±0.23 | **34.26** ±0.18 | **28.91** ±0.27 | **27.96** ±0.29 | **43.51** ±0.14 | **47.79** ±0.03 | **48.27** ±0.11 | **42.41** ±0.13 | **56.45** ±0.09 | **64.77** ±0.07 | **32.76** ±3.04 | **53.74** ±0.13 | **57.21** ±0.28 | **51.67** ±0.12 | **43.80** ±0.35 |
| | RoTTA | 16.90 ±0.15 | 17.88 ±0.11 | 17.25 ±0.08 | 12.89 ±0.17 | 12.51 ±0.05 | 23.96 ±0.03 | 35.26 ±0.16 | 36.26 ±0.01 | 32.32 ±0.07 | 47.25 ±0.02 | 63.98 ±0.13 | 17.46 ±0.18 | 42.77 ±0.09 | 48.21 ±0.24 | 39.35 ±0.15 | 30.95 ±0.11 |
| | + SNAP | **18.63** ±0.07 | **19.94** ±0.08 | **19.35** ±0.06 | **14.88** ±0.08 | **14.34** ±0.05 | **25.88** ±0.03 | **36.47** ±0.03 | **37.13** ±0.02 | **33.32** ±0.03 | **47.74** ±0.11 | 63.96 ±0.21 | **19.08** ±0.17 | **42.98** ±0.17 | **48.73** ±0.07 | **40.27** ±0.20 | **32.18** ±0.09 |
| 0.1 | Tent | 22.00 ±3.47 | 23.51 ±3.92 | 23.07 ±3.85 | 19.38 ±2.30 | 18.86 ±2.06 | 32.15 ±3.40 | 42.29 ±2.45 | 39.70 ±3.27 | 34.33 ±0.60 | 51.62 ±2.30 | 63.70 ±0.29 | 15.79 ±4.61 | 47.74 ±2.84 | 52.35 ±2.27 | 45.54 ±2.98 | 35.47 ±2.71 |
| | + SNAP | **26.21** ±4.92 | **27.85** ±5.36 | **27.50** ±5.30 | **23.62** ±4.23 | **22.73** ±4.11 | **36.01** ±5.57 | **44.11** ±3.72 | **42.19** ±4.49 | **38.15** ±3.37 | **52.95** ±3.47 | **64.57** ±1.18 | **30.23** ±5.15 | **48.56** ±4.29 | **53.71** ±3.31 | **47.09** ±4.09 | **39.03** ±4.17 |
| | CoTTA | 10.97 ±0.32 | 11.92 ±0.32 | 11.98 ±0.18 | 11.45 ±0.04 | 11.38 ±0.34 | 22.39 ±0.02 | 34.96 ±0.15 | 30.88 ±0.14 | 29.89 ±0.09 | 44.09 ±0.23 | 61.96 ±0.05 | 13.08 ±0.28 | 40.20 ±0.18 | 45.27 ±0.16 | 36.71 ±0.10 | 27.81 ±0.17 |
| | + SNAP | **15.13** ±0.06 | **16.03** ±0.09 | **15.91** ±0.04 | **13.86** ±0.00 | **14.02** ±0.07 | **24.90** ±0.05 | **36.51** ±0.05 | **32.56** ±0.06 | **31.81** ±0.12 | **46.02** ±0.04 | **63.60** ±0.09 | **15.69** ±0.04 | **41.94** ±0.09 | **46.78** ±0.12 | **38.03** ±0.09 | **30.19** ±0.07 |
| | EATA | 22.43 ±0.05 | 23.78 ±0.16 | 23.26 ±0.43 | 19.38 ±0.26 | 19.42 ±0.51 | 32.18 ±0.31 | 43.22 ±0.19 | 40.65 ±0.15 | 36.64 ±0.16 | 52.38 ±0.27 | 63.87 ±1.52 | 24.59 ±0.40 | 48.13 ±0.12 | 52.89 ±0.14 | 46.33 ±0.32 | 36.61 ±0.32 |
| | + SNAP | **26.10** ±0.09 | **27.29** ±0.13 | **27.13** ±0.20 | **22.38** ±0.32 | **22.15** ±0.14 | **33.45** ±0.08 | **43.92** ±0.08 | **40.96** ±0.16 | **36.68** ±0.01 | **52.71** ±0.09 | 63.77 ±0.10 | **27.93** ±0.24 | **48.47** ±0.09 | **53.23** ±0.17 | **47.46** ±0.15 | **38.24** ±0.15 |
| | SAR | 26.12 ±0.17 | 27.56 ±0.01 | 26.93 ±0.11 | 22.51 ±0.24 | 23.35 ±0.21 | 36.03 ±0.21 | 44.48 ±0.09 | 43.19 ±0.09 | 37.26 ±0.32 | 53.82 ±0.21 | 64.15 ±0.11 | 19.87 ±2.10 | 50.78 ±0.12 | 54.78 ±0.18 | 48.43 ±0.07 | 38.62 ±0.28 |
| | + SNAP | **30.28** ±0.16 | **31.97** ±0.24 | **31.30** ±0.12 | **26.67** ±0.34 | **26.31** ±0.37 | **39.66** ±0.25 | **46.08** ±0.04 | **45.43** ±0.09 | **40.26** ±0.13 | **54.76** ±0.23 | **64.62** ±0.05 | **36.12** ±0.67 | **51.26** ±0.06 | **55.42** ±0.20 | **49.63** ±0.06 | **41.99** ±0.20 |
| | RoTTA | 14.77 ±0.04 | 15.59 ±0.04 | 15.33 ±0.04 | 13.17 ±0.07 | 13.19 ±0.10 | 23.85 ±0.05 | 35.38 ±0.05 | 32.73 ±0.03 | 30.77 ±0.04 | 45.22 ±0.15 | **63.08** ±0.12 | 15.62 ±0.02 | 41.05 ±0.10 | 46.15 ±0.07 | 37.19 ±0.13 | 29.54 ±0.07 |
| | + SNAP | **15.35** ±0.03 | **16.20** ±0.01 | **16.01** ±0.07 | **13.67** ±0.09 | **13.66** ±0.07 | **24.27** ±0.03 | **35.62** ±0.01 | **33.04** ±0.07 | **31.02** ±0.04 | **45.38** ±0.11 | 62.95 ±0.08 | **15.96** ±0.08 | **41.06** ±0.11 | **46.17** ±0.07 | **37.44** ±0.19 | **29.85** ±0.07 |

Table 27: STTA classification accuracy (%) comparing with and without SNAP on ImageNet-C through Adaptation Rates(AR) (0.05, 0.03, and 0.01). **Bold** numbers are the highest accuracy.

| AR | Methods | Gau. | Shot | Imp. | Def. | Gla. | Mot. | Zoom | Snow | Fro. | Fog | Brit. | Cont. | Elas. | Pix. | JPEG | Avg. |
|---|---|---|---|---|---|---|---|---|---|---|---|---|---|---|---|---|---|
| 0.05 | Tent | 23.77 ±0.40 | 24.65 ±0.43 | 24.44 ±0.58 | 20.54 ±0.70 | 20.27 ±0.69 | 32.73 ±0.30 | 43.57 ±0.14 | 40.82 ±0.15 | 35.92 ±0.33 | 52.78 ±0.12 | 63.82 ±0.02 | 15.95 ±1.18 | 49.33 ±0.18 | 53.46 ±0.09 | 47.19 ±0.03 | 36.62 ±0.35 |
| | **+ SNAP** | **29.12 ±0.09** | **30.46 ±0.22** | **30.30 ±0.48** | **25.77 ±0.20** | **25.22 ±0.23** | **38.21 ±0.43** | **46.14 ±0.00** | **44.29 ±0.13** | **39.95 ±0.07** | **54.65 ±0.15** | **65.47 ±0.09** | **33.81 ±1.10** | **50.83 ±0.13** | **55.59 ±0.10** | **49.21 ±0.03** | **41.27 ±0.23** |
| | CoTTA | 11.03 ±0.30 | 11.91 ±0.57 | 11.75 ±0.33 | 11.03 ±0.24 | 11.20 ±0.46 | 22.30 ±0.18 | 34.98 ±0.05 | 30.87 ±0.08 | 29.78 ±0.01 | 43.99 ±0.11 | 61.87 ±0.06 | 12.92 ±0.36 | 40.26 ±0.19 | 45.23 ±0.17 | 36.63 ±0.07 | 27.72 ±0.21 |
| | **+ SNAP** | **15.22 ±0.08** | **15.97 ±0.11** | **15.93 ±0.03** | **13.91 ±0.06** | **14.05 ±0.12** | **24.87 ±0.04** | **36.48 ±0.00** | **32.60 ±0.07** | **31.65 ±0.04** | **46.09 ±0.03** | **63.59 ±0.07** | **15.67 ±0.05** | **42.00 ±0.03** | **46.71 ±0.09** | **37.96 ±0.09** | **30.18 ±0.06** |
| | EATA | 19.53 ±0.31 | 20.65 ±0.66 | 20.72 ±0.75 | 16.74 ±0.41 | 16.96 ±0.58 | 29.11 ±0.49 | 41.22 ±0.18 | 37.96 ±0.23 | 34.84 ±0.21 | 50.75 ±0.13 | 63.29 ±1.26 | 19.86 ±0.35 | 45.92 ±0.17 | 51.15 ±0.09 | 44.13 | 34.19 ±0.41 |
| | **+ SNAP** | **22.83 ±0.10** | **23.95 ±0.34** | **23.62 ±0.30** | **19.43 ±0.09** | **19.70 ±0.19** | **30.34 ±0.56** | **41.59 ±0.08** | **38.06 ±0.11** | **35.06 ±0.21** | **50.98 ±0.18** | **63.30 ±0.13** | **23.72 ±0.30** | **46.26 ±0.16** | **51.52 ±0.16** | **45.46 ±0.18** | **35.72 ±0.21** |
| | SAR | 23.25 ±0.21 | 24.23 ±0.34 | 23.66 ±0.30 | 19.98 ±0.09 | 20.38 ±0.16 | 33.05 ±0.30 | 43.04 ±0.16 | 40.73 ±0.02 | 36.06 ±0.10 | 52.61 ±0.11 | 64.09 ±0.84 | 20.17 ±0.11 | 49.00 ±0.10 | 53.35 ±0.11 | 46.73 ±0.20 | 36.69 ±0.20 |
| | **+ SNAP** | **27.54 ±0.16** | **29.03 ±0.05** | **28.66 ±0.04** | **24.05 ±0.16** | **23.42 ±0.08** | **36.28 ±0.12** | **44.12 ±0.10** | **42.89 ±0.11** | **38.54 ±0.07** | **53.24 ±0.07** | **64.25 ±0.05** | **31.83 ±0.24** | **48.79 ±0.23** | **54.04 ±0.19** | **47.80 ±0.08** | **39.63 ±0.12** |
| | RoTTA | 14.42 ±0.06 | 15.22 ±0.05 | 15.02 ±0.10 | 13.25 ±0.11 | 13.31 ±0.07 | 23.79 ±0.03 | 35.27 ±0.08 | 32.09 ±0.05 | 30.43 ±0.07 | 44.71 ±0.13 | 62.64 ±0.14 | 15.24 ±0.09 | 40.63 ±0.10 | 45.55 ±0.07 | 36.75 ±0.16 | 29.22 ±0.09 |
| | **+ SNAP** | **14.65 ±0.06** | **15.48 ±0.02** | **15.29 ±0.08** | **13.43 ±0.09** | **13.45 ±0.09** | **23.93 ±0.03** | **35.33 ±0.05** | **32.18 ±0.04** | **30.53 ±0.05** | **44.71 ±0.16** | **62.58 ±0.10** | **15.41 ±0.04** | **40.64 ±0.09** | **45.55 ±0.10** | **36.81 ±0.14** | **29.33 ±0.08** |
| 0.03 | Tent | 21.76 ±0.17 | 22.76 ±0.35 | 22.58 ±0.17 | 19.06 ±0.04 | 18.90 ±0.12 | 30.85 ±0.22 | 42.34 ±0.12 | 38.94 ±0.26 | 35.53 ±0.31 | 51.58 ±0.18 | 63.42 ±0.91 | 18.61 ±0.11 | 47.96 ±0.26 | 52.41 ±0.21 | 45.56 ±0.08 | 35.48 ±0.23 |
| | **+ SNAP** | **26.42 ±0.14** | **28.20 ±0.26** | **27.81 ±0.37** | **23.79 ±0.46** | **22.82 ±0.21** | **35.77 ±0.11** | **44.80 ±0.16** | **42.37 ±0.34** | **38.81 ±0.14** | **53.34 ±0.06** | **64.95 ±0.11** | **30.05 ±0.62** | **49.28 ±0.17** | **54.16 ±0.09** | **47.57 ±0.08** | **39.34 ±0.22** |
| | CoTTA | 10.61 ±0.18 | 12.36 ±0.36 | 11.78 ±0.57 | 11.66 ±0.57 | 11.32 ±0.26 | 22.25 ±0.11 | 35.01 ±0.18 | 30.88 ±0.24 | 29.84 ±0.07 | 44.09 ±0.11 | 61.83 ±0.16 | 12.92 ±0.12 | 40.26 ±0.19 | 45.20 ±0.11 | 36.58 ±0.09 | 27.77 ±0.22 |
| | **+ SNAP** | **15.29 ±0.08** | **16.02 ±0.07** | **16.00 ±0.09** | **13.99 ±0.07** | **14.06 ±0.11** | **24.78 ±0.05** | **36.54 ±0.07** | **32.62 ±0.06** | **31.70 ±0.08** | **46.01 ±0.01** | **63.49 ±0.04** | **15.69 ±0.04** | **42.05 ±0.18** | **46.75 ±0.19** | **37.97 ±0.08** | **30.20 ±0.08** |
| | EATA | 17.17 ±0.41 | 18.34 ±0.19 | 17.94 ±0.36 | 14.48 ±0.82 | 15.04 ±0.22 | 26.31 ±0.25 | 39.47 ±0.33 | 35.51 ±0.50 | 33.41 ±0.33 | 49.16 ±0.19 | 63.06 ±0.05 | 18.01 ±0.88 | 44.16 ±0.31 | 49.90 ±0.09 | 42.47 ±0.31 | 32.30 ±0.35 |
| | **+ SNAP** | **20.75 ±0.32** | **21.87 ±0.41** | **21.28 ±0.35** | **17.34 ±0.30** | **17.90 ±0.34** | **28.08 ±0.34** | **39.84 ±0.16** | **36.27 ±0.13** | **33.54 ±0.11** | **49.50 ±0.12** | **63.04 ±0.07** | **20.86 ±0.33** | **44.68 ±0.28** | **49.97 ±0.03** | **43.53 ±0.23** | **33.90 ±0.23** |
| | SAR | 20.38 ±0.10 | 21.34 ±0.14 | 21.18 ±0.36 | 18.24 ±0.18 | 18.28 ±0.27 | 30.56 ±0.08 | 41.63 ±0.12 | 38.57 ±0.17 | 35.23 ±0.28 | 51.19 ±0.22 | 63.74 ±0.04 | 20.40 ±0.20 | 47.32 ±0.09 | 52.02 ±0.09 | 44.81 ±0.19 | 34.99 ±0.17 |
| | **+ SNAP** | **25.11 ±0.23** | **26.27 ±0.31** | **26.00 ±0.10** | **22.02 ±0.49** | **21.25 ±0.56** | **33.51 ±0.31** | **42.86 ±0.14** | **40.83 ±0.16** | **37.09 ±0.21** | **51.87 ±0.18** | **63.83 ±0.10** | **28.36 ±0.29** | **47.19 ±0.34** | **52.63 ±0.06** | **45.80 ±0.30** | **37.64 ±0.25** |
| | RoTTA | 14.36 ±0.04 | 15.12 ±0.03 | 14.95 ±0.08 | 13.30 ±0.08 | 13.34 ±0.08 | 23.78 ±0.04 | 35.23 ±0.05 | 31.89 ±0.04 | 30.33 ±0.07 | 44.52 ±0.11 | 62.48 ±0.12 | 15.20 ±0.01 | 40.50 ±0.11 | 45.36 ±0.07 | 36.63 ±0.17 | 29.13 ±0.07 |
| | **+ SNAP** | **14.45 ±0.04** | **15.21 ±0.02** | **15.06 ±0.08** | **13.35 ±0.08** | **13.42 ±0.07** | **23.83 ±0.04** | **35.26 ±0.06** | **31.92 ±0.02** | **30.36 ±0.08** | **44.53 ±0.10** | **62.47 ±0.09** | **15.27 ±0.04** | **40.50 ±0.10** | **45.39 ±0.08** | **36.65 ±0.16** | **29.18 ±0.07** |
| 0.01 | Tent | 17.09 ±0.14 | 17.70 ±0.10 | 17.69 ±0.13 | 14.91 ±0.23 | 15.25 ±0.09 | 25.23 ±0.25 | 38.66 ±0.27 | 34.15 ±0.27 | 32.28 ±0.21 | 48.14 ±0.21 | 62.65 ±0.48 | 15.76 ±0.23 | 43.44 ±0.04 | 49.14 ±0.10 | 41.18 ±0.10 | 31.55 ±0.19 |
| | **+ SNAP** | **20.66 ±0.02** | **21.73 ±0.12** | **21.55 ±0.18** | **18.46 ±0.34** | **18.28 ±0.33** | **29.88 ±0.12** | **40.63 ±0.14** | **36.97 ±0.21** | **34.89 ±0.10** | **49.85 ±0.26** | **64.29 ±0.10** | **22.64 ±0.14** | **45.13 ±0.29** | **50.77 ±0.51** | **43.17 ±0.19** | **34.59 ±0.19** |
| | CoTTA | 11.11 ±0.61 | 13.24 ±0.12 | 11.86 ±0.65 | 10.85 ±0.59 | 10.97 ±0.98 | 22.18 ±0.05 | 34.96 ±0.18 | 30.88 ±0.14 | 29.63 ±0.21 | 44.09 ±0.21 | 61.71 ±0.22 | 12.81 ±0.53 | 40.16 ±0.20 | 45.14 ±0.22 | 36.73 ±0.12 | 27.75 ±0.34 |
| | **+ SNAP** | **15.09 ±0.04** | **16.00 ±0.09** | **15.83 ±0.14** | **13.84 ±0.09** | **14.06 ±0.02** | **24.70 ±0.07** | **36.47 ±0.02** | **32.59 ±0.11** | **31.66 ±0.03** | **46.10 ±0.15** | **63.62 ±0.07** | **15.60 ±0.06** | **42.03 ±0.10** | **46.74 ±0.01** | **38.17 ±0.20** | **30.17 ±0.08** |
| | EATA | 14.85 ±0.13 | 15.61 ±0.21 | 15.69 ±0.21 | 13.26 ±0.04 | 13.37 ±0.06 | 23.72 ±0.19 | 36.18 ±0.13 | 32.57 ±0.09 | 31.14 ±0.06 | 46.06 ±0.29 | 62.35 ±0.09 | 13.88 ±0.35 | 41.91 ±0.17 | 47.00 ±0.15 | 38.88 ±0.09 | 29.76 ±0.15 |
| | **+ SNAP** | **16.73 ±0.12** | **17.55 ±0.10** | **17.30 ±0.19** | **14.35 ±0.09** | **14.64 ±0.10** | **24.13 ±0.36** | **36.83 ±0.23** | **32.81 ±0.08** | **31.09 ±0.10** | **46.63 ±0.19** | **62.20 ±0.16** | **15.26 ±0.54** | **42.34 ±0.12** | **47.44 ±0.18** | **39.81 ±0.34** | **30.61 ±0.19** |
| | SAR | 16.08 ±0.08 | 17.04 ±0.07 | 16.69 ±0.10 | 14.72 ±0.16 | 14.78 ±0.12 | 25.92 ±0.05 | 37.85 ±0.24 | 34.07 ±0.11 | 32.25 ±0.13 | 47.66 ±0.05 | 63.15 ±0.15 | 17.20 ±0.20 | 43.05 ±0.09 | 48.78 ±0.20 | 40.14 ±0.13 | 31.29 ±0.13 |
| | **+ SNAP** | **18.89 ±0.15** | **19.45 ±0.15** | **19.70 ±0.12** | **16.70 ±0.14** | **16.55 ±0.15** | **27.69 ±0.16** | **38.57 ±0.11** | **35.34 ±0.22** | **33.09 ±0.31** | **48.08 ±0.07** | **63.04 ±0.12** | **20.39 ±0.33** | **42.95 ±0.29** | **48.76 ±0.26** | **40.99 ±0.33** | **32.68 ±0.18** |
| | RoTTA | 14.30 ±0.05 | 15.06 ±0.03 | 14.89 ±0.07 | 13.30 ±0.07 | 13.37 ±0.08 | 23.78 ±0.04 | 35.22 ±0.06 | 31.79 ±0.04 | 30.27 ±0.06 | 44.40 ±0.14 | 62.40 ±0.11 | 15.16 ±0.06 | 40.42 ±0.10 | 45.27 ±0.05 | 36.54 ±0.16 | 29.08 ±0.07 |
| | **+ SNAP** | **14.30 ±0.06** | **15.07 ±0.03** | **14.92 ±0.08** | **13.30 ±0.08** | **13.38 ±0.07** | **23.78 ±0.04** | **35.22 ±0.06** | **31.78 ±0.04** | **30.26 ±0.07** | **44.41 ±0.14** | **62.40 ±0.11** | **15.15 ±0.05** | **40.43 ±0.09** | **45.27 ±0.04** | **36.54 ±0.15** | **29.08 ±0.07** |

## C.4 Additional results on ablation study

In this section, we provide additional details on the ablation study to evaluate the contributions of the CnDRM and IoBMN components in SNAP. Specifically, we measured the average accuracy across 15 corruption types on CIFAR10-C and CIFAR100-C datasets under varying adaptation rates (0.3, 0.1, 0.05) to thoroughly assess the effectiveness of each component.

Tables 28 and 29 summarize the results for different combinations of CnDRM and IoBMN across these adaptation rates. The results indicate that the combination of CnDRM (Class and Domain Representative sampling) and IoBMN (inference using memory statistics corrected to match the test batch) consistently yields the highest accuracy. This trend is observed across all evaluated adaptation rates, suggesting that both components contribute significantly to enhancing adaptation performance.

Moreover, individual evaluations show that each component has a distinct positive effect, as evidenced by consistently higher accuracy compared to using no adaptation or only a single component. This emphasizes the complementary nature of CnDRM and IoBMN, which together provide robust adaptation capabilities for domain-shifted scenarios. These tables provide further insight into the benefits of each configuration and how the synergy of CnDRM and IoBMN results in improved robustness against various corruptions.

Table 28: STTA classification accuracy (%) of ablative settings on the CIFAR10-C, adaptation rate (AR) 0.3, 0.1, and 0.05. Averaged over all 15 corruptions. **Bold** numbers are the highest accuracy.

| AR | Methods | Tent | CoTTA | EATA | SAR | RoTTA |
|---|---|---|---|---|---|---|
| 0.3 | Naïve | 78.86 ±0.12 | 69.75 ±0.08 | 79.02 ±0.14 | 77.83 ±0.11 | 75.39 ±0.09 |
| | Random | 78.90 ±0.15 | 66.04 ±0.10 | 78.97 ±0.13 | 77.77 ±0.12 | 75.06 ±0.07 |
| | LowEntropy | 78.68 ±0.11 | 63.74 ±0.16 | 78.42 ±0.09 | 76.21 ±0.10 | 72.83 ±0.14 |
| | CRM | 80.32 ±0.07 | 66.50 ±0.12 | 80.14 ±0.08 | 75.78 ±0.13 | 75.49 ±0.06 |
| | CnDRM | 79.62 ±0.13 | 77.68 ±0.10 | 79.63 ±0.12 | 78.22 ±0.09 | 75.85 ±0.08 |
| | CnDRM+EMA | 80.96 ±0.06 | 72.42 ±0.14 | 80.27 ±0.11 | 78.19 ±0.13 | 76.73 ±0.07 |
| | **CnDRM+IoBMN** | **81.23** ±0.09 | **78.75** ±0.10 | **81.30** ±0.07 | **79.77** ±0.08 | **77.41** ±0.06 |
| 0.1 | Naïve | 76.81 ±0.18 | 66.42 ±0.12 | 76.29 ±0.11 | 76.01 ±0.07 | 74.78 ±0.15 |
| | Random | 77.08 ±0.14 | 65.61 ±0.08 | 76.59 ±0.10 | 76.33 ±0.13 | 75.01 ±0.16 |
| | LowEntropy | 75.66 ±0.09 | 63.19 ±0.14 | 74.89 ±0.12 | 74.41 ±0.18 | 72.60 ±0.10 |
| | CRM | 77.77 ±0.05 | 65.71 ±0.19 | 77.18 ±0.08 | 74.36 ±0.11 | 75.27 ±0.17 |
| | CnDRM | 77.46 ±0.07 | 77.69 ±0.10 | 77.17 ±0.06 | 76.85 ±0.09 | 75.64 ±0.08 |
| | CnDRM+EMA | 78.02 ±0.12 | 72.19 ±0.15 | 77.05 ±0.11 | 76.84 ±0.13 | 76.18 ±0.05 |
| | **CnDRM+IoBMN** | **78.95** ±0.09 | **78.83** ±0.06 | **78.61** ±0.13 | **78.06** ±0.07 | **77.07** ±0.10 |
| 0.05 | Naïve | 75.75 ±0.18 | 67.22 ±0.12 | 75.55 ±0.14 | 75.25 ±0.17 | 74.80 ±0.11 |
| | Random | 75.82 ±0.13 | 65.90 ±0.21 | 75.56 ±0.16 | 75.27 ±0.15 | 74.91 ±0.10 |
| | LowEntropy | 74.07 ±0.20 | 64.08 ±0.25 | 73.73 ±0.19 | 73.58 ±0.22 | 72.83 ±0.14 |
| | CRM | 76.55 ±0.11 | 66.14 ±0.17 | 76.06 ±0.13 | 74.02 ±0.15 | 75.23 ±0.09 |
| | CnDRM | 76.53 ±0.14 | 77.67 ±0.16 | 76.29 ±0.18 | 76.18 ±0.12 | 75.61 ±0.13 |
| | CnDRM+EMA | 76.86 ±0.10 | 71.69 ±0.19 | 75.98 ±0.15 | 75.43 ±0.14 | 75.95 ±0.11 |
| | **CnDRM+IoBMN** | **77.93** ±0.09 | **78.73** ±0.13 | **77.76** ±0.12 | **77.21** ±0.11 | **77.05** ±0.08 |

Table 29: STTA classification accuracy (%) of ablative settings on the CIFAR100-C, adaptation rate (AR) 0.3, 0.1, and 0.05. Averaged over all 15 corruptions. **Bold** numbers are the highest accuracy.

| AR | Methods | Tent | CoTTA | EATA | SAR | RoTTA |
|---|---|---|---|---|---|---|
| 0.3 | Naïve | 53.36 ±0.22 | 39.11 ±0.17 | 49.97 ±0.19 | 56.65 ±0.20 | 49.84 ±0.18 |
| | Random | 53.00 ±0.24 | 33.49 ±0.21 | 49.24 ±0.17 | 56.06 ±0.26 | 49.00 ±0.16 |
| | LowEntropy | 53.53 ±0.20 | 32.29 ±0.28 | 45.51 ±0.23 | 55.84 ±0.22 | 44.77 ±0.19 |
| | CRM | 54.21 ±0.18 | 32.86 ±0.24 | 47.42 ±0.20 | 56.40 ±0.19 | 46.68 ±0.17 |
| | CnDRM | 55.15 ±0.21 | 50.02 ±0.14 | 51.36 ±0.16 | 57.72 ±0.18 | 50.74 ±0.15 |
| | CnDRM+EMA | 55.39 ±0.16 | 41.34 ±0.20 | 50.11 ±0.19 | 57.68 ±0.21 | 49.88 ±0.17 |
| | **CnDRM+IoBMN** | **57.27** ±0.13 | **50.32** ±0.15 | **52.19** ±0.14 | **58.44** ±0.16 | **51.55** ±0.12 |
| 0.1 | Naïve | 52.84 ±0.19 | 35.86 ±0.23 | 49.70 ±0.18 | 53.49 ±0.21 | 49.11 ±0.17 |
| | Random | 52.68 ±0.22 | 33.18 ±0.26 | 49.39 ±0.20 | 53.42 ±0.18 | 48.84 ±0.14 |
| | LowEntropy | 51.76 ±0.20 | 32.30 ±0.28 | 46.03 ±0.23 | 52.15 ±0.24 | 45.18 ±0.19 |
| | CRM | 52.43 ±0.17 | 32.54 ±0.25 | 47.68 ±0.21 | 53.12 ±0.20 | 47.01 ±0.16 |
| | CnDRM | 54.46 ±0.16 | 50.06 ±0.13 | 51.41 ±0.19 | 55.24 ±0.14 | 50.47 ±0.12 |
| | CnDRM+EMA | 54.36 ±0.15 | 41.63 ±0.22 | 50.21 ±0.18 | 54.84 ±0.17 | 49.95 ±0.13 |
| | **CnDRM+IoBMN** | **55.84** ±0.14 | **50.52** ±0.11 | **52.35** ±0.15 | **55.76** ±0.13 | **51.33** ±0.10 |
| 0.05 | Naïve | 51.24 ±0.18 | 33.20 ±0.25 | 49.81 ±0.16 | 51.50 ±0.21 | 49.12 ±0.19 |
| | Random | 51.35 ±0.20 | 33.71 ±0.22 | 49.57 ±0.17 | 51.48 ±0.20 | 48.98 ±0.15 |
| | LowEntropy | 49.79 ±0.24 | 32.36 ±0.26 | 46.65 ±0.19 | 49.51 ±0.23 | 45.41 ±0.18 |
| | CRM | 50.17 ±0.19 | 32.74 ±0.27 | 47.47 ±0.20 | 50.49 ±0.22 | 46.58 ±0.16 |
| | CnDRM | 52.86 ±0.14 | 50.08 ±0.13 | 51.47 ±0.17 | 53.09 ±0.15 | 50.44 ±0.13 |
| | CnDRM+EMA | 52.68 ±0.13 | 41.43 ±0.21 | 50.32 ±0.18 | 52.80 ±0.17 | 50.04 ±0.14 |
| | **CnDRM+IoBMN** | **54.13** ±0.11 | **50.63** ±0.14 | **52.43** ±0.16 | **53.59** ±0.12 | **51.41** ±0.10 |

# D  License of assets

**Datasets**  CIFAR10/CIFAR100 (MIT License), CIFAR10-C/CIFAR100-C (Creative Commons Attribution 4.0 International), ImageNet-C (Apache 2.0), and ImageNet-R/Scketch (MIT License).

**Codes**  Torchvision for ResNet18, ResNet50, and VitBase-LN (Apache 2.0), the official repository of CoTTA (MIT License), the official repository of Tent (MIT License), the official repository of EATA (MIT License), the official repository of SAR (BSD 3-Clause License), the official repository of RoTTA (MIT License), the official repository of T3A (MIT License), the official repository of FOA (NTUITIVE License) and the official repository of MECTA (Sony AI).

