# OpenReview forum: "SNAP: Low-Latency Test-Time Adaptation with Sparse Updates"
_NeurIPS.cc/2025/Conference — NeurIPS 2025 poster_

### Official Review · Reviewer_Vzgx · 2025-06-29

**Clarity:** 3
**Significance:** 3
**Originality:** 2
**Rating:** 5
**Confidence:** 3

**Summary:**

This paper proposes SNAP a test-time adaptation algorithm with low latency. It has two components: (1) CnDRM for selecting representative samples to update the model with partial data, and (2) IoBMN for leveraging these representative samples in adaptation. The algorithm saves lots of computation, while preserve most of the effectiveness of TTA algorithms.

**Questions:**

- In CnDRM, the author argue in criteria 2 that "selecting samples close to the domain centroid would enhance performance in STTA". In the previous EATA paper (one of the baselines the author consider), they instead drop the testing samples whose prediction is too close to the testing sample mean, which is a clear contradiction with this paper. Although the similarities are calculated in slightly different levels (early layer feature v.s. final prediction), could you explain what is the key differences, and are there any contradiction when combining these two methods?

**Ethical Concerns:**

["NO or VERY MINOR ethics concerns only"]

**Final Justification:**

The author did a good job in the rebuttal providing additional clarification and details. My concerns are fully solved.

**Limitations:**

Yes.

**Quality:**

3

**Strengths And Weaknesses:**

# Sterngths
- Achieving better efficiency-effectiveness trade-off in TTA algorithms is an important topic in TTA, especially when recent works are more and more complicated with only marginal improvements. It is important to evaluate whether TTA algorithms can be useful in practice.
- The proposed algorithm in clear and intuitive in high-level, easy to understand.
- The experiment in comprehensive on benchmarking datasets.
- I appreciate the experiments on popular edge devices, which are rarely seen in previous works.

# Major Weaknesses
- I am confused about the role of IoBMN. It is proposed as an replacement of standard BN adaptor, and use only the chosen samples in CnDRM. (1) However, it requires a forward pass to get the prediction and calculated scores used in CnDR. Does it mean the algorithm requires two forward passes for adaptation? (2) All the feature map for all samples are always available, why not just use the standard BN, with additional moving average?

# Minor Weaknesses
- Equation (1): I think this equation might only be true if you assume the feature distribution is Gaussian, with diagonal covariance matrix, (and requires a few lines of derivation). The author should (1) clearly define the distribution considered here (which is NOT data distribution, but the distribution of each scalar in the network) and (2) clearly state the assumptions to compute Wasserstein distance.
- Notation around Equation (2) is quite confusing. I believe m is a sample index, instead of a channel index. Should it be averaged over channel C and locaion L, instead of B and L? Also, if it is already normalized, why divided by C*M in Equation 3? I suggestion the author check these notation and give clear definition.

---

> ### Author Rebuttal · Authors · 2025-07-31
>
> **Weakness 1 (W1).** Thank you for your questions.
>
> (1) **Does IoBMN require two forward passes for adaptation?**
>
> No, IoBMN does **not** require an additional forward pass. The prediction and confidence scores used in CnDRM sampling are obtained during the *same* forward pass used for inference. The normalization statistics used by IoBMN are computed from the memory (CnDRM) that stores samples involved in sparse model updates. These statistics are reused without extra passes.
>
> (2) **Why not use standard BN with a moving average over all feature maps?**
>
> Rather than relying on all feature maps at inference, IoBMN uses the memory-derived statistics—representative of previously adapted domain-class conditions—as a stable reference. During inference, these memory statistics are shifted toward the current batch statistics using a shrink function. This *memory → batch* directional correction helps compensate for outdated memory statistics from infrequent updates, and improves normalization stability under sparse adaptation. *Table 2* and *Appendix D.4* show that IoBMN outperforms standard BN and batch-stat-only approaches across various adaptation rates and datasets.
>
> ---
> **W2.** The closed-form expression in Equation 1 indeed assumes a Gaussian distribution with diagonal covariance. Here we clarify both the assumptions and the derivation.
>
> (1) **Distribution definition**: The distribution in Equation 1 refers to the empirical distribution of scalar **feature activations per channel**, not the input data distribution. We consider each channel's activation statistics (mean and variance) in a deep layer’s feature map.
>
> (2) **Wasserstein approximation and assumptions**:
> To compute the Wasserstein distance efficiently, we approximate the feature distributions as **univariate Gaussian distributions** for each channel. The assumption is that the feature distributions can be well-approximated by Gaussians, and we assume these distributions are independent (i.e., **diagonal covariance**). The detailed assumptions are as follows:
>
> (2-1) **Gaussian assumption**:
> We assume that the feature activations across each channel follow a Gaussian distribution $\mathcal{N}(\mu, \sigma^2)$, simplifying the problem as Gaussian distributions are fully defined by their mean $\mu$ and variance $\sigma^2$. This assumption holds in many deep learning applications, particularly where feature activations are approximately Gaussian, especially in high-dimensional feature spaces.
>
> (2-2) **Diagonal covariance**:
>    We assume the covariance matrix for each Gaussian is diagonal, meaning features across different channels are independent. This is a common assumption in domain adaptation and transport-based methods, as it reduces the complexity of computing full covariance matrices and instead focuses on the variance of each channel (i.e., no correlation between channels).
>
> (2-3) **2-Wasserstein distance for Gaussian distributions**:
>    Given these assumptions (independent Gaussian distributions), the squared 2-Wasserstein distance between two univariate Gaussian distributions $\mathcal{N}(\mu_1, \sigma_1^2)$ and $\mathcal{N}(\mu_2, \sigma_2^2)$ is given by:
>
>    $$
>    W_2^2 = (\mu_1 - \mu_2)^2 + (\sigma_1 - \sigma_2)^2
>    $$
>
> This simplification enables efficient computation of the Wasserstein distance without requiring a full covariance matrix, which is computationally expensive.
>
> This approximation is widely used in domain adaptation literature [r1, r2] as it balances **computational efficiency** and **empirical performance**. The simplification allows us to avoid the complexities of full covariance matrices while providing meaningful distance measurements for aligning feature distributions.
>
> In the final manuscript, we will explicitly state these assumptions, the detailed derivations, and the rationale behind using this approximation.
>
> ---
> **W3.** Thank you for pointing this out. The subscript m is not an index, but indicates that these statistics are derived from the memory. We will revise the text to avoid any confusion.
>
> - In Equation 2, as we compute per-channel statistics, the averaging is correctly done over the batch $B$ and spatial dimensions $L$, not over channels.
>
> - In Equation 3, there was a typo. As correctly noted around line 218, the denominator should be L×M, not C×M. We apologize for the mistake. The division by L×M is necessary as the sample mean and variance are computed over L spatial locations for each M memory samples. Assuming i.i.d. features across spatial dimensions, the total number of independent observations per channel is L×M. Therefore, when estimating the variance of the sample mean and variance, the correct normalization factor is LxM, following standard results in sampling theory. We will revise the notations and equations accordingly in our final manuscript.
>
> ---
> **Question.** One might think there is a contradiction between our domain-centric sampling strategy (criterion 2 in CnDRM) and the approach used in EATA [29], which discards test samples whose predictions are close to the prediction mean. However, these **two strategies differ fundamentally in context, granularity, and update regime**:
>
> In EATA, the sampling occurs at the *prediction* level (final logits), and the update is applied at **every batch**. This setting favors more challenging samples by dropping the samples whose prediction is too close to the testing sample mean.
>
> In contrast, our method targets the **Sparse TTA (STTA)** scenario, where only ~10% of test samples are selected for model updates. In such a setting, choosing **reliable and representative** samples near the **feature-level domain centroid** (early-layer statistics) leads to better stability and performance, as shown empirically in *Figure 4* and theoretically supported in *Appendix C.1.*
>
> While both methods use centroid-based criteria, they operate at different representation levels and serve different adaptation strategies: EATA prioritizes informativeness under high-frequency updates, whereas CnDRM prioritizes reliability under sparse updates. The strategies are therefore complementary, not contradictory.
>
> Regarding the integration of EATA and SNAP (as in our EATA+SNAP evaluation), we first apply SNAP’s sample selection criteria (CnDRM) to identify a reliable subset, and then apply EATA’s strategy to that subset. This design ensures compatibility between the two methods.
>
> We will clarify this distinction and sampling logic more explicitly in our final manuscript.
>
> ---
> *Reference*
>
> [r1] Y. Li, N. Wang, J. Shi, J. Liu, and X. Hou, “Revisiting Batch Normalization for Practical Domain Adaptation.” ICLR 2017.
>
> [r2] E. F. Montesuma, F. M. N. Mboula, and A. Souloumiac, “Optimal Transport for Domain Adaptation through Gaussian Mixture Models.” TMLR 2025.

---

> ### Author Response · Authors · 2025-08-07
> **Gentle reminder regarding our response**
>
> Dear Reviewer Vzgx,
>
> Thank you once again for your thoughtful and constructive review.
>
> We have carefully addressed all the questions and concerns you raised in our rebuttal. As the discussion period is coming to an end, we just wanted to gently remind you in case you haven’t had a chance to revisit our response.
>
> If there are any remaining concerns or points requiring clarification, we would be happy to elaborate.
>
> We would be grateful if you could take our responses into consideration when finalizing your evaluation.
>
> Best regards,
> Authors

---

> > ### Comment · Reviewer_Vzgx · 2025-08-08
> > **Thanks for your rebuttal**
> >
> > Thanks for your rebuttal and apologize for my misunderstanding regarding W1. I don't have any more concerns regarding the "weakness" part. I suggest the author to revise the paper with more details given in the rebuttal.
> >
> > **[NOT A WEAKNESS]** I am still a little confused about Q1. I acknowledge that the author summarized several differences which may lead to different algorithm design:
> > - SNAP focus on sparse TTA, update the model for every K forward pass, instead
> > - EATA, though also focus on "efficiency", update the model for every forward pass, with fewer samples.
> > And,
> > - SNAP uses hidden representation, while
> > - EATA uses the final logits (although you might also say this is a representation..)
> >
> > I believe the author did a good job in explaining why these two scenarios can lead to different results, but it is a little bit high-level (and maybe vague?) to claim that "EATA prioritizes informativeness under high-frequency updates, whereas CnDRM prioritizes reliability under sparse updates", and "the strategies are therefore complementary, not contradictory".
> >
> > But still, I would emphasize that this is **not a weakness**. I would like to keep my positive score. Good luck!

---

> > > ### Author Response · Authors · 2025-08-09
> > >
> > > Thank you for the clarification and your constructive feedback. We appreciate your positive assessment and will incorporate all details in the rebuttal into the final manuscript.
> > >
> > > **[Regarding Q1]** We appreciate the opportunity to clarify the fundamental difference between SNAP and EATA. SNAP selects reliable samples whose updates are stable and generalizable across a bunch of test samples—critical under sparse update regimes. In contrast, EATA explicitly drops samples whose predictions are overly aligned with the test-time mean prediction, assuming they lack discriminative value and are less effective for adaptation.
> > > - We clarified that this strategic difference stems from the assumed **_update frequency_**. EATA performs *frequent updates* and tolerates riskier or more exploratory samples, as errors can be corrected in subsequent steps. In contrast, SNAP performs *sparse updates*, each with lasting effect, so it must avoid noisy or atypical samples and instead relies on highly reliable ones for generalizable model adaptation.
> > >
> > > We hope this resolves the ambiguity. We will reflect this clarification in the manuscript.

---

### Official Review · Reviewer_aqRo · 2025-07-01

**Clarity:** 3
**Significance:** 3
**Originality:** 3
**Rating:** 5
**Confidence:** 5

**Summary:**

This paper proposes SNAP, a sparse and efficient Test-Time Adaptation (TTA) framework designed for resource-constrained edge environments. Unlike existing TTA methods that require frequent updates and high computation, SNAP maintains high accuracy while adapting with as little as 1% of the test data. It introduces two key components: (1) Class and Domain Representative Memory (CnDRM), which stores a compact set of representative samples for efficient adaptation, and (2) Inference-only Batch-aware Memory Normalization (IoBMN), which updates normalization statistics during inference using these samples.

**Questions:**

Please refer to the above.

**Ethical Concerns:**

["NO or VERY MINOR ethics concerns only"]

**Final Justification:**

Thanks for the author's rebuttal. Since other reviewers also considered raising their score, I will also keep my positive score here.

**Limitations:**

yes

**Paper Formatting Concerns:**

No issues

**Quality:**

3

**Strengths And Weaknesses:**

1. Strengths

- This paper addresses an important yet often overlooked challenge in current TTA research: the trade-off between latency and accuracy. By focusing on resource-constrained edge environments and identifying frequent adaptation as a key bottleneck, the authors highlight a novel and practical direction for the field.
- A particularly noteworthy contribution is that, after adaptation, the model often outperforms its initial performance (e.g., in Tent, EATA, and CoTTA). This is significant, as most prior TTA methods only aim to maintain the original accuracy rather than improve upon it.
- The proposed method is plug-and-play, making it easy to integrate into existing pipelines. Additionally, the extensive results in the Appendix across different adaptation rates are helpful and provide a comprehensive understanding of the method’s robustness and generalizability.

2. Weakness
- This work adopts a fixed adaptation rate/CnDRM safeguard. Can the authors provide dynamic tuning (i.e., learning this rate) and efficiency metrics?

---

> ### Author Rebuttal · Authors · 2025-07-31
>
> **Weakness 1.** Thank you for your insightful comment. The fixed adaptation rate is currently used as a baseline, and dynamic tuning of the adaptation rate would improve flexibility and responsiveness, which is future work.
>
> One direction is to use the **Z-score**, which measures how much a feature's value deviates from its mean, normalized by its standard deviation. By tracking Z-scores between consecutive batches, we can adjust the adaptation rate: if the shift in Z-score exceeds a threshold, we increase the adaptation rate for more frequent updates. Otherwise, the adaptation rate could be reduced to save computational resources. This would help the model respond quickly when there is a significant domain shift while conserving resources during stable periods.
>
> Additionally, we could incorporate **dynamic accuracy estimation**. If accuracy on a validation set starts to drop, the system could trigger an increase in the adaptation rate to improve performance. This could be done similarly to the method used in **AETTA** [r1].
>
> We will explore these ideas further to balance adaptation rate, efficiency, and performance in future work.
>
> ---
> *Reference*
>
> [r1] T. Lee, S. Chottananurak, T. Gong and S.-J. Lee, "AETTA: Label-Free Accuracy Estimation for Test-Time Adaptation", CVPR 2024.

---

> ### Author Response · Authors · 2025-08-07
> **Gentle reminder regarding our response**
>
> Dear Reviewer aqRo,
>
> Thank you once again for your thoughtful and constructive review.
>
> We have carefully addressed all the questions and concerns you raised in our rebuttal. As the discussion period is coming to an end, we just wanted to gently remind you in case you haven’t had a chance to revisit our response.
>
> If there are any remaining concerns or points requiring clarification, we would be happy to elaborate.
>
> We would be grateful if you could take our responses into consideration when finalizing your evaluation.
>
> Best regards,
> Authors

---

> > ### Comment · Reviewer_aqRo · 2025-08-08
> >
> > Thanks for the author's rebuttal. I will keep my rating.

---

> > > ### Author Response · Authors · 2025-08-09
> > >
> > > Thank you for your continued positive assessment and for confirming your rating. We appreciate your constructive feedback and will incorporate it into the final manuscript.

---

### Official Review · Reviewer_wZnu · 2025-07-02

**Clarity:** 2
**Significance:** 3
**Originality:** 3
**Rating:** 4
**Confidence:** 5

**Summary:**

This paper introduces SNAP, a sparse Test-Time Adaptation (TTA) framework designed to address the high latency and computational cost of existing TTA methods on resource-constrained edge devices. SNAP employs two key components: Class and Domain Representative Memory (CnDRM) for efficient sample selection and Inference-only Batch-aware Memory Normalization (IoBMN) for dynamic feature alignment. Extensive experiments on CIFAR-C, ImageNet-C, and real edge devices (e.g., Raspberry Pi, Jetson Nano) demonstrate significant latency reductions while maintaining accuracy. The framework integrates seamlessly with five SOTA TTA algorithms, showcasing broad applicability.

**Questions:**

Please refer to the weaknesses.

**Ethical Concerns:**

["NO or VERY MINOR ethics concerns only"]

**Final Justification:**

The authors have addressed my previous concerns. Hence, I raise my score accordingly.

**Limitations:**

Yes. The authors have addressed the limitations in Section 5.

**Paper Formatting Concerns:**

None.

**Quality:**

2

**Strengths And Weaknesses:**

Strengths:
1. The paper effectively identifies the critical challenge of TTA latency on edge devices, a gap overlooked in prior work, and provides a targeted solution with sparse adaptation.

2. CnDRM and IoBMN are innovative and combines class-domain representativeness and lightweight normalization adjustments to balance accuracy and efficiency.

3. Experiments cover diverse datasets, adaptation rates, and edge devices, with thorough ablation studies validating component contributions. Latency and memory overhead are rigorously measured, enhancing reproducibility.

4. The framework’s compatibility with existing TTA methods and minimal memory overhead (0.02–1.74%) make it highly applicable to real-world edge deployments.

Weaknesses:

1. In lines 191-196, the authors aim to determine domain-representative samples. However, after introducing the calculation of the Wasserstein distance, the text abruptly stops. It fails to elaborate on what domain-representative samples are and how to obtain them. Specifically, the definition of domain-representative samples (e.g., whether they refer to samples whose feature distributions are close to the target domain centroid) and the operational steps for selection (e.g., threshold criteria or sampling strategies based on Wasserstein distance rankings) remain unclear. This leaves readers without a complete understanding of the sample selection mechanism, potentially affecting the reproducibility and interpretability of the method.

2. Besides, in Algorithm 1 (Table), the operations of "Remove domain-centroid farthest sample" and "Update domain-centroid" are not described anywhere in the paper. Specifically, there is no explanation of the criteria for determining the "farthest sample" (e.g., whether it is based on the Wasserstein distance defined in Equation 1) or the specific mechanism for "updating the domain centroid" (e.g., momentum update parameters or frequency). These omissions make it difficult for readers to fully understand the dynamic management process of the memory pool, potentially affecting the reproducibility of the method and the interpretation of experimental results.

3. In the paper, the two key concepts of Adaptation Rate (mentioned in Line 132) and sampling ratio (mentioned in Line 134) proposed in relation to Sparse TTA are not clearly explained.

4. In the experimental section, the paper states that two networks were used for ImageNet. However, neither the caption of Table 1 nor the main text specifies which network was employed for the ImageNet data presented in Table 1.

---

> ### Author Rebuttal · Authors · 2025-07-31
>
> **Weakness1 (W1).** Thank you for pointing this out. We acknowledge that the explanation of domain-representative sampling was unclear, and we appreciate the opportunity to clarify.
>
> A *domain-representative sample* is a sample whose early-layer feature distribution lies close to the domain centroid—computed as a moving average of incoming sample features. Wasserstein distance measures the proximity between a sample’s distribution and the domain centroid. We maintain a fixed-size memory by continuously replacing the furthest (i.e., high-distance) samples with new ones that are closer to the centroid. This ensures that the memory retains only the most representative samples under dynamic distribution shifts (as described in lines 163–166, 191–196, and 201–202).
>
> In our final manuscript, we will explicitly define domain-representative samples and clearly describe the replacement mechanism based on Wasserstein distance ranking.
>
> ---
> **W2.** While "Remove domain-centroid farthest sample" and "Update domain-centroid" were described in lines 201–202 and 184–186, respectively, we acknowledge that the connection between *Algorithm 1* and the operations “Remove domain-centroid farthest sample” and “Update domain-centroid” was not sufficiently clear in the original manuscript, and we appreciate the reviewer’s attention to this point.
>
> To clarify:
> - **Farthest sample removal**: As described in lines 201–202, the “farthest sample” refers to the memory sample with the largest Wasserstein distance to the domain centroid, as defined in Equation 1 and discussed in lines 191–196. This sample is considered the least representative and is removed when the memory pool exceeds its capacity.
> - **Domain centroid update**: As noted in lines 184–186, the domain centroid is updated after every batch using a momentum-based moving average over early-layer features. The momentum coefficient is fixed at 0.9 throughout all experiments, allowing the centroid to adapt to the evolving feature distribution.
>
> We will revise the manuscript to explicitly link these procedures to *Algorithm 1* and clarify their implementation details to enhance reproducibility and reader understanding.
>
> ---
> **W3.** We apologize for the confusion.
>
> To clarify:
> - **Adaptation Rate** refers to the frequency of model updates in Sparse TTA, as described in lines 116–118, and noted in the caption of *Figure 1*. It is the ratio of batches that trigger an update over the total number of test batches.
>
> - **Sampling Ratio** denotes the proportion of samples (across complete data) that are used for model updates. This term follows the convention used in data-efficient learning literature[2,49].
>
> In cases where the memory size equals the batch size and updates occur batch-wise, these values coincide. However, they are conceptually distinct and can diverge depending on the implementation. We will clarify this in the final manuscript.
>
> ---
> **W4.** Thank you for pointing this out. The results presented in *Table 1* were obtained using **ResNet-50** for the ImageNet experiments. We will revise the main text and the table caption to explicitly state this in our final manuscript.

---

> > ### Comment · Reviewer_wZnu · 2025-08-07
> >
> > I appreciate the authors' efforts in addressing my previous concerns, which have clarified several ambiguities. I strongly urge the authors to rectify all the identified issues as specified. It is imperative that these corrections are explicitly reflected in the final version of the manuscript to ensure the clarity, reproducibility, and rigor of the work.

---

> ### Author Response · Authors · 2025-08-07
> **Gentle reminder regarding our response**
>
> Dear Reviewer wZnu,
>
> Thank you once again for your thoughtful and constructive review.
>
> We have carefully addressed all the questions and concerns you raised in our rebuttal. As the discussion period is coming to an end, we just wanted to gently remind you in case you haven’t had a chance to revisit our response.
>
> If there are any remaining concerns or points requiring clarification, we would be happy to elaborate.
>
> We would be grateful if you could take our responses into consideration when finalizing your evaluation.
>
> Best regards,
> Authors

---

> ### Author Response · Authors · 2025-08-07
>
> We sincerely appreciate the reviewer’s acknowledgment. We are fully committed to addressing all the concerns and will ensure that the corresponding revisions are thoroughly and explicitly reflected in the final version of the manuscript.

---

### Official Review · Reviewer_GNGJ · 2025-07-08

**Clarity:** 3
**Significance:** 2
**Originality:** 2
**Rating:** 4
**Confidence:** 4

**Summary:**

This paper proposes a sparse and training-free TTA framework designed for resource-constrained edge environments. It introduces two key components: (i) CnDRM, which selects representative samples for efficient adaptation, and (ii) IoBMN, an inference-only normalization strategy. SNAP achieves up to 93.12% latency reduction with less than 3.3% accuracy drop, showing potential for real-world deployment.

**Questions:**

See weaknesses and try to address all of my concerns.

**Ethical Concerns:**

["NO or VERY MINOR ethics concerns only"]

**Final Justification:**

As most of my concerns have been addressed, I will raise my score.

**Limitations:**

yes

**Quality:**

3

**Strengths And Weaknesses:**

Strengths:

1. The approach is well-motivated, clearly addressing the challenges of deploying TTA models at the edge by proposing a training-free solution that leverages class/domain representatives samples.

2. The structure of the method/work is articulated in a way that facilitates understanding for the reader.

Weaknesses

1. The paper lacks empirical evaluation over realistic datasets aligned with the motivating applications mentioned by the authors, such as autonomous driving, real-time health monitoring, or high-frame-rate video analysis. This limits the demonstration of the practical utility and scalability of the method under real-world conditions.
2. Another important concern relates to the dynamics of the model during early inference steps. Since the proposed class and domain representative memory requires accumulation over time, it is unclear how the model performs in the initial stages when batch size is small (e.g., bs=1). The paper does not provide sufficient analysis or discussion on convergence behavior or how long it takes for the memory to stabilize and yield reliable performance.
3. The core methodological components are not particularly novel. The use of memory banks for test-time adaptation has been explored in prior work, such as RoTTA[51], and the strategy of correcting Batch Normalization statistics at inference time has also been studied (e.g., in Note[6]). The authors should position their contributions more clearly by providing module-level comparisons to these related methods to highlight what is new and effective.
4. One of the key contributions of this work lies in its inference-only normalization strategy. However, training-free TTA paradigms are not new and have been widely explored in existing literature. The authors have not sufficiently discussed related works in this area, such as [1,2,3,4]. Authors are suggested to provide a more in-depth analysis to better highlight the unique contributions of their method in this aspect.
5. Several key claims are not sufficiently substantiated by empirical evidence. For instance, the statements in lines 62–64 and 157–159 make strong assertions about the effectiveness of the method in correcting skewed feature distributions and selecting diverse and reliable samples without ground-truth supervision. However, the paper does not include specific ablations or diagnostic experiments to validate these points.


[1] AdaNPC: Exploring Non-Parametric Classifier for Test-Time Adaptation

[2] Parameter-free Online Test-time Adaptation

[3] Unraveling Batch Normalization for Realistic Test-Time Adaptation

[4] Dynamically Instance-Guided Adaptation: A Backward-free Approach for Test-Time Domain Adaptive Semantic Segmentation

---

> ### Author Rebuttal · Authors · 2025-07-31
>
> **Weakness1 (W1).** Thank you for the insightful comment. We have conducted additional evaluation on the **HARTH**[r1] dataset, a realistic *health monitoring* benchmark consisting of three-axial sensory inputs and human activity recognition labels. Unlike our main evaluations (which focus on 2D vision and corruption-based domain shifts), HARTH introduces a **distinct domain shift** caused by ***sensor positioning*** and ***user variation***, providing a complementary scenario to validate SNAP.
>
> We compared the naïve Sparse TTA (STTA) baseline and our proposed SNAP module under an adaptation rate (AR) of 0.1, using Tent[46] and SAR[30] as TTA methods. The source domain is the data collected from the back (15 users), and the target domain is the data collected from the thigh (from the remaining 7 users). Table A shows that SNAP improves accuracy even with sparse updates, demonstrating its effectiveness under realistic shifts.
>
> *Table A. Performance on HARTH[r1].*
> | Method | |Average Accuracy |
> |-|-|:-:|
> | Tent | naïve STTA (AR=0.1) | 19.64 |
> | | **SNAP (AR=0.1)**|**30.67**|
> | SAR | naïve STTA (AR=0.1) | 21.10 |
> | |**SNAP (AR=0.1)**|**26.63**|
>
> For other motivating scenarios, such as autonomous driving and high-frame-rate video, we believe that these are aligned with the main evaluations of ImageNet-C's 2D frame-based corruptions. Given the shared methodological assumptions, we expect similar gains for SNAP. We will conduct additional validation on these scenarios and include the results as resources permit.
>
> ---
> **W2.** Sparse TTA scenarios, including SNAP, require at least one adaptation (model update) cycle before achieving effective performance, regardless of batch size. After the first sparse update, SNAP functions as intended and exhibits robust performance. Specifically, our Class and Domain Representative Memory (CnDRM) is designed to be batch-size agnostic. For domain-representative sampling, the domain centroid is computed using a moving average of early-layer feature statistics, stabilizing quickly even under bs=1. For class-representative sampling, CnDRM selects samples based on per-sample prediction confidence and class balance. While small batches can slow down class distribution balancing (particularly under skewed distributions), we found empirically that this has a limited effect on overall adaptation (*Appendix C.12*). In practice, as shown in *Appendix C.13*, SNAP maintains robust performance even in bs=1 scenarios. We will clarify this and add such visualizations in our final manuscript.
>
> ---
> **W3.** We agree that memory-based sampling and BN correction have been partly explored in prior TTA work. Our key contribution lies in re-designing these components specifically for the sparse update regime, which differs significantly in its constraints and requirements.
>
> **CnDRM vs. Prior Memory Sampling (RoTTA[51], NOTE[6], SAR[30], EATA[29]):**
> Previous methods assume frequent adaptation (≥50% of test samples used for update) and focus on selecting time-wise fair and low-entropy samples (e.g., RoTTA, NOTE, SAR, EATA). In contrast, our target is *Sparse TTA* (e.g., 10% adaptation rate), where such filtering leaves too few useful samples for effective adaptation. Hence, SNAP's CnDRM (1) does **not** rely on low-entropy filtering, and (2) selects *representative* samples based on domain centroids and class confidence, enabling efficient adaptation with minimal latency. We provided theoretical analysis (*Appendix C.1*) and ablation results (*Table 2, Appendix D.4*) that showed our entropy-based filtering performs worse than even random selection under sparse settings.
>
> **IoBMN vs. Instance-wise BN Correction (NOTE):**
> NOTE corrects BN statistics per instance, incurring high latency proportional to sample count. In contrast, our IoBMN uses the memory (CnDRM)-derived domain-class statistics to correct batch BN stats efficiently, maintaining compatibility with batch inference. Additionally, we shift the memory stats toward batch stats to mitigate skew from sparse updates. This design not only improves computational efficiency but also leverages adaptation-involved samples for normalization, generating performance gains (*Table 2, Appendix D.4*).
>
> We will emphasize these distinctions in the manuscript to better position the novelty of each module.
>
> ---
> **W4.** Thank you for the pointers to the related work. In our manuscript (*Intro* and *Appendix C.8*), we included a discussion and empirical comparison against recent and representative **training-free TTA** methods T3A [14] and FOA [28]. These studies and your suggested papers are standalone TTA algorithms that generally augment a memory bank for a KNN classifier [r2], adapt model outputs directly through regularization [r3], improve batch normalization statistics to handle reduced class diversity [r4], and use instance-guided statistics and prototypes for semantic segmentation [r5].
>
> In contrast, **SNAP is a modular extension that addresses the challenge of making computationally intensive, backpropagation-based TTAs fast and efficient for edge deployment.** These model-adaptation methods are widely used due to their ability to perform deeper, feature-level updates, which offer improved robustness under complex and evolving distribution shifts. However, their practicality is often limited by latency and resource constraints. SNAP makes such methods viable in real-world scenarios by enabling them to operate sparsely, thus preserving their adaptation benefits while significantly reducing computational cost.
>
> ---
> **W5.** We address the two claims as follows.
>
> **Feature distribution correction:** In sparse update settings, memory-based statistics are computed from a small and outdated subset of data, leading to skewed normalization. To mitigate this, IoBMN shifts the memory statistics toward each incoming batch direction, correcting skew adaptively. As the adaptation rate decreases (i.e., updates become sparser), this correction becomes more effective—demonstrated by consistent performance gains when IoBMN is added (*Table 2, Appendix D.4*). Compared to prior BN correction-only methods, including EMA [27,39,24,46,30,48,51], IoBMN outperforms them under sparse TTA by leveraging representative memory statistics with minimal overhead.
>
> **Diverse and reliable sample selection:** Diversity is achieved by maintaining prediction-balanced sampling (as validated in prior works[51,6]). Reliability is ensured by selecting high-confidence samples, whose pseudo-label accuracy is empirically verified to be significantly better than random (Appendix C.3). The ablation in the Evaluation section ("Contribution of SNAP individual components") further supports this: CRM outperforms Rand, and CnDRM+IoBMN outperforms CnDRM alone.
>
> To further back up, we will incorporate additional validations, such as the analysis of the proportion and magnitude of IoBMN correction, as well as the distribution of memory samples, in our final manuscript.
>
> ---
>
> *Reference*
>
> [r1] Logacjov A, Bach K, et al. "Harth: A human activity recognition dataset for machine learning", Sensors, 21(23), 2021
>
> [r2] Zhang, Yifan, et al. "AdaNPC: Exploring Non-Parametric Classifier for Test-Time Adaptation", ICML 23
>
> [r3] Boudiaf, Malik, et al. "Parameter-free Online Test-time Adaptation", CVPR 22
>
> [r4] Su, Zixian, et al. "Unraveling Batch Normalization for Realistic Test-Time Adaptation", AAAI 24
>
> [r5] Wang, Wei, et al. "Dynamically Instance-Guided Adaptation: A Backward-free Approach for Test-Time Domain Adaptive Semantic Segmentation", CVPR 23

---

> > ### Comment · Reviewer_GNGJ · 2025-08-09
> >
> > Thanks for your response. As most of my concerns have been addressed, I will raise my score.

---

> ### Author Response · Authors · 2025-08-07
> **Gentle reminder regarding our response**
>
> Dear Reviewer GNGJ,
>
> Thank you once again for your thoughtful and constructive review.
>
> We have carefully addressed all the questions and concerns you raised in our rebuttal. As the discussion period is coming to an end, we just wanted to gently remind you in case you haven’t had a chance to revisit our response.
>
> If there are any remaining concerns or points requiring clarification, we would be happy to elaborate.
>
> We would be grateful if you could take our responses into consideration when finalizing your evaluation.
>
> Best regards,
> Authors

---

> ### Author Response · Authors · 2025-08-09
>
> Thank you for your thoughtful follow-up on our rebuttal. We are pleased that most of your concerns have been addressed and will ensure that the relevant clarifications are incorporated into the final manuscript.

---

### Note · Authors · 2025-08-16

We thank all reviewers for their constructive feedback and engagement throughout the process. **All four reviewers recognized key strengths of our work**, including (1) addressing an often-overlooked yet central TTA challenge—balancing accuracy and latency in resource-constrained deployments, (2) methodological simplicity and extensibility, and (3) empirical rigor across diverse datasets, adaptation rates, and edge devices.

Following the rebuttal, two reviewers (*aqRo, Vzgx*) maintained **positive scores**, highlighting practicality, clarity, and robustness. One reviewer (*GNGJ*) **raised** the initial borderline-reject score after the rebuttal, noting that **most concerns had been addressed**. The remaining reviewer (*wZnu*) mentioned our rebuttal clarified ambiguities and suggested incorporating our rebuttal in the final manuscript; we appreciate the important detailed clarifications requested and will **ensure these points are reflected in the final manuscript**.

- For *aqRo*, we discussed future work on *dynamically tuning the adaptation rate*, e.g., using Z-score–based shift detection or validation accuracy monitoring, to further balance efficiency and performance.

- For *Vzgx*, we (i) confirmed IoBMN requires no extra forward pass, (ii) explained why memory-derived BN statistics are preferable to batch-only updates, (iii) clarified Gaussian/diagonal assumptions in *Equation 1* and explanations in *Equations 2–3*, and (iv) differentiated our feature-level, sparse-update sampling from *EATA*’s prediction-level, high-frequency strategy.

- For *GNGJ*, we (i) added new experiments on *HARTH* (real-world health monitoring) demonstrating **consistent improvements under sensor domain shift**, (ii) clarified early-stage performance with bs=1, (iii) distinguished our approach from prior memory/BN methods (e.g., *RoTTA*, *NOTE*), and (iv) provided further empirical support for feature correction and sample selection reliability.

- For *wZnu*, we (i) clarified domain-representative sampling and its Wasserstein-distance-based memory updates, (ii) detailed farthest-sample removal and centroid updates in *Algorithm 1*, (iii) clarifying *Adaptation Rate vs. Sampling Ratio*, and (iv) specified the ImageNet backbone (*ResNet-50*). **These will be clearly documented in the final manuscript**.

We believe all concerns have been addressed and will incorporate the discussed clarifications and improvements into the final manuscript.

---

### Decision · Program_Chairs · 2025-09-17

**Decision:**

Accept (poster)

**Comment:**

The paper presents a test-time adaptation method which is demonstrated to work on edge devices. The paper received 2 accept and 2 borderline accept votes. The reviewers appreciate the work for addressing an important and practical problem (from real-world deployment perspective) and for the comprehensive experiments. The reviewers raised some concerns in their original reviews and the rebuttal provides responses to most of them, which the reviewers largely seem to be satisfied with. From my own reading of the paper and considering the reviews, rebuttal, and discussion, I agree with the general consensus of the reviewers. I recommend the paper for acceptance. The authors are also advised to incorporate the changes they promised to make in the rebuttal.